

# Mineralogy and physicochemical features of Saharan dust wet deposited in the Iberian Peninsula during an extreme red rain event

Carlos Rodriguez-Navarro, Fulvio di Lorenzo, and Kerstin Elert

Dept. Mineralogy and Petrology, University of Granada, Fuentenueva s/n, 18002 Granada, Spain

*Correspondence to*: Carlos Rodriguez-Navarro (carlosrn@ugr.es)

**Abstract.** The mineralogy and physicochemical features of Saharan dust particles help to identify source areas and determine their biogeochemical, radiative and health effects, but their characterization is challenging. Using a multianalytical approach, here we characterized with unprecedented level of detail the mineralogy and physicochemical properties of Saharan dust particles massively wet deposited (~18 g m$^{-2}$) following an extreme "red rain" event triggered by a north
African cyclone that affected the southern Iberian Peninsula during February 21-23, 2017. Abundant palygorskite and illite, and relatively high carbonate contents, well-known northern and north-western Saharan dust indicators, along with low chlorite content and significant amounts of smectites and kaolinite, whose abundance increases southwards in the western Sahara, complemented by satellite imagery and back/forward trajectories, show that the most probable dust source areas were (i) south/central Algeria, north Mali and northwest Niger, and (ii) north Algeria, south Tunisia and north-west Libya.
Scanning and transmission electron microscopy analyses, including Z-contrast high angle annular dark field (HAADF) imaging and analytical electron microscopy (AEM) show that clay minerals include abundant structural Fe (57 % of the total Fe) and typically form nanogranular aggregates covered or interspersed with amorphous/poorly crystalline iron oxyhydroxide nanoparticles (ferrihydrite), which account for 28 % of the free Fe, the rest being goethite and hematite. These nanogranular aggregates tend to form rims lining large silicate and carbonate particles. Such internally mixed iron-containing
phases are main contributors to the observed absorption of solar and thermal radiation, and along with the abundant coarse/giant particles (>10 µm), strongly affect the dust direct radiative forcing. The lack of secondary sulfates in aggregates of unaltered calcite internally mixed with clays/iron-rich nanoparticles shows that iron-rich nanoparticles did not form via atmospheric (acid) processing but were already present in the dust source soils. Such iron-rich nanoparticles, in addition to iron-containing clay (nano)particles, are an important source for bioavailable (soluble) iron. The dust particles are a potential
health hazard, specially the abundant and potentially carcinogenic iron-containing palygorskite fibers. Ultimately, we show that different source areas are activated over large desert extensions, and large quantities of complex dust mixtures are transported thousands of kilometers and wet-deposited during such extreme events, which thwart any other Saharan dust event affecting south-western Europe. The past, present, and future trends, as well as impacts, of such extreme events must be taken into account when evaluating and modeling the manifold effects of the desert dust cycle.



# 1 Introduction

Aeolian erosion of semi-arid and arid desert surfaces contributes to an estimated ~ 1,000 to 3,000 Tg yr$^{-1}$ global emission of mineral dust aerosol (Goudie and Middleton, 2001; Engelstaedter et al., 2006; Cakmur et al., 2006). The impact of desert dust is enormous (Gieré and Querol, 2010): (i) it directly affects atmospheric radiative balance due to scattering and

absorption of solar and terrestrial radiation, thereby affecting atmospheric dynamics and climate (Carlson and Benjamin, 1980; Tegen and Lacis 1996: Ramanathan et al., 2001; Tegen, 2003; Balkanski et al., 2007). Indirectly, it also affects climate via its potential for altering atmospheric physics/microphysics (i.e., acting as nuclei for liquid and solid cloud droplets) and the hydrological cycle (Ramanathan et al., 2001). Such direct and indirect forcings strongly depend on mineral dust physicochemical properties, including particle size, shape and composition/mineralogy (Tegen, 2003; Lafon et al., 2006;

Formenti et al., 2011; Mahowald et al., 2014; Zhang et al., 2015; Kok et al., 2017); (ii) it supplies sediments to downwind marine and continental areas, affecting the surface albedo of the later, whereas dust entrainment in source regions is a major erosive agent, strongly affecting soil quality (Goudie and Middleton, 2001); (iii) it also supplies key micronutrients (e.g., iron and phosphorous) to distal ocean and inland water environments, directly affecting the C cycle via stimulated bioproductivity, and indirectly affecting climate via atmospheric $CO_2$ sequestration (Jickells et al., 2005; Raiswell and

Canfield, 2012); (iv) it is involved in a range of heterogeneous reactions with manifold implications (Usher et al., 2003). For instance, carbonates in Saharan dust increase the pH of precipitation acting as a buffer for acid rain in Europe (Loÿe-Pilot et al., 1986); (v) desert dust storms are a hazard with detrimental effects on transportation (e.g., reduced visibility), infrastructure, and (solar) energy generation (Middleton, 2017), also causing soiling and discoloration of monuments (Comite et al., 2017); (vi) desert mineral dust is a health hazard to humans (Karanasiou et al., 2012; Goudie, 2014). Exposure

to desert dust (particulate matter with size < 10 μm, $PM_{10}$ and/or with size < 2.5 μm, $PM_{2.5}$) has been associated with morbidity and premature death due to dust-related (or enhanced) cardiovascular and respiratory problems (Perez et al., 2008), as well as several diseases related to dust-borne microorganisms (short-term effects) (Griffin, 2007). In addition, silicosis/pulmonary fibrosis (desert lung) and cancer-related illnesses have been associated with desert dust exposure (long term effects) (Giannadaki et al., 2014).

25          More than half of the global mineral dust aerosol comes from northern Africa (Sahara/Sahel) (Goudie and Middleton, 2001; Prospero et al., 2002; Engelstaedter et al., 2006), with an estimated ~700−1600 Tg of Saharan dust exported yearly across the Mediterranean sea to Europe and the near-East, the Red sea to the near-East and Asia, and the North Atlantic ocean to the Americas (D'Almeida, 1986; Prospero, 1996; Goudíe and Middleton, 2001). Although most Saharan dust is transported across the Atlantic Ocean (Carlson and Prospero, 1972), an estimated 80–120 Tg yr$^{-1}$ is

transported northward across the Mediterranean sea to Europe (D'Almeida, 1986). It has been pointed out that the strength of the Saharan dust input increased since ca. mid 20$^{th}$ century due to recurrent droughts in North Africa (Prospero and Lamb, 2003), anthropogenic-induced desertification (Moulin and Chiapello, 2006), changes in land use, including an increase in cultivable lands in the Sahel region (Mulitza et al., 2010), and other larger-scale phenomena (e.g., atmospheric circulation





patterns and/or climate change) (Sala et al., 1996; Moulin et al., 1997). Nonetheless, there is evidence for significant seasonal to decadal variability of Saharan dust strength. Indeed, since the end of the 1980s, a trend towards decreasing Saharan dust across the tropical North Atlantic has been reported (Ridley et al., 2014; Evans et al. 2016). In contrast, the strength of Saharan dust affecting southern Europe has reportedly increased in recent decades (Antoine and Nobileau, 2006),

with dust plume intrusions being currently rather common (Escudero et al., 2005; Avila et al., 2007; Titos et al., 2017). They lead to both dry and wet deposition of mineral dust (Escudero et al., 2005). Wet deposition typically occurs as "red rain", "dust rain", "blood rain" or "muddy rain", events that periodically and persistently affect southern Europe (Prodi and Fea, 1979), and most particularly the Iberian Peninsula (Sala et al., 1996; Avila et al., 1997; White et al., 2012), the European region closest to North Africa. Although known since ancient times (Gieré and Querol, 2010), red rain events have

experienced a remarkable increase in their frequency and intensity over the last decades (Sala et al., 1996; Escudero et al., 2005; Fiol et al., 2005; Avila et al., 2007). In some cases they are extreme, with 10−40 g m$^{-2}$ of dust deposited after a single red rain event (Avila et al., 1997; 2007; Fiol et al., 2005), thwarting the average yearly Saharan dust deposition in south-western Europe, estimated to be about 3−14 g m$^{-2}$ (Goudie and Middleton, 2001). This was the case of the last extreme red rain event that took place in the area of Granada (south of Spain) in February 21$^{st}$–23$^{rd}$, 2017.

The global significance and impact of desert-derived mineral dust aerosol has attracted extensive research focused on analyzing dust composition, mineralogy, physical properties, sources, and entrainment-transport-deposition mechanisms and patterns (see reviews by Goudie and Middleton, 2001; Prospero et al., 2002; Scheuvens et al., 2013). Data from these studies have contributed to a better understanding and modeling of the dust cycle, as well as its impact on global atmospheric dynamics, climate, and biogeochemical cycles. However, there are several aspects of the current knowledge that are far from

complete. This is the case of the physicochemical and mineralogical features of Saharan dust, which are highly variable and event-specific and, therefore, poorly constrained. Although it has been acknowledged that the mineralogy and the physicochemical features of desert dust particles are relevant factors to be considered in modeling dust and climate, they are in general poorly constrained (Formenti et al., 2011; Titos et al., 2017) and have been largely ignored in most models (Krueger et al., 2004). This is likely due to the limited amount of studies dedicated to the detailed characterization of desert

dust mineralogy and physicochemical features, particularly those of individual particles (Jeong et al., 2016).

Using a multianalytical approach, here we studied Saharan dust samples collected immediately after the extreme red rain event that took place in Granada (37.17806°N, 3.60083°W, ~680 m above sea level, a.s.l.) on February 21$^{st}$–23$^{rd}$, 2017. One specific objective of this study was to analyze in detail the content, physical and textural properties, mineralogy, and composition -including individual particles- of the clay fraction (ϕ < 2 µm). This is typically a major fraction in desert dust,

is considered responsible for most of the scattering of sun light, includes most of the bioavailable iron, and due to its long atmospheric residence time is the one that can affect most distant locations (Tegen and Lacis, 1996; Sokolik and Toon, 1999; Lafon et al., 2006; Journet et al., 2008; Formenti et al., 2014a,b; Jeong and Achterberg, 2014; Jeong et al., 2016). Another goal of our study was to compare the mineralogy and physicochemical properties of the clay fraction with those of the two other relevant size fractions: i.e., sand and silt, an aspect that has been generally neglected in previous studies. It should be



noted that such an analysis is complex and tedious, and requires a relatively large amount of sample. The studied red rain event thus offered a unique opportunity to collect a sufficient amount of material from a single Saharan dust event as to perform a detailed multianalytical study of wet-deposited desert dust. In addition, the analysis of the synoptic scale meteorological conditions during this event, and the identification of transport routes and potential dust source areas

(compatible with the results of the mineralogical analysis) were performed using satellite imagery, synoptic reanalysis, dust forecast, and air masses backward/forward trajectory modeling. Ultimately, we strived to shed light on the potential biogeochemical, radiative and health effects that such extreme Saharan dust events can have locally as well as globally.

## 2 Methods

### 2.1 Synoptic situation, satellite data, dust transport modeling and backward/forward trajectories

We used NASA/NOAA Suomi NPP (National Polar-orbiting Partnership) satellite`s VIIRS (Visible Infrared Imaging Radiometer Suite) and Terra and Aqua satellites' MODIS (Moderate Resolution Imaging Spectroradiometer) true color imagery to analyze dust transport from the Sahara towards the Iberian Peninsula. This information was complemented by a post-hoc synoptic analysis of the evolution of geopotential height and wind field at ground level and 850 hPa over north-west Africa and south-west Mediterranean areas. We also analyzed the high level (300 hPa) synoptic conditions preceding

(and triggering) this event. The reanalysis was performed using NOAA´s NCEP/NCAR Reanalysis Data resource. Backward and forward trajectories of air masses were calculated using NOAA´s Hybrid Single Particle Lagrangian Integrated Trajectory (HYSPLIT) atmospheric transport and dispersion modeling system (Stein et al., 2015). We used 3 days multiple (25) backwards trajectories with end-point at 00.00 UTC (February 23$^{rd}$, 2017) over Granada at 50, 500 and 1000 m above ground level (AGL). We only report results for 500 m AGL because no significant differences related to end-point height

were observed. Multiple (25) forward trajectories, starting at 50 or 500 m AGL, 72 h or 42 h before the main dust rain event (February 23$^{rd}$, 00.00 UTC) were computed for selected point source locations. No significant differences in trajectories were observed depending on height at starting point. Modeling by the Barcelona Dust Forecast (BDF) (Pérez et al., 2011) was used to evaluate intercontinental dust transport and load (resolution 0.1° x 0.1°) for the days of the dust event. The model, which provides short to medium-range dust forecasts for global and regional domains, has been developed by the Barcelona

Supercomputer Center (BSC) and NASA, and is fully embedded into the Non-hydrostatic Multiscale Model (NMMB) developed at NCEP (Pérez et al., 2011). Ground level measurement of $PM_{10}$ load in Granada city during the dust event were included in this study. Data were collected at two stations within the city of Granada: (i) the Almanjayar and (ii) the Palacio de Congresos weather stations owned and operated by the Consejería de Medioambiente of the Junta de Andalucía (Spain).

### 2.2 Dust samples collection and processing for analysis

Wet deposited dust was collected at 3 different locations in the urban area of Granada: (i) roof of the Science Faculty of the University of Granada (37.1796°N, 3.6096°W); (ii) Fuentenueva area (nearby the Science Faculty); and (iii) the Albayzin




neighborhood (37.1792°N, 3.5945°W). In all cases, dry dust was collected (scrapped) directly from clean horizontal surfaces and/or clean ceramic rain/dust collection dishes (21 cm in diameter) with the aid of a spatula and/or a brush. Dust collection was performed within the next 24 h after the red rain event. Collection areas were measured and the sample mass was determined using a Sartorius Acculab Atilon microbalance (error ±0.001 g) after sample drying (60 °C for 24 h).

5       For clay mineralogical analysis, ~2 g of collected dust were subjected to carbonate elimination using 0.2 N acetic acid. Following complete dissolution of the carbonates, the supernatant was eliminated and the dust residue was subjected to rinsing in DI water (5 times). To favor particle dispersion, the dust sample was dispersed in 0.1 g L$^{-1}$ sodium hexametaphosphate solution. Following 10 min sonication of this dispersion, the sand ($\phi > 62.5$ μm), silt (2 μm $< \phi < 62.5$ μm), and clay fractions ($\phi < 2$ μm) were hydrodynamically separated using the Stokes's law without (silt and sand fractions)

and with (clay fraction) the use of a centrifuge (Kubota KS 8000, set at 1000 rpm / 5 min).

      Once the silt and clay fractions were separated, small aliquots of the aqueous dispersions (~2 ml) were deposited on glass slides and dried under room conditions in order to prepare oriented aggregates for XRD analysis. For the silt and clay fractions 4 oriented aggregates per sample were prepared: untreated (hereafter noted air-dried; AD), solvated at 60 °C for 72 h in a saturated atmosphere of ethylene glycol (sample noted EG), or dimethyl sulfoxide (sample noted DMSO), and one

hour heat-treated in an electric furnace set at 550 °C (sample noted 550 °C). These standard treatments were done to facilitate the identification of the different phyllosillicates (clays) in these two size fractions (Moore and Reynolds, 1989).

      The total amount of carbonates was determined by calculating the mass difference between the initial non-decarbonated (bulk) sample and the final decarbonated fractions (silt+sand+clay).

**2.3 Analysis of dust**

The mineralogy of solids was determined by x-ray diffraction (XRD) on a PANalytical XPert Pro diffractometer equipped with Ni filter. Measurement parameters were: Cu Kα radiation, wavelength, λ = 1.5405 Å, 45 kV, 40 mA, 3 to 70 °2θ exploration range, steps of 0.001 °2θ, and goniometer speed of 0.01 °2θ s$^{-1}$. Powders were back loaded on sample holders to minimize orientation effects, whereas oriented mounts of the clay and silt fractions were directly subjected to XRD analysis without further preparation. Mineral phases were identified by comparison with JCPDS powder spectra (Joint Committee on

Powder Diffraction Standards). Semiquantitative phase analysis was performed by the Rietveld method (Rietveld, 1969) using HighScore Plus 2.1.d software from PANalytical. However, due to the high residuals obtained after (semi)quantitative Rietveld analysis (associated with the large number of phases present in the dust samples), it was necessary to also perform semiquantitative analyses using the RIR (reference intensity ratio) method. For this task, we used both reported (JCPDF files) and experimental RIR values determined in house. Our experimental RIR values were in general comparable to those

reported in JCPDS files of the corresponding phases. For the case of clay minerals (oriented mounts), however, the high dispersion of RIR values reported in the corresponding JCPDS files, prompted us to use our own RIR values: this was the case of phases such as palygorskite, smectites (beidellite and/or montmorillonite, and mixed-layer clays), illite, chlorite and kaolinite. Overall, more consistent results were obtained using the RIR method than the Rietveld method. In particular, the



RIR method enabled us to get total carbonate phase contents that accurately matched the values determined independently by mass analysis of samples prior and after carbonate elimination following acid attack, and by thermogravimetry (TG).

TG and differential scanning calorimetry (DSC) analyses of the bulk samples (i.e., not subjected to any size separation or carbonate elimination) were performed simultaneously on a Mettler-Toledo TGA/DSC1. Samples of ~40 mg were placed in Al crucibles and analyzed both in flowing air and $N_2$ (50 mL min$^{-1}$) at a heating rate of 20 °C min$^{-1}$ (25 °C to 950 °C). We used both an oxidizing (air) and an inert ($N_2$) atmosphere in order to determine the total content of organic matter in the dust samples.

Chemical analysis of the dust samples was performed by means of inductively coupled plasma-optical emission spectrometry (ICP-OES, Perkin-Elmer Optima 8300) for major elements and ICP-mass spectrometry (ICP-MS, Perkin-Elmer Sciex Elan 5000 MS) for minor and trace elements. Measurements were taken in triplicate using rhenium and rhodium as internal standards. The instrumental error is ±2% and ±5% for elemental concentrations of 50 ppm and 5 ppm respectively. Note that acid digestion of dust samples (~100 mg sample mass) using HF and $HNO_3$ required for ICP analyses results in the partial loss of Si (Formenti et al., 2011). To properly measure Si concentration, additional x-ray fluorescence (XRF) analysis of major (and some minor/trace) elements was performed on a Bruker AXS S4 Pioneer. Samples were prepared as fused beads by melting about 1 g of bulk dust with lithium tetra borate flux. The quality of the analysis was monitored with reference materials showing high precision with $1\sigma = 1.0$–3.4% on 16 data-sets at the 95% confidence level.

Additional compositional and micro-/nano-structural features of dust particles were determined by means of transmission electron microscopy (TEM, Phillips CM20, operated at an acceleration voltage 200 kV and FEI Titan, operated at 300 kV). Bulk powder samples were dispersed in ethanol, sonicated 30 s, and fished with holey Formvar™ C-coated Cu grids. TEM observations were performed using a 30 µm objective aperture. SAED patterns were collected using a 10 µm aperture, which allowed collection of diffraction data from a circular area ~0.2 µm in diameter. Quantitative analytical electron microscopy (TEM-AEM) analyses were performed in STEM mode using the CM20 with an EDAX solid-state energy dispersive x-ray (EDX) detector. A scan window of ~$20 \times 100$ nm was used for the analysis of individual particles. In the case of the Titan, compositional analyses were performed in STEM mode using a High Angle Annular Dark Field (HAADF) detector. In this later case compositional maps of selected areas were obtained, which also enabled collection of data from individual mineral particles. Albite, biotite, muscovite, spessartine, olivine and titanite standards were used to obtain $k$-factors for the transformation of intensity ratios to concentration ratios according to Cliff and Lorimer (1975).

Textural features (particularly, grain size and shape of larger particles) were studied using a field emission scanning electron microscope (FESEM, Auriga, Zeiss) equipped with EDX microanalysis. Samples were carbon coated prior to analysis. Gas sorption measurements were performed on a Micromeritics TriStar 3000. The surface area (BET method) and pore size distribution (BJH method) (Sing et al., 1985) of dust samples was determined by means of $N_2$ adsorption/desorption at 77 K following degassing for 3 h at 80 °C under vacuum on a Micromeritics FlowPrep device. The particle size distribution (PSD) of dust dispersions in water was determined by laser scattering on a Malvern Hydro 2000µp equipment.




Further details on the phases and compositional features of the dust samples were studied using Fourier transform infrared spectroscopy (FTIR; JASCO 6200); frequency range 400−4000 cm$^{-1}$ (2.5−25 μm wavelength range), with 2 cm$^{-1}$ spectral resolution. The FTIR used is equipped with an attenuated total reflectance (ATR) device for spectra collection without sample preparation (i.e., to minimize artifacts). Additional spectral information of the bulk dust samples, as well as

the different size fractions, was gathered by ultraviolet-visible-near infrared (UV-Vis-NIR) diffuse reflectance spectroscopy (DRS) analyses on a VARIAN CARY-5E spectrometer equipped with a 110 mm diameter integrating sphere coated with Halon (Labsphere, Inc.). Analyses were performed in the spectral range 200−2000 nm with 1 nm resolution. The second derivative of the absorbance curves in the 300−800 nm spectral range was obtained using Microcalc Origin 6.0 computer code (this procedure included pre- and post-smoothing using 20 points adjacent averaging). From the amplitude of the

minima at ~435 nm and ~565 nm, corresponding to absorption by goethite and hematite, respectively, the relative concentration of these two phases was calculated following the procedure outlined by Scheinost et al. (1998).

## 3 Results and Discussion

### 3.1 The extreme winter Saharan dust event

On February 20$^{th}$ and 21$^{st}$, a large Saharan dust plume coming from northern Africa (off the coast of Algeria and Morocco)

crossed the Alboran sea (western Mediterranean) and wiped the southern portion of the Iberian Peninsula in a counter clockwise N-NW motion (Fig. 1a). This initial dust intrusion, which was clearly visible in Suomi VIIRS as well as in Terra and Aqua MODIS true color imagery on February 21$^{st}$ (on February 20$^{th}$ the southern portion of the Iberian peninsula was covered by clouds, thereby precluding the direct observation of the dust plume), caused a minor red rain event in the Granada area during the night of February 21$^{st}$. Subsequently, a more massive African dust plume penetrated in the southern

part of the Iberian peninsula during February 22$^{nd}$ (Fig. 1a), leading to significant wet deposition in Granada (and in almost all areas in the southern portion of the Iberian Peninsula) during the night of February 22$^{nd}$ and the early hours of February 23$^{rd}$. The volume of precipitation was minor (~1−2 mm), but the amount of dust deposited was massive: we measured an average of 18±8 g m$^{-2}$ deposited in the city of Granada. Note that the relatively high scattering in the amount of deposited dust was due to orientation differences among the different collection sites (i.e., shielding effects of nearby buildings and/or

vegetation). Streets, houses and cars across the city appeared covered by reddish-brown mud and in nearby Sierra Nevada the snow cap displayed a dramatic reddish-brown discoloration (compare the before and after satellite images of Sierra Nevada shown in Fig. 1c and 1d). Such an amount of deposited Saharan dust is about 4 times higher than the average input of 5.1 g m$^{-2}$ yr$^{-1}$ measured by Avila et al. (1997) at NE Spain for the period 1981−1992. Remarkably, Avila et al. (1997) reported that just two red rain events accounted for nearly 62% of the total desert dust mass deposited in the area over 11

years: the extreme events in November 1984 and in March 1991 delivered 16.4 g m$^{-2}$ and 19.4 g m$^{-2}$, respectively. These values are in very good agreement with the value reported here.



**Figure 1.** The extreme Saharan dust and red rain event. (a) NASA SUOMI NPP/VIIRS true color images of the Iberian peninsula and northern Africa during the days of the extreme red rain event. On February 19$^{th}$ dust plumes were visible over north Mali, northwest Niger and south-central Algeria, which moved north/north-east reaching the western Mediterranean on February 20$^{th}$, and wiping in a counter-clockwise movement the southern portion of the Iberian peninsula on February 21$^{st}$. On February 22$^{nd}$ and 23$^{rd}$ dust plumes coming from north Algeria (and Tunisia) continued to sweep the Iberian peninsula. Finally on February 24th, the Saharan dust plume moved towards the central Mediterranean. The blue dashed arrow shows the path of the dust plume. (b-c) NASA´s Terra MODIS true color images of Sierra Nevada (Spain) (b) before (February 15$^{th}$, 2017, 11.30 UTC) and (c) after (February 26$^{th}$ 2017, 11.30 UTC) the extreme Saharan red rain event. Note the brownish color of the snow cap after the red rain event. Images credit: NASA/Earthdata (http//: gibs.earthdata.nasa.gov). (d) Ground level PM10 concentration at Granada city for the days prior, during and after the red rain event (the red and blue lines refer to Almanjayar and Palacio de Congresos stations, respectively). Data from Consejería de Medio Ambiente, Junta de Andalucía.

During the last hours of February 22$^{nd}$ and early hours of February 23$^{rd}$, a maximum ground level $PM_{10}$ concentration of 275 µg m$^{-3}$ was measured in the Granada urban area (Fig. 1d), a value more than five times higher than the daily limit (50 µm m$^{-3}$) established by Directive 2008/50/EU of the European Union. Surface $PM_{10}$ values rapidly decreased to values ~50−75 µg m$^{-3}$ following the red rain event. Note that Saharan dust events with daily $PM_{10}$ >100 µg m$^{-3}$ are

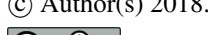

considered "extreme" and are infrequent (1 % of all dust events) in the western Mediterranean, but more frequent (2−5 %) in the eastern Mediterranean (Pey et al., 2013). Nonetheless, for the case of the Iberian peninsula, more than 9 red rain events with a dust deposition > 1 g m$^{-2}$ per event (i.e., extreme events) were reported for the period 1984−1993 (Avila et al.,1997), and 4 more between 1996 and 2002 (Avila et al., 2007).

NMMB-BDF modeling showed massive dust plumes initially coming from south Algeria and north Mali (and north-west Niger), with subsequent contributions from northern Algeria, southern Tunisia and western Libya, entering the Iberian peninsula on February 20$^{th}$, and wiping the southern portion of Spain in two successive waves. The second wave was more intense than the first, with an aerosol optical depth, AOD at 550 nm of 1.6−3.2. Finally, the dust plume shifted course towards the central Mediterranean on February 24$^{th}$ (Fig. 2). It should be noted that there is a remarkable coincidence

between the forecasted dust outbreak and its spatio-temporal evolution, and the satellite observations (cf., Figs. 1a and 2).

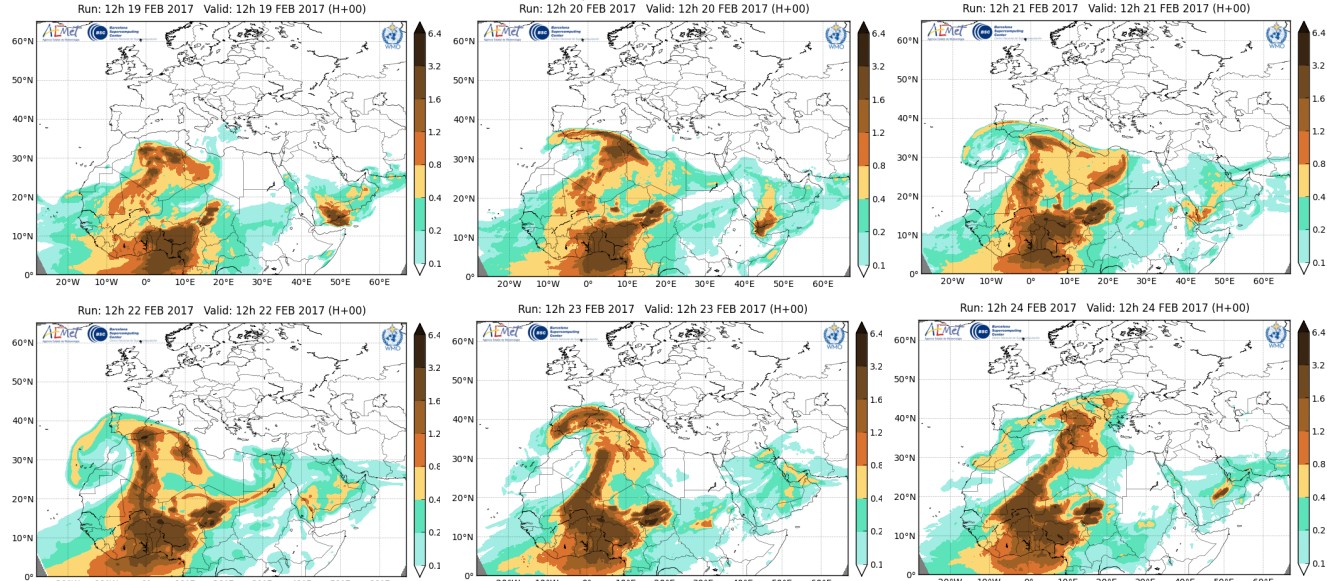

**Figure 2.** NMMB-BSC Dust forecast Aerosol Optical Depth (AOD) at 550 nm for the days before, during and after the extreme red rain event. Images from the Barcelona Dust Forecast Center.

Overall, the studied red rain event stands out as one of the most extreme of the last decades. Modeling and satellite imagery demonstrate that the event was associated with a massive desert dust outbreak affecting a large portion of the central- and north-western Sahara. Entrained Saharan dust was rapidly transported to the Iberian Peninsula where massive wet deposition of desert dust took place.

## 3.2 Synoptic situation and backward/forward trajectories

The synoptic scale meteorological situation during the extreme dust event was characterized by a marked NW Africa depression, nearly centered in the leeside of the Atlas Mountains (eastern Morocco-western Algeria). Such a depression was





bounded by two anticyclones, one centered in the North Atlantic (Azores area) and another centered in Libia (Fig. 3 and Fig. S1). Air masses were advected by strong winds with westward direction in the area of north-west Niger and north Mali that turned north upon entering the southern border of Algeria, heading straight north/north-east towards the south-east part of the Iberian peninsula (Fig. 3b and 3d). Advected air masses turned counter clockwise upon reaching the mid portion of the

Iberian peninsula and then moved west into the Atlantic ocean off the coast of Portugal, and then to the south, following the cyclonic wind field. Figure S1 shows the temporal (daily) and spatial evolution of the cyclone based on NOAA/ESRL reanalysis of geopotential height and wind field at 850 hPa. Remarkably, the center of the depression remained stationary over the same area (NW Africa) for nearly 4 days (February 19−22, 2017).

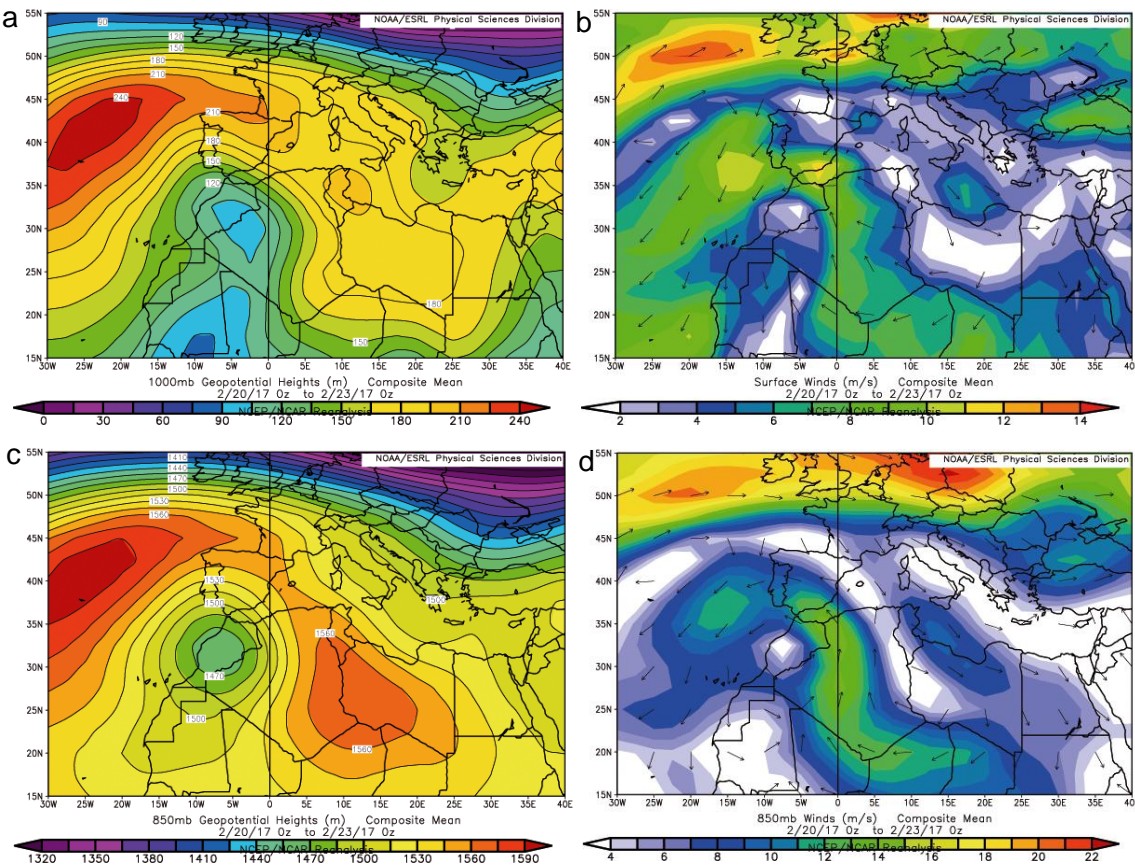

**Figure 3.** Synoptic scale meteorological situation during the red rain event based on NOAA/ESRL reanalysis. Averaged geopotential high (a and c) and wind field (b and d) for February 20−23, 2017 at both ground level (1000 hPa) and 850 hPa (~1500 m a.s.l.) are displayed.

Escudero et al. (2005) reported that major Saharan dust episodes affecting the Iberian peninsula are associated with: a) a north African high pressure system (located at surface or upper levels); b) an Atlantic depression, or c) a north African depression. The later synoptic scenario is associated with dust outbreaks which in 87% of the cases led to red rain. These red

rain events tend to occur during winter-spring and autumn, with dust typically coming from Algeria (mostly from northern areas, although dust in some events reportedly came from central Algeria) (Avila et al., 2007). Conversely, in summer, dry



deposition prevails, associated with a high thermal anticyclone system in north Africa (Rodriguez et al., 2001; Escudero et al., 2005). The amount of dust deposited per event is smaller in this later scenario. In our case, a strong north-west Saharan depression was responsible for the Saharan dust entrainment and transport to the Iberian peninsula. The synoptic scenario has strong similarities with a Sharav cyclone (Bou Karam et al., 2010). Cyclogenesis was triggered by an upper level (i.e.,

300 hPa) N−S trough west of the Iberian peninsula which favored the injection of strong cold north winds to the warmer north African troposphere (Fig. S2), thereby favoring the conditions for baroclinic instability. A very similar overall synoptic situation has been previously reported for other major red rain events taking place in the Spanish Mediterranean area (Sala et al., 1996), inland within the south and northeast part of the Iberian Peninsula (Avila et al., 1997, 2007; Rodriguez et al., 2001), and in the western-central Mediterranean (Fiol et al., 2005). Remarkably, two recent major winter dust events

affecting the western Mediterranean and associated with north African cyclones took place on February 20−23, 2007 (Bou Karam et al., 2010) and on February 20−25, 2016 (Titos et al., 2017). These observations suggest that there is a recurrent pattern in the spatio-temporal synoptic conditions (e.g., north African cyclones) leading to such extreme (winter) Saharan dust outbreaks and associated red rain events.

The backward and forward trajectory analyses showed that the air masses arriving at Granada during the studied

event entrained dust over a broad area spanning from central and south Algeria, the northern part of Mali and the northwestern part of Niger to the north of Algeria, south of Tunisia and western Libia (Fig. 4). This broad area includes two well-known Saharan dust potential source areas (PSA) as defined by Formenti et al. (2011) and Scheuvens et al. (2013): (i) PSA1, which covers the zone of chotts (ephemeral lakes) and dry lakebeds south of the Tell Atlas in northeast Algeria, south Tunisia and northwest Libia. This area is characterized by a relatively high illite, palygorskite and carbonate content; and (ii)

PSA3, which is one of the largest, most persistent and intense dust source areas in the Sahara (Goudie and Middleton, 2001; Prospero et al. 2002). It is located at the central-south section of Algeria, and spreads along the basins located south-west of the Ahaggar massif, the northern part of Mali and the frontier with Niger (e.g., Adrar des Iforas). Dust entrained from this area is characterized by relatively high illite, kaolinite and smectite contents, with minor amounts of palygorskite (Scheuvens et al., 2013). The importance of these two major source areas, particularly the second one, has been recognized by several

authors (e.g., D'Almeida, 1996; Goudie and Middleton, 2001; Prospero et al., 2002). However, previous studies did not recognize the contribution of such a distant area as PSA3 to extreme dust events affecting Western Europe. Avila et al (1997) reported that three main source areas were active during red rain events affecting the Iberian peninsula: (i) western Sahara (between 26°−30°N and 14°−8°W), (ii) Moroccan Atlas (between 30°−35°N and 8°−0°W), and (iii) central Algeria (between 26°−30°N and 4°W−5°E). Avila et al. (1997) report that the later source area is activated by a depression over the

Iberian peninsula or over north Africa. Such a source area is the one most closely matching the potential source areas identified here. However, our back- and forward-trajectory analysis shows that dust was mobilized from Saharan regions located much further south as well as from northern areas.





**Figure 4.** Results of NOAA's HYSPLIT modeling of: (a) 72 h multiple (25) back trajectories for air masses arriving at Granada on February 23[th], 2017 (00.00 UTC) and (b-d) forward multiple (25) trajectories for selected dust source areas. Shaded areas in (a) show potential source areas PSA1 and PSA3.

5        In summary, the synoptic scenario leading to the studied extreme red rain event appears to be recurrent, typically occurring in late winter or early spring, and being associated with a north African depression. Remarkably, this situation can mobilize and wet-deposit huge amounts of Saharan dust entrained from distant southern areas in the Sahara, about 3000 km away from the Iberian peninsula, as well as from closer areas in northern Africa.





### 3.3 XRD analysis: Linking mineralogy with potential dust source areas

The analysis of the bulk dust deposit (powder samples) showed the presence of (in order of decreasing abundance) (Table 1): clay minerals (see below details on individual clay minerals and their content), quartz ($SiO_2$), calcite ($CaCO_3$), plagioclase (albite, $NaAlSi_3O_4$), K-feldspar (microcline, $KAlSi_3O_4$), dolomite ($CaMg(CO_3)_2$), goethite ($\alpha$-FeOOH), hematite ($\alpha$-$Fe_2O_3$)

and rutile ($TiO_2$) (Fig. 5a). Note that phase amounts determined by XRD (RIR or Rietveld methods) are not considered to be purely quantitative, but rather semiquantitative due to the errors associated with this technique (typically ±5 wt%) (Formenti et al., 2011). Note also that amorphous phases (e.g., amorphous silica -in diatoms- and amorphous iron oxyhydroxides, see TEM results below) can not be detected using XRD, and their content was not negligible as shown by the broad hump at 18−32 °2θ in the XRD pattern (Fig. 5a). We observed that the Rietveld method did not allow the quantification of oxides,

plagioclase, and clays other than illite and kaolinite (Fig. S3). It also grossly overestimated the amounts of carbonate phases, i.e., 32±1 wt%, a value more than twice the amount determined using the RIR method (14±1 wt%), or by measuring the mass differences after acid-elimination of carbonates (13±1 wt%), or by TG analysis (16±1 wt%) (see below). The use of the Rietveld method for quantifying such complex multiphase dust samples is therefore not advised. In contrast, the RIR method yielded more reliable results, especially in the case of carbonate phases and clay minerals.

The detected mineral phases and contents are in general typical for Saharan dust deposited all across Europe, both during dry and wet deposition events (Scheuvens et al., 2013). Avila et al. (1997) reported a very similar mineral composition for the case of Saharan dust deposited after red rain events on the northeastern part of the Iberian peninsula. However, the authors did not report the presence of goethite, hematite and rutile. The total amount of clays measured here (~47 wt%) is smaller than the average value reported by Avila et al. (1997) (~64 wt%), or the values of 60−90 wt% reported

for Saharan dust collected in Africa or after long-range transport (Formenti et al., 2014a). This is likely due to the fact that very intense Saharan dust events, as the one studied here, typically mobilize larger particles with a relatively lower clay content of ~50 wt% (Caquineau et al., 2002; Formenti et al., 2014a).

**Table 1.** Semiquantitative XRD analysis of Saharan dust powder samples (wt%; ±1σ).

| Sample | Qtz | Kfd | Plg | Cal | Dol | Goe | Hem | Rut | Clay minerals |
|---|---|---|---|---|---|---|---|---|---|
| Bulk$_{RIR}$[†] | 23±1 | 6±1 | 7±1 | 12±2 | 2±1 | 1.7±0.8 | 0.5 ±0.4 | 0.8±0.3 | 47±3[*] |
| Bulk$_{Rietveld}$[‡] | 24±1 | 5±1 | - | 24±1 | 8±1 | - | - | - | 44±2 |
| Sand[†§] | 73±5 | 5±2 | 3±1 | - | - | 1±1 | - | - | 18±3[*] |
| Silt[†§] | 51±3 | 5±2 | 4±2 | - | - | 1±1 | 1±1 | - | 38±4[*] |
| Clay[†§] | 7±2 | - | - | - | - | 3±2 | 2±1 | - | 88±5[*] |

[*] Values determined considering the intensity of the general Bragg reflection of phyllosilicates at 4.49 Å; [†] Values obtained using the RIR method; [‡] Values obtained using the Rietveld method; [§] This size-fraction powder sample was subjected to carbonate elimination; Legend: Qtz: quartz; Kfd: microcline; Plg: plagioclase; Cal: calcite; Dol: dolomite; Goe: goethite; Hem: hematite; Rut: rutile



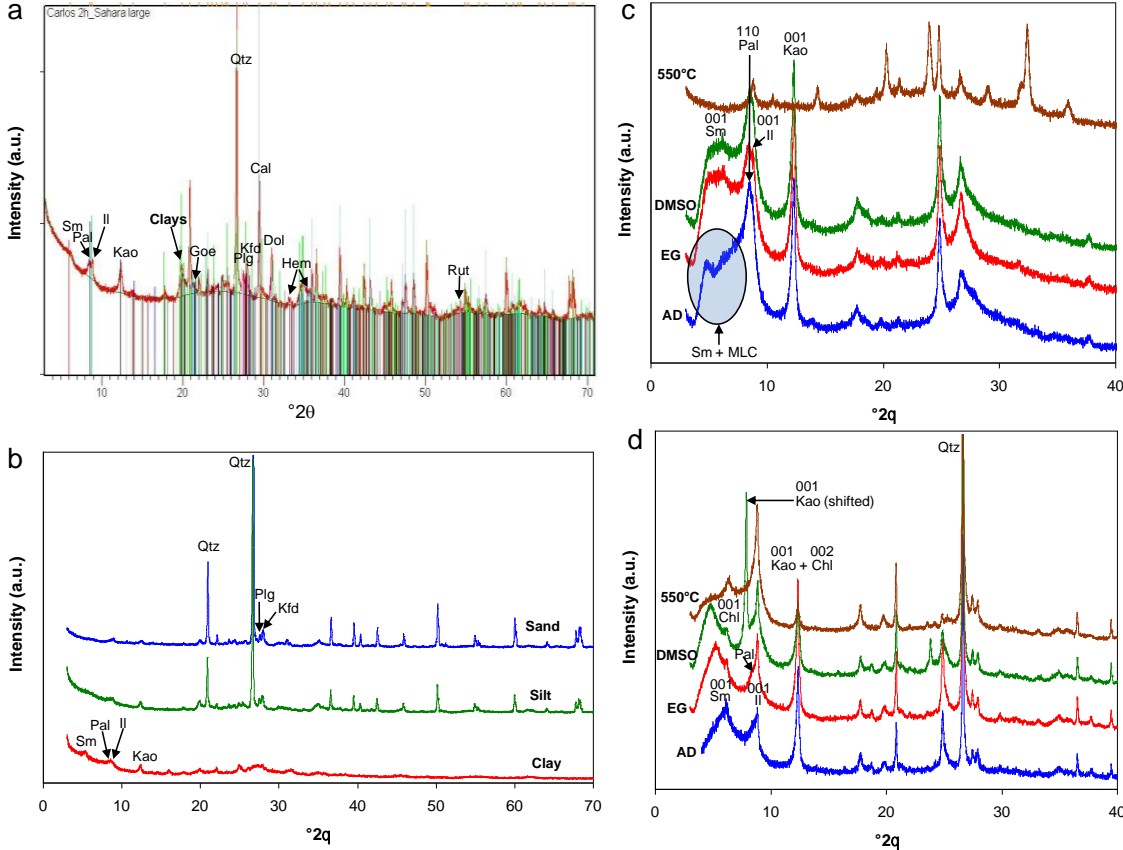

**Figure 5.** XRD analysis of Saharan dust: (a) XRD pattern of the bulk sample showing the presence of smectites (Sm), palygorskite (Pal), illite (Il), kaolinite (Kao), goethite (Goe), quartz (Qtz), K-feldspar (Kfd), plagioclase (Plg), calcite (Cal), dolomite (Dol), rutile (Rut) and hematite (Hem). The general reflection of clay minerals (Clays) at 4.49 Å is also indicated. Vertical color lines show the peak position of the different phases (only the main peaks corresponding to each phase are labeled). (b) XRD patterns of the sand, silt and clay fractions (decarbonated). (c) and (d) XRD patterns of oriented aggregates of the clay and silt fractions, respectively (with different treatments).

The XRD pattern of the sand fraction (after elimination of carbonates) showed intense reflections corresponding to quartz (main phase), with minor reflections corresponding to nearly all other identified phases (Fig. 5b and Table 1). It was unexpected to find clays and iron oxyhydroxides within this fraction, especially considering that acid dissolution of the carbonates and thorough washing were performed prior to size separation. Apparently, clays and iron oxyhydroxides are intimately associated (cemented) with the coarser quartz and feldspar grains (see SEM and TEM results below). This is a very important finding due to the effect that internal mixing (i.e., aggregates) of clays plus iron oxyhydroxides surrounding larger silicate mineral grains has on the radiative properties of desert dust (Sokolik and Toon, 1999; Formenti et al., 2014b; Kemppinen et al., 2015; Zhang et al., 2015; Jeong et al., 2016). Abundant quartz along with small amounts of feldspars, and abundant clay minerals were detected in the silt fraction (Fig. 5b and Table 1). Conversely, very small amounts of quartz were detected in the clay fraction (Fig. 5b), where clay minerals were the most abundant phases, along with iron oxyhydroxides.





Figure 5c shows the XRD patterns of oriented aggregates of the clay fraction. An intense Bragg peak at 10.4 Å which did not change position following EG and DMSO solvation, and collapsed to 10 Å upon heat treatment at 550 °C was observed. These are standard features of the 110 Bragg peak of palygorskite (($Mg,Al)_2Si_4O_{10}(OH)\cdot4(H_2O)$) (Moore and Reynolds, 1989). Note that in some publications this reflection has been mistakenly assigned to illite

($K(Al,Mg,Fe)_2(Si,Al)_4O_{10}(OH)_2$) (e.g., Prodi and Fea., 1979). In addition, we also observed the $d_{200}$-spacing of palygorskite at 6.4 Å, unambiguously confirming its presence. The intense peak at 7.17 Å, which did not shift position upon EG and DMSO solvation, and disappeared upon heat treatment corresponded to the 001 reflection of kaolinite ($Al_2Si_2O_5(OH)_4$). There was a poorly defined broad band at 12−18 Å that shifted position towards higher d-spacings upon EG and DMSO treatments, and collapsed to 10 Å upon heat treatment. These features are indicative of swelling clays (smectites), most

likely montmorillonite (($Na,Ca)_{0.3}(Al,Mg,Fe)_2Si_4O_{10}(OH)_2\cdot n(H_2O)$) (see TEM-AEM results below), and mixed-layer clays (MLC, interstratified illite-smectite; see TEM-AEM results below), accounting for the broad reflection centered at d-spacings ~14-17 Å and ~12 Å, respectively (AD sample). Note, however, that the broadness of the Bragg peaks precluded an unambiguous identification of any individual swelling clay. Finally, a shoulder located on the right of the 110 Bragg peak of palygorskite was observed at 10 Å, best seen in EG-solvated samples, pointing to the presence of illite (001 reflection),

confirmed by the appearance of its 002 and 003 order reflections at 5 Å and 3.33 Å, respectively.

Some mineralogical changes were detected in the silt fraction as compared to the clay fraction (Fig. 5d). The 001 reflection of illite was more intense than the 110 reflection of palygorskite, pointing to a relative increase in the abundance of illite in this coarser fraction. Kaolinite expanded upon DMSO treatment as shown by the shifting of the 001 peak at 7.14 Å to 11.18 Å. This is a standard feature of this clay mineral which enables its distinction from chlorite, which also has a

strong reflection at ~7.1 Å (González-García and Sánchez-Camazano, 1968). However, the formation of a DMSO-kaolinite intercalation complex is dependent on kaolinite crystallinity, as shown here by the fact that kaolinite in the clay fraction, which displayed a very broad 001 reflection (due to poor crystallinity), did not expand. These observations suggest size fractionation of kaolinite particles from two different sources, with the higher crystallinity coarser phase in the silt fraction likely coming from a closer-to-Europe source area (i.e., PSA1), and the less crystalline kaolinite in the clay fraction likely

coming from a more distant source (i.e., PSA3). Finally, it was observed that lattice expansion of swelling clays (smectite and MLC) upon EG and DMSO treatment unveiled the presence of a Bragg peak at 14 Å which upon heat treatment remained unaltered. The corresponding 2$^{nd}$ order 002 reflection at 7.1 Å was also visible after heat treatment. This behavior is characteristic of chlorite (($Mg,Fe^{++})_5Al(Si_3Al)O_{10}(OH)_8$), whose presence was masked by the broad 001 reflection of swelling clays at ~14 Å and the 001 reflection of kaolinite at 7.1 Å in the oriented AD mount or in the bulk powder sample.

The results of the semiquantitative XRD analysis of the clay minerals in the bulk fraction (powder samples) as well as in the clay and silt fractions (oriented mounts) are presented in Table 2. Apparently, the most abundant clay mineral in the bulk sample was illite. However, the analysis of oriented mounts showed that the most abundant clay was palygorskite, being illite the second most abundant clay mineral. This discrepancy can be explained considering that the RIR (or the Rietveld) method for (ideally non-oriented) illite fails to fully detract the contribution of the main reflection of (abundant)



quartz at 3.33 Å, which overlaps the 003 main reflection of illite. To this effect, one has to add the intensity increase related to preferential orientation of this phyllosilicate in the bulk powder sample. As a result, an overestimation of illite takes place. More reliable results are thus obtained using oriented mounts and in-house experimental RIR values referred to the 001 reflections (or 110, in the case of palygorskite) of the clay minerals. The amount of kaolinite was relatively high in the silt

and clay fractions, but still much lower than that of illite. The amount of chlorite, only present in the silt fraction, was low (~5 wt%), whereas the amount of smectites and MLC were relatively high. Note that MLC were only identified in the clay fraction. Finally, these results show that, contrary to common assumptions regarding the lack of abundant clay minerals in the silt fraction (Journet et al., 2014), these phases are very abundant in this coarser fraction.

**Table 2.** Semiquantitative analysis (RIR method) of clay minerals (wt%). Values in parenthesis show the wt% content of each type of clay mineral in the bulk sample.

| Sample | Ill | Pal | Sm | Kao | MLC | Chl |
|---|---|---|---|---|---|---|
| Bulk [powder] | 47±2 (22) | 24±3 (11) | 15±4 (7)[*] | 14±1 (7)[§] | - | - |
| Clay fraction [OA] | 29±1 (4.2) | 43±2 (6.3) | 8±1 (1.2) | 14±1 (2) | 7±2 (1) | - |
| Silt fraction [OA] | 32±1 (10.3) | 33±3 (10.7) | 11±1 (3.5) | 19±2 (6.2) | - | 6±2 (1.9) |
| Total Silt+Clay fractions | (14.5) | (17) | (4.7) | (8.2) | (1) | (1.9) |

[*] Includes the contribution of MLC and Chl; [§] Includes the contribution of Chl; Legend: Ill: illite; Pal: Palygorskite; Sm: smectite; Kao: kaolinite; MLC: mixed layer clay; Chl: chlorite

Most of the phases detected by XRD, with the exception of carbonates and some of the clays, cannot be used as a reliable indicator for dust source identification because they are common to all north African dust source areas (Scheuvens et al., 2013; Formenti et al., 2014a; Journet et al., 2014). Calcite and dolomite are recognized as good indicator of a northern and northwestern Saharan origin (Avila et al., 1997; Scheuvens et al., 2013), although southern Algeria and the Mali-Algeria border are areas that also contribute significant amounts of carbonates to Saharan dust (Scheuvens et al., 2013). Interestingly,

the TEM analysis (see below) showed the presence of abundant fibrous calcite, which typically forms in playa and/or ephemeral (and paleo) lakes in arid regions (Wanas, 2012). This points to PSA1 (chotts in north Algeria-Tunisia) as the most likely source area for this mineral.

The illite/kaolinite (I/K) ratio has been used to identify Saharan dust source regions (Caquineau et al., 2002). I/K >2 has been associated with northwestern Saharan source areas, I/K <0.5 with south Saharan and Sahelian source areas, and

intermediate values with central Saharan source areas. The I/K ratio in our dust samples ranged between 1.68 (silt fraction) and 2.07 (clay fraction) (we ignore here the I/K ratio of the bulk powder sample, because the illite content is overestimated for the reasons discussed above). Considering that the sand, silt and clay fractions represents 2, 68,  and 30 wt% of the whole sample (values determined weighting each fraction after hydrodynamic separation, see methods), the overall I/K ratio of the clay plus silt fractions is 1.77. This points to a central Saharan source area (i.e., south-central Algeria). However, and in




agreement with satellite imagery, BSA dust forecast, and backward/forward-trajectory analyses, the existence of two types of kaolinite with low (no expansion with DMSO) and high crystallinity (expansion with DMSO) suggests the possibility of mixing of dust entrained both from south (low I/K ratio) and north (high I/K ratio) Saharan areas (i.e., PSA3 and PSA1). Indeed, the relatively high carbonate content and abundant palygorskite point to an additional north(western) Saharan source.

Palygorskite has been recognized as a reliable indicator for a Saharan provenance (Coudé-Gaussen, 1991; Scheuvens et al., 2013), and has been detected in red rain events not only in the Iberian peninsula (Avila et al., 1997), but also in western (Fiol et al., 2005), central (Molinaroli, 1996) and eastern Mediterranean areas (Ganor et al., 2009), the Alps (De Angelis and Gaudichet, 1991), the Netherlands (Reiff et al., 1986) and the British Islands (Bain and Tait, 1977). Palygorskite is a common clay in soils of northwestern Sahara (PSA2), as well as north Algeria and Tunisia (PSA1) (Scheuvens et al., 2013).

Nonetheless, occurrences of palygorskite, which typically forms in saline lakes and alluvial sediments of arid regions (Singer and Galan, 1984), has also been documented in central and southern Algeria, as well as in northern Mali (PSA3) (Coudé-Gaussen, 1991; Scheuvens et al., 2013). Remarkably, the amount of palygorskite in our studied dust samples is exceptionally high (17 wt% of the bulk sample; see Table 2). Previous studies of Saharan dust deposited in the central Mediterranean (Mallorca and Sardinia) reported palygorskite values ≤5 wt % (Molinaroli, 1996, Fiol et al., 2005), although Avila et al.

(1997) observed a concentration of up to 12 wt% in red rains in NW Spain. It is likely that in our case, the close proximity to northern Africa and the intensity of the dust event favored the entrainment of palygorskite-rich dust from northern Algeria and Tunisia areas, and its transport with minimal segregation of the silt-sized palygorskite particles prior to their deposition in southern Spain. Alternatively, it might be argued that in previous studies the amount of palygorskite was underestimated due to the difficulty of quantifying this mineral using XRD. Such a problem is rooted in the general lack of reliable and accurate RIR values for palygorskite: this is why we experimentally determined the RIR value of palygorskite. The same

applies for quantitative Rielveld analysis of palygorskite due to the difficulty of obtaining accurate structural factors for clay minerals in general, and for this clay mineral in particular. We will show below that an appropriate characterization and quantification of this mineral in Saharan dust is of great significance due to its potential health effects. Finally, it is worth to comment on the chlorite to kaolinite ratio (Chl/K), which has also been proposed for discriminating Saharan dust source regions (Scheuvens et al., 2013). Values <0.3 have been reported for south Algeria and north Mali, while higher values have

been observed in soils from the north and northwestern Sahara (see compilation by Scheuvens et al., 2013). We obtained Chl/K ~0.2, a value which is consistent with a south Algeria and/or north Mali source region.

Overall, and in agreement with previous studies that pointed out that major Saharan dust outbreaks affecting large desert areas typically involve mixing of dust entrained from different active dust source areas (Skonieczny et al., 2011;

Formenti et al., 2014a), our mineralogical analysis, in combination with satellite imagery, BSC dust forecast, and forward- and backward-trajectory analyses, shows that this is the case here too, with PSA1 and PSA3 as the most probable dust source areas. Despite the difficulty in accurately pin-pointing the different source areas down to a local scale during such an extreme event, our results show that it is still possible to identify different regional dust signatures that help constraint potential dust source areas.




### 3.4. Chemical composition

Table 3 shows the results of XRF and ICP analysis of the dust samples. Both techniques yielded consistent and comparable results. Si and Al were the two most abundant elements, which indicates that quartz and aluminosilicates (clays and feldspars) are the main mineral phases in Saharan dust (Goudie and Middleton, 2001). The Si/Al ratio of 3.68 (note: all ratios

calculated using the average values of XRF and ICP results) falls within the range of values reported for Saharan dust source areas and Saharan dust deposited in western and central Mediterranean areas (Scheuvens et al., 2013). It is in good agreement with the value (3.92) reported for soils in southern Algeria (Guieu and Thomas, 1996). The later is consistent with the PSA3 source area, as discussed above. This value is, however, larger than the Si/Al values (1.6−3.0) reported by Formenti et al. (2014b) for Saharan dust and soils. Likely, the higher Si content in our samples is due to the favored

entrainment and transport of abundant coarse $SiO_2$ particles during such an extreme event. The relatively high Ca, and to a lesser extent Mg, content shows that carbonates (calcite and dolomite) are abundant. Note, however, that such elements are also present as octahedral or interlayer cations in clay minerals, specially smectites and MLC (Mg and Ca), palygorskite (Mg), and chlorite (Mg) (see AEM results below). The Ca/Al and Mg/Al ratios of 0.21 and 0.92, respectively, agree with those reported for Saharan dust from Algerian source areas deposited in southern Europe (Avila et al., 2007; Scheuvens et

al., 2013). Fe is the fourth most abundant element with an average concentration of 3.56±0.18 wt%, being present both in phyllosilicates (see TEM-AEM results below) and in iron oxyhydroxides. This value falls within the range ~2−11 wt% reported for Saharan dust and soils (Zhang et al., 2015), but is slightly lower than the average value of 4.45 wt% Fe proposed as characteristic of Saharan dust (Guieu et al., 2002), or the range of values (4.3-6.1 wt% Fe) reported by Lafon et al. (2006) for dust and soil samples from the Sahel, south Morocco, central Sahara, and Tunisia. Nonetheless, previous studies have

reported that the total iron content of Saharan dust deposited in Europe ranges between 3.5 and 5.6 wt% Fe (Goudie and Middleton, 2001). Moreover, the Fe/Al ratio of 0.54 in our samples falls within the range of values (0.50−0.57) reported for red rains in NE Spain (Avila et al. 1998; 2007). Similarly, the (Ca+Mg)/Fe ratio is 2.06, a value that match that reported by Avila et al. (2007) for Saharan dust from central Algeria deposited in the north-eastern Iberian peninsula during red rain events. The contents of K and Na are consistent with the presence of K-feldspar, plagioclase and clay minerals (i.e.,

abundant K-containing clays such as illite and MLC). The amount of Ti is in good agreement with the amount of rutile detected using XRD (~1 wt% $TiO_2$). Regarding minor/trace elements, the amounts of Ba, Mn, Zn, and Sr, associated with carbonates, as well as Ni, Cr, P and V, associated with silicates, are in good agreement with those reported for dust in red rains coming from northern Saharan sources (Avila et al., 2007). Interestingly, the Pb content (27 ppm) is very low and nearly identical to the average values (24 ppm) in Saharan soils with negligible anthropogenic perturbation (Guieu et al.,

2002). Typically, mixing of Saharan dust plumes with polluted air masses (i.e.., with anthropogenic perturbations) leads to a significant increase in the Pb/Al ratio in dust wet-deposited in the central and western Mediterranean (Guieu et al., 2002). However, in our case the Pb/Al ratio of $3.78 \times 10^{-4}$ is very similar to the Pb/Al of $3.41 \times 10^{-4}$ reported by Guieu et al. (2002) for Saharan endmember. Similarly, the amount of S, an element commonly associated with anthropogenic pollution, is very



low. This value (0.04 wt% S) is almost two orders of magnitude lower than those reported by Avila et al. (2007) for red rains in the NE Iberian peninsula, and is identical to that of western Saharan desert soil (Castillo et al., 2008). These results demonstrate that the close proximity of the southern Iberian Peninsula to north Africa and the rapid, intense dust advection during this Saharan dust event led to negligible contamination from polluted (e.g., European) sources (Lyamani et al., 2005).

**Table 3.** Chemical composition of Saharan dust deposited in Granada during an extreme red rain event.

| | Major/minor elements (wt%) | | | | | | | | |
|---|---|---|---|---|---|---|---|---|---|
| | Si | Al | Fe | Mg | Ca | Na | K | Ti | P |
| XRF | 24.05 | 6.62 | 3.69 | 1.24 | 5.72 | 1.66 | 1.53 | 0.45 | 0.04 |
| ICP | nd | 6.44 | 3.43 | 1.48 | 6.29 | 1.62 | 1.57 | nd | 0.06 |
| | Minor/trace elements (ppm) | | | | | | | | |
| | Ba | Sr | Mn | Zn | V | Cr | Co | Ni | Pb | S |
| XRF | nd | 162 | 400 | nd | nd | 78 | nd | 51 | nd | nd |
| ICP | 529 | 231 | 741 | 367 | 103 | 84 | 16 | 44 | 27 | 390 |

nd: Not determined

In summary, the content of major, minor and trace elements in our studied samples is fully consistent with the reported composition of Saharan dust samples (Scheuvens et al., 2013), and closely matches the composition of dust

deposited during red rain events in the western Mediterranean, especially those from an "eastern" provenance, as defined by Avila et al. (2007): i.e., those whose source areas are located in the northern and central parts of the Sahara (Algeria, Libia and Tunisia). These compositional results are thus consistent with the results of previous sections pointing to PSA1 and PSA3 as the source regions for the dust deposited in Granada during the studied event.

### 3.5 TEM-AEM analyses

TEM identification of individual mineral particles was performed combining information provided by morphology, AEM microanalysis, and selected area electron diffraction (SAED). Figure 6 shows representative TEM images of dust particles and aggregates. Abundant fibrous particles were observed, either with size >1−5 μm, made up of Si, Mg, and Al (EDX results) with a SAED pattern matching that of palygorskite (Fig. 6a, 6b, 6e and 6f), or with size ~0.5−1 μm, identified as calcite by their high Ca content and SAED pattern (Fig. 6a and 6c). Abundant non-fibrous clay mineral particles were also

identified, which based on their composition (EDX results) and SAED pattern were: kaolinite (Fig. 6a, 6d and 6j), illite (Fig. 6a and 6e), smectites (Fig. 6a and 6g-h), and chlorite (Fig. 6h). Goethite particles up to ~250 nm in size were also identified by EDX and SAED (Fig. 6f) along with scattered iron-rich nanoparticles <100 nm in size (identified by EDX microanalysis), typically associated with clay minerals (Fig. 6h). These nanoparticles did not produce any diffraction spots in the SAED pattern: only the  diffraction spots of the underlying clay minerals were observed along with diffuse Debye rings at 4.5 Å,

corresponding to the general *hk*0 reflection of (poorly crystalline) clay minerals, and at ~2.5 Å and at ~1.5 Å which can be





ascribed to poorly crystalline 2-line ferrihydrite ($Fe_4HO_8·4H_2O$) (Jambor and Dutrizac, 1998), with contributions of higher order $hk0$ reflections of the clay minerals (Fig. 6h, inset). Amorphous and/or poorly crystalline iron-rich nanoparticles, possibly ferrihydrite, in desert dust have been previously reported (Shi et al., 2009; 2011; 2012), typically forming coatings on clay minerals (Wagner et al., 2012) or appearing dispersed in the clay matrices of clay-rich particles (Jeong et al., 2016).

Feldspars (plagioclase) were also identified (Fig. 6a). Interestingly, we observed silica particles with a complex structure (septa) which were amorphous to the electron beam (i.e., their SAED showed no diffraction spots or Debye rings, but diffuse haloes characteristic of an amorphous phase) (Fig. 6i). These are standard features of mineralized skeletons of diatoms, which are abundant in dust from the Bodélé (Formenti et al., 2011), but also occur in fluvial deposits and ephemeral lakes, as well as paleolake basins all across the Sahara (Shi et al., 2011), especially in central-south Algeria and Tunisia, and are

found in Saharan dust deposited in marine sediments and continental Europe (Gasse et al., 1989). Large quartz grains, typically covered by phyllosilicates and iron oxyhydroxides rims, were also identified (Fig. 6g). The presence of clay- and iron-rich coatings on large quartz and/or feldspars, as well as on carbonate grains, appears to be a general feature of Saharan dust particles (Jeong et al., 2016). Finally, calcite rhombohedra (Fig. 6j), as well as very small (~100 nm) rutile and scarce ilmenite ($FeTiO_3$) crystals (identified by both SAED and EDX microanalysis), were also observed.

15        Table 4 shows average values for the structural formula of the different clay minerals identified by TEM-AEM (see Table S1 for a complete list of all individual AEM analyses). Microanalysis results confirmed the presence of illite, palygorskite, kaolinite, smectite, and MLC. Due to its scarcity and mixing with other clay phases, we could not collect any clean and reliable AEM analysis for chlorite. Illite had a composition typical for this clay mineral, with a substantial phengitic component as shown by the relatively high Fe and Mg content in the octahedral layer (Weaver and Pollard, 1973).

Most palygorskite analyses (Table S1) had excess Si and relatively low Al and Mg in octahedral sites. This is due to beam damage during AEM analysis of this beam-sensitive mineral. Nonetheless, the average structural formula of palygorskite obtained here is standard for this fibrous clay, with a slightly elevated Fe content, not unusual for this fibrous clay (Weaver and Pollard, 1973). The AEM analysis of kaolinite revealed a slight Al deficit and very small amounts of Fe, an element that substitutes Al in the octahedral layer, especially in the case of poorly crystalline kaolinite (Mestdagh et al., 1980). In the case

of analysis #7 (Table S1), however, a possible contamination with Fe-rich nanoparticles deposited on the kaolinite surface may account for the anomalously high Fe content (0.88 mol Fe per formula unit). Note that analysis #7 was not considered for the calculation of the average structural formula of kaolinite reported in Table 4. We identified a smectitic phase with a composition compatible with montmorillonite. The relatively high Al content in tetrahedral positions, resulting in a reduced amount of Si, and the abundant Al and Fe in octahedral positions, point to beidellitic and nontronitic contributions (i.e., solid

solution between the extreme terms montmorillonite and beidellite, with a minor nontronitic component) (Weaver and Pollard, 1973). In addition, we identified a MLC with a relatively high K content in the interlayer, which is consistent with an illite-smectite mixed layer phase, also with a relatively high Fe content. MLC can be distinguished from smectite by its lower Si content and corresponding higher Al content in tetrahedral positions. It is also distinguished from illite by the reduced K content and interlayer charge.





**Figure 6.** TEM photomicrographs of Saharan dust particles. (a) General low-magnification overview of the particles. Note the abundance of fibrous particles. The long ones are palygorskite (Pal), while the short ones (~0.5 µm) are calcite (Cal). The larger particles or aggregates corresponds to feldspars (plagioclase, Plg), illite (Il) and smectites (Sm). The smaller particles include kaolinite (Kao) and rutile (Rut). (b) Detail of elongated palygorskite fiber. The [001] zone axis SAED pattern is shown in inset. (c) Aggregate of fibrous calcite (SAED pattern in inset). (d) Plate-like kaolinite particles (SAED pattern in inset). (e) Illite (SAED in inset), rutile, palygorskite and fibrous calcite. (f) Goethite (Goe) (SAED pattern in inset) and palygorskite. (g) Quartz (Qtz) grains (SAED in inset) surrounded by smectites (Sm). The red circled area shows clay minerals plus iron oxyhydroxide nanoparticles forming a rim around the quartz grain. (h) Chlorite (Chl) (SAED in inset) with adhered smectite and iron oxyhydroxide nanoparticles. The blue and the red arrows in the SAED pattern show the diffuse Debye rings corresponding to the reflections with d-spacing 4.49 Å and 2.5 Å, respectively (see discussion in the main text). (i) Amorphous silica structure (diffuse haloes in SAED pattern in inset). This is part of the skeleton of a diatome. (j) Rhombohedral calcite crystal (SAED in inset) surrounded by kaolinite.

All analyzed clay minerals contained significant amounts of Fe, an important finding regarding its biogeochemical (e.g., iron bioavailability) and radiative implications. It could be argued, however, that the intimate mixing/attachment of Fe-



rich nanoparticles with/on clay minerals might yield an excess of Fe during AEM analyses. However, with a few exceptions (e.g., the above-reported high iron-content of one of the kaolinite particles), this is very unlikely: special care was taken to collect AEM analysis only from areas of the clay mineral particles free of Fe-rich nanoparticles.

The average Fe content in the clay minerals varied between 1.7 wt% in kaolinite and 5.74 wt% in smectite, with intermediate values of 3.4, 3.6 and 4.8 wt % in MLC, palygorskite and illite, respectively. These values are consistent with those reported by Jeong and Achterberg (2014) and Jeong et al. (2016) for clay minerals in Saharan dust analyzed using TEM-AEM. Overall, the average Fe content for all clay minerals analyzed here is 3.84 wt%. If one considers that there is also chlorite, which has a typical Fe content of 14.8 wt % in desert dust (Jeong and Achterberg, 2014), then the average Fe content in our studied clays rises to 5.7 wt %. This later value is in very good agreement with the average 5.4 wt% Fe content of clay minerals in Saharan dust analyzed by Jeong and Achterberg (2014).

**Table 4.** Structural formulae of clay minerals in Saharan dust from TEM-AEM analysis

| Si | Al$^{IV}$ | Al$^{VI}$ | Mg | Fe | Sum Oct.[1] | K | Ca | Na | Sum Int.[2] |
|---|---|---|---|---|---|---|---|---|---|
| \multicolumn{10}{c}{Illite based on $O_{10}(OH)_2$} | | | | | | | | | |
| 3.46±0.10 | 0.54±0.10 | 1.49±0.28 | 0.20±0.01 | 0.37±0.33 | 2.06±0.04 | 0.46±0.17 | 0.06±0.08 | 0.06±0.08 | 0.63±0.06 |
| \multicolumn{10}{c}{Palygorskite based on $O_{10}OH$} | | | | | | | | | |
| 4.12±0.19 | | 0.71±0.20 | 0.83±0.37 | 0.26±0.17 | 1.79±0.26 | 0.05±0.04 | 0.03±0.02 | | |
| \multicolumn{10}{c}{Kaolinite based on $O_5(OH)_4$} | | | | | | | | | |
| 2.01±0.02 | | 1.91±0.08 | 0.01±0.03 | 0.06±0.05 | | | 0.01±0.01 | | |
| \multicolumn{10}{c}{Smectite based on $O_{10}(OH)_2$} | | | | | | | | | |
| 3.65±0.17 | 0.35±0.17 | 1.41±0.29 | 0.24±0.11 | 0.41±0.25 | 2.06±0.08 | 0.18±0.08 | 0.11±0.06 | | 0.40±0.13 |
| \multicolumn{10}{c}{Illite-smectite mixed layer based on $O_{10}(OH)_2$} | | | | | | | | | |
| 3.43±0.19 | 0.57±0.19 | 1.50±0.43 | 0.47±0.37 | 0.32±0.14 | 2.28±0.08 | 0.12±0.02 | 0.02±0.03 | | 0.17±0.05 |

[1] Sum of octahedral cations; [2] Sum of interlayer charge ($M^+ + M^{2+}$)

Further detailed textural and compositional insights were obtained using the HAADF and EDX detectors of the Titan TEM operated in STEM mode. Figures 7 and 8 (as well as Figs. 4S and 5S) show dark field HAADF images (Z-contrast) and corresponding elemental maps of representative aggregates and individual particles. Most abundant particles appeared as micrometer-sized aggregates made up of internally mixed silicate (quartz and amorphous silica), aluminosilicate (feldspars), carbonate (calcite) and abundant phyllosilicate particles (mainly kaolinite, illite, palygorskite, smectite and MLC), interspersed or covered with abundant iron-rich (nano)particles as well as scarce titanium oxide (nano)particles. The internal mixing of such phases, in particular iron oxyhydroxides/clay minerals, has been previously observed using standard TEM imaging (Jeong and Achterberg, 2014; Jeong et al., 2016) as well as energy-loss TEM tomography (Deboudt et al., 2012). Most of the iron-rich particles embedded and dispersed within the micrometer-sized clay-rich aggregates, as well as



those covering or attached to individual clay particles, were <100 nm in size (Fig. 8). The tendency of iron oxyhydroxides to concentrate in the smallest (<100−200 nm) size-fraction, typically forming nanogranular coatings or "nanoclusters" attached to clay minerals, appears to be a general feature of Saharan dust (Lieke et al., 2011; Wagner et al., 2012; Zhang et al., 2015). Such iron oxyhydroxide nanoparticles were texturally different from the much larger individual goethite crystals (compare

5    Fig. 6f with Fig. 8). These observations further suggest that these iron-rich nanoparticles are amorphous or poorly-crystalline ferrihydrite (Shi et al., 2009). However, it was unclear whether or not such nanoparticles resulted from atmospheric processing of crystalline iron-rich phases such as goethite and hematite (Shi et al., 2009; 2012) and/or clay minerals (Journet et al., 2008), or were primarily formed by in situ weathering at the source regions (i.e., prior to entrainment) (Shi et al., 2011). We will discuss this point below.

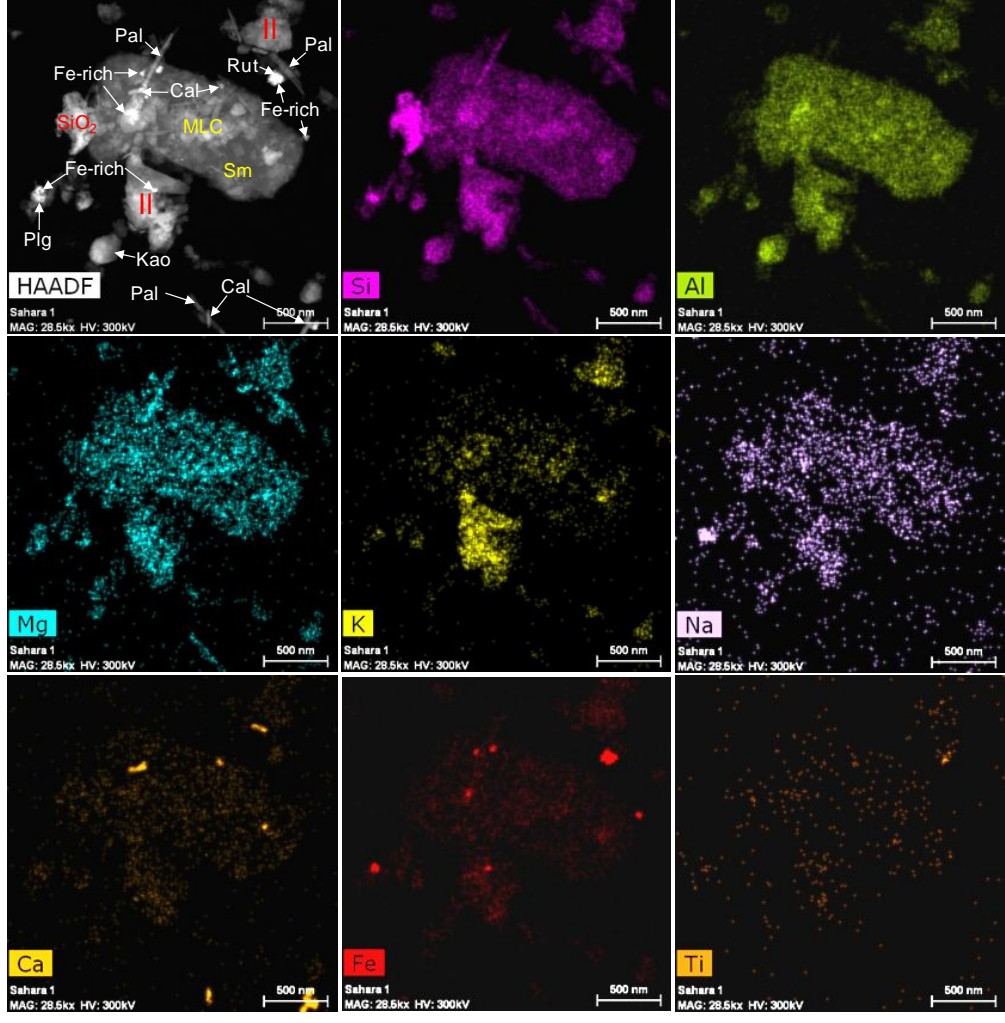

**Figure 7.** STEM-HAADF photomicrographs and corresponding EDX elemental maps of Saharan dust particles forming a micrometer-sized aggregate. Based on the compositional analysis, particles of palygorskite (Pal), calcite (Cal), iron oxyhydroxides (Fe-rich), rutile (Rut), illite (Il), smectite (Sm), kaolinite (Kao), silica ($SiO_2$), which according to SAED results is amorphous (i.e., diatome), plagioclase (Plg) and mixed layer clays (MLC) were identified.



Interestingly, elemental maps of dust aggregates also revealed that Ca-rich particles corresponding to calcite were commonly associated with the abundant Fe-rich nanoparticles and clay minerals, forming internally mixed aggregates (Figs. 7, 8, S4 and S5). This contradicts the common assumption regarding calcite mixing state in Saharan dust, which is considered to be external (Shi et al., 2012). Moreover, no nitrogen enrichment (i.e., formation of calcium nitrates) (Krueger et al., 2004) or sulfur-containing phases (such as gypsum) resulting from the reaction of acid pollutant gases with carbonates were detected (Figs. 7, 8, S4 and S5). This is in agreement with the above reported very low Pb concentration, showing negligible mixing with air masses including anthropogenic-derived pollutants. These results have important implications regarding the possible atmospheric acid processing of the iron-containing phases, as we will discuss below.

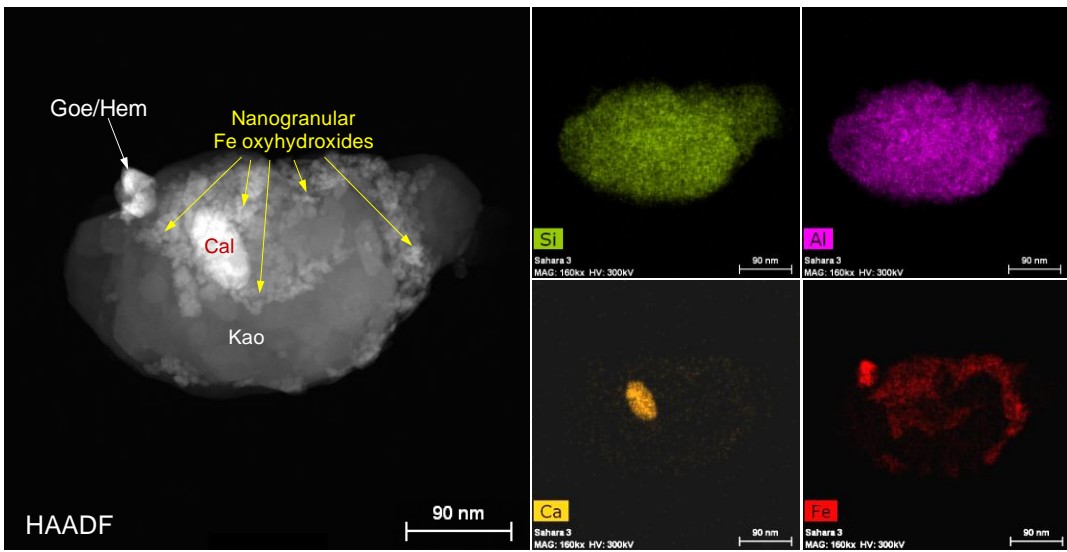

**Figure 8.** STEM/HAADF photomicrograph and corresponding EDX elemental maps of a kaolinite (Kao) particle internally mixed with a nearly rhombohedral-shaped goethite or hematite (Goe/Hm) crystal, a calcite (Cal) particle, and aggregates of Fe-rich nanoparticles (nanogranular Fe oxyhydroxides), likely ferrihydrite.

### 3.6 Textural features of Saharan dust: FESEM, PSD and N$_2$ sorption analyses.

Figure 9a-d shows representative FESEM photomicrographs of dust mineral particles. Abundant particles ~3 to ~30 μm in size (silt fraction) were observed. Interestingly, the larger particles (Fig. 9a) were typically ellipsoidal in shape with angular, sharp edges, all common features of coarse Saharan dust particles (Wagner et al., 2012; Jeong et al., 2016). They were made up of an aggregate of smaller, well-cemented (i.e., the aggregates were intact after sonication), micrometer-sized particles (Fig. 9b). The later, in turn, displayed very rough surfaces made up of nanoparticle aggregates (Fig. 9c and 9d). EDX microanalyses showed that with the exception of some large quartz grains (showing O and Si with trace amounts of Al and Fe in the EDX spectrum), or carbonates (showing O plus Ca -calcite-, or Ca and Mg -dolomite-, with trace amounts of Al, Mg, Si, and Fe in the EDX spectrum), the majority of the particles did not display a clean EDX spectrum (see inset in Fig. 9b), but included significant amounts of Ca, Mg, Al, Si, K, Na, and Fe (and Ti), likely due to the contributions of clay minerals and iron oxyhydroxides (and rutile) nanoparticles that covered nearly all dust particles and aggregates (i.e., these



phases were internally mixed). Surface coatings of clays and oxyhydroxides (as well as other amorphous phases) reportedly mask the identity of underlying mineral grains (Engelbrecht et al., 2016). These results suggest that great care should be taken when using SEM-EDX for the chemical analysis and/or mineralogical identification of individual desert dust particles.

Figure 9e shows the volume particle size distribution (PSD) of the Saharan dust determined by laser scattering. A polymodal PSD was observed, with a principal mode at 17 µm, a secondary mode at 0.2 µm, and additional less marked modes at 2 µm and 0.7 µm. A very similar polymodal PSD has been previously reported for Saharan dust transported during major outbreaks (D'Almeida and Schütz, 1983; Menéndez et al., 2014; Titos et al., 2017), including those involving red rains (Sala et al., 1996; Guieu et al., 2002). The samples studied had 24, 75, and 1 vol% clay, silt and sand content, respectively. In general, these values were in agreement with the wt% content of the different size fractions determined using hydrodynamic size-separation. However, some differences were observed, especially in the case of the clay fraction (i.e., 24 vol% vs. 30 wt%). The later can be explained by the fact that iron oxyhydroxides, with a density, $\rho$ nearly double than that of clay minerals (i.e., $\rho$ of goethite, 4.3 g cm$^{-3}$ and hematite, 5.3 g cm$^{-3}$ vs. $\rho$ of clay minerals, ~2.6−2.8 g cm$^{-3}$), were preferentially concentrated in the clay fraction (see XRD and TEM results, above).

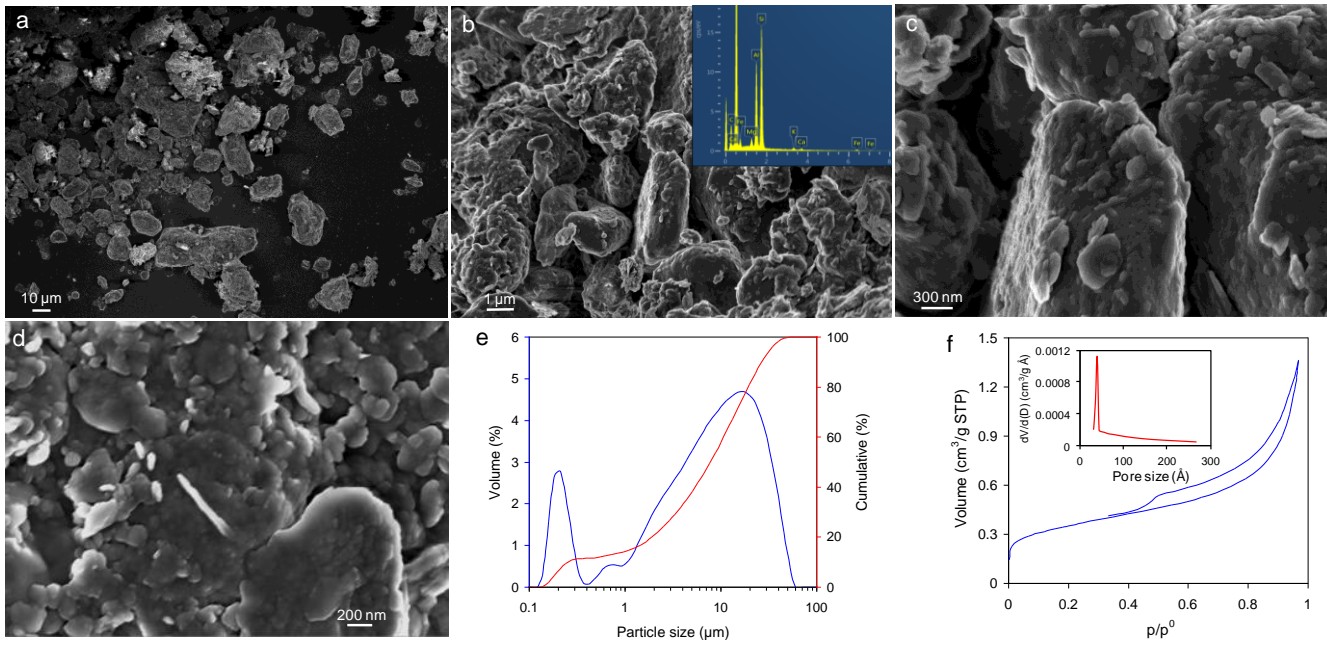

**Figure 9.** FESEM photomicrographs of Saharan dust particles. Large particles (a) are made up of smaller, micrometer-sized particles (b) which in turn are covered by nanoparticles (c). (d) Detail of the nanogranular surface texture of dust particles. The inset in (b) shows a representative EDX spectrum (corresponding to the particle in the center of the image) showing a high content of Si, Al and Mg, with small amounts of K, Ca and Fe. Very similar spectra are recorder for most of the areas in the sample demonstrating that the larger particles (quartz, carbonates and minor amounts of feldspars) are covered by clays and iron oxyhydroxide (nano)particles. (e) Particle size distribution (vol% PSD and cumulative curves) and (f) nitrogen sorption isotherm of dust particles. The inset shows the BJH pore size distribution.



Figure 9f shows the $N_2$ sorption isotherm of Saharan dust. It was of type IV, typical for mesoporous materials such as clay minerals, and showed a type H3 hysteresis loop (Sing et al.,1985). The later is due to the presence of slit-shaped pores associated with aggregates of plate-like (nano)particles, such as the clay minerals present in the Saharan dust. The average surface area was $25 \pm 1$ m$^2$ g$^{-1}$. The main contributors to such a relatively high surface area are the smallest particles, that is, clays (specially smectites and MLC) and iron oxyhydroxides (Elert et al., 2015). $N_2$ sorption measurements yielded a pore volume of $0.039 \pm 0.002$ cm$^3$ g$^{-1}$. The pore size distribution determined using the BJH method (inset in Fig. 9f) was unimodal with a maximum at 5 nm. The presence of relatively abundant (nano)pores, an aspect of desert dust that has been typically neglected, is of relevance due to their effect on the dust's reactivity, hygroscopicity, and radiative properties (Kemppinnen et al., 2015).

Combined, FESEM, PSD and $N_2$ sorption analyses show that although clay-sized particles constitute a significant fraction of the Saharan dust, and strongly contribute to its overall porosity and surface area, coarser particles are particularly abundant. For instance, 42 vol% of the dust particles were >10 μm in size. Although abundant coarse and even giant particles have been reported for major Saharan dust events involving dust entrainment and transport under strong wind conditions (Jeanicke and Schütz, 1978; D'Almeida and Schütz, 1983; Coude-Gaussen et al., 1987; Goudie and Middleton, 2001; Otto et al., 2007; Weinzierl et al., 2009; Menéndez et al., 2014), most analyses of Saharan dust typically report values <10 μm (Reid et al., 2003; Lyamani et al., 2005; Mahowald et al., 2014). Note, however, that PSD analyses of Saharan dust are commonly performed using cascade impactors or optical inversion techniques (e.g., AEronet RObotic NETwork, AERONET), which either typically exclude particles with size >10 μm or underestimate their vol% (Raiswell 2011). Conversely, in our case, a possible bias towards larger particle sizes might occur due to the use of a laser scattering system for PSD analysis (Reid et al., 2003). However, this effect appears to be minor as demonstrated our SEM observations showing the presence of abundant particles larger than 10 μm (Fig. 9a). It could be argued that the larger particles observed here may result from aggregation phenomena taking place after in-cloud and/or below-cloud scavenging during the red rain event (Mahowald et al., 2014). Coarsening due to aggregation of Saharan dust particles scavenged within cloud droplets followed by drying prior to dry deposition has been claimed responsible for the formation of the so-called "iberulites" (Cuadros et al. 2015; Diaz-Hernandez and Sanchez-Navas, 2016). Iberulites are nearly spherical, clay-rich giant particles (~100 μm in diameter) that are dry deposited in Spain typically during summer Saharan dust events. However, we observed no iberulite-like aggregates in our wet deposited samples. According to Fiol et al. (2005) such giant spherical structures rapidly disaggregate upon contact with water. The former observation and the facts that (i) the studied coarse and giant aggregates show sharp edges (Fig. 9a) resulting from saltation-sandblasting at entrainment areas (Coude-Gaussen et al., 1987; Mahowald et al., 2014), and (ii) Saharan dust analyzed during major outbreaks in different areas, both at ground stations or aloft, including airborne measurements such as those of SAMUN and FENNEC aircraft campaigns, typically show similar PSD with a significant contribution of coarse and even giant particles (Weinzierl et al., 2009; Ryder et al., 2013; Titos et al., 2017), suggest that irreversible aggregation (i.e., coarsening) due to cloud processing during the red rain event is not a significant process affecting the PSD of the Saharan dust studied here.



The PSD of desert dust aerosol plays a critical role in its direct radiative forcing (Tegen and Lacis, 1996; Reid et al., 2003; Mahowald et al., 2014). In this respect, it is also very important to consider that most of the individual particles in desert dust are not single-phase, but a complex aggregate of internally mixed phases as shown by our FESEM and TEM observations. A correct evaluation of the PSD and mixing state of Saharan dust particles can have profound effects on the

radiative properties of desert dust, as we will discuss below.

### 3.7 TG-DSC and spectroscopic analyses

TG-DSC analyses showed a first weight loss between 100 and 600 °C corresponding to the dehydration and dehydroxylation of clay minerals (Fig. 10). A minor contribution to such a weight loss was due to the dehydroxylation of ferrihydrite and goethite, which reportedly occurs at ~150 °C and ~280−400 °C, respectively (Jambor and Dutrizac, 1998). These

dehydroxylation processes are endothermic, as shown by the endothermic broad band in the DSC trace. Organic matter also decomposes in this $T$ interval (Elert et al., 2015). The presence of organics led to a weight loss difference of ~0.6−1.1 wt% between runs carried out in air and in inert $N_2$ atmospheres (Fig. 10). Organic carbon undergoes oxidative combustion in air, which was reflected here by an exothermic peak at 350 °C in the DSC trace. Such an exothermic peak was absent in the run carried out in $N_2$. Note that the above-indicated weight loss values only represent a fraction of the organic carbon present in

the Saharan dust, because upon oxidative combustion, elemental carbon will be produced (charring effect) which we could not quantify. In any case, the values presented above are consistent with reported average value of organic carbon in Saharan dust aerosol (e.g., 1.7 wt%: Gonçalves et al., 2014). Analysis of organic matter in desert dust has shown that the elemental plus organic carbon content is on average < 2 wt%, with a large compositional variability (Jaenicke and Schütz, 1978; Eglinton et al., 2002; Gonçalves et al., 2014). Elemental carbon is typically associated with biomass burning (Eglington et

al., 2002), whereas organic carbon has been associated with more or less decomposed biological residues, including micro-organisms and microbial biofilms, as well as humic substances (Conen et al., 2011). Micro-organisms and organic residues in desert dust aerosol, which tend to cover mineral grains, have been suggested to be efficient atmospheric ice nuclei, thereby having an important indirect radiative forcing effect (Conen et al., 2011).

At $T$ >600 °C carbonates (first dolomite and subsequently calcite) decompose into CaO (or CaO+MgO in the case

of dolomite) releasing $CO_2$ (Rodriguez-Navarro et al., 2009; 2012). This is an endothermic process, as shown by the DSC traces. By measuring the total weight loss in the 600−950 °C interval, the total amount of carbonates was calculated to be 16 ±1 wt%, a value ~20% higher than the values determined by XRD (RIR) and weight loss following acid treatment. Apparently, TG analysis overestimates the carbonate content due to the contribution to the measured weight loss of residual OH released during clay minerals dehydroxylation at $T$ >600 °C. Note that complete dehydroxilation of most clay minerals

(at heating rates comparable to the one used here) is not complete up to ~700 °C (Guggenheim and van Groos, 2001).





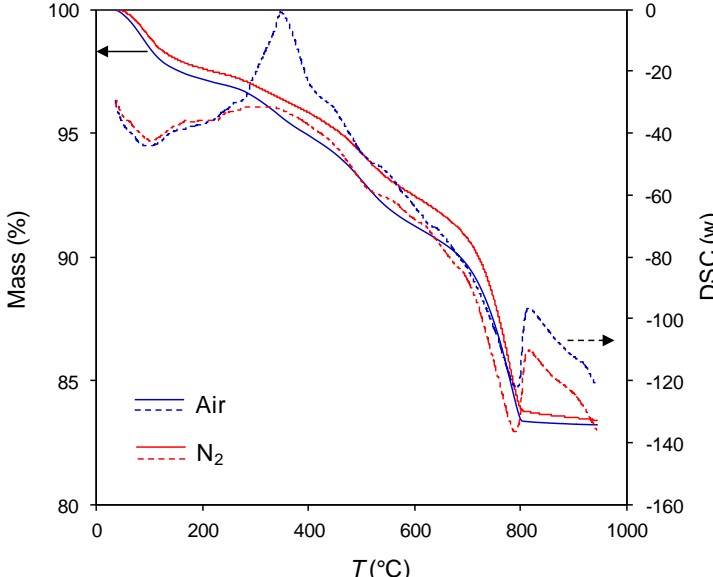

**Figure 10.** TG/DSC traces of Saharan dust collected in air (blue curves) and in inert $N_2$ (red curves) atmospheres.

Figure 11a shows the FTIR spectrum of the collected desert dust. The broad band centered at ~3400 cm[-1] corresponded to $\upsilon$OH stretching of structural OH in clays and iron oxyhydroxides, interlayer (solvation) $H_2O$ of clay minerals, and structural OH and water in ferrihydrite (Russell, 1979). The sharper peaks at 3693 and 3617 cm[-1], and the shoulder at 3637 cm[-1], corresponded to the $\upsilon$OH stretching of well-ordered (crystalline) gibbsite layers of dioctahedral phyllosilicates. The three bands are characteristic of kaolinite, the two at the highest wavenumber are common to dioctahedral smectites (i.e., montmorillonite), while the band at the lowest wavenumber is characteristic of illite. The small band at 3550 cm[-1] corresponds to the $\upsilon$OH stretching of the brucite layer in palygorskite (inset in Fig. 11a). Note, however, that ferrihydrite also has a characteristic $\upsilon$OH stretching band at ~3615 cm[-1] (Russell, 1979). We could not resolve the $\upsilon$OH band of goethite at ~3150 cm[-1], because it was masked by the broad OH band of clay minerals. However, the bands corresponding to the $\delta$OH in-plane deformation at 870 cm[-1] and the $\gamma$OH at 780 cm[-1] (Schwertmann et al., 1985) were observed, confirming the presence of this iron oxyhydroxide. The small bands at 2898 and 2984 cm[-1] corresponded to the C-H stretching of organic matter, thereby confirming the TG results showing the presence of organics in the Saharan dust. However, the lack of other well-defined bands precluded an unambiguous identification of the specific organic compounds in the dust. Nonetheless, the presence of a band at 1624 cm[-1], which can be ascribed to both $\delta$OH bending and COO[-] symmetric stretching, points to the presence of carbohydrates (e.g., polysaccharides) with carboxylic functional groups. The later would be consistent with the presence of microbial exopolymeric substances and/or humic substances (Conen et al., 2011). The broad band at 1406 cm[-1] corresponded to the $\upsilon_3$ anti-symmetric stretching of $CO_3^{2-}$ groups in both calcite and dolomite. The presence of calcite was confirmed by the very small $\upsilon_4$ bending band of $CO_3^{2-}$ groups at 712 cm[-1]. The strong




band at 986 cm$^{-1}$ and the shoulder at 1030 cm$^{-1}$ corresponded to the Si-O stretching of phyllosilicates, while the shoulder at 1090 cm$^{-1}$ corresponded to the Si-O stretching of quartz. The small peak at 908 cm$^{-1}$ corresponded to the Al-OH deformation of kaolinite. The doublet at 780 and 795 cm$^{-1}$ corresponded to the Si-O bending of quartz (note that these later bands overlap with those of goethite). The strong band at 515 cm$^{-1}$ corresponded to the Al-O-Si deformation in illite and smectites, while the strong bands at 460 and 430 cm$^{-1}$ corresponded to the Si-O-Si bending and Si-O deformation of phyllosilicates and quartz (Di Biagio et al., 2014). Finally, the very strong band at ~410 cm$^{-1}$ corresponded to the FeO$_6$ lattice mode of iron oxyhydroxides, including goethite, hematite and ferrihydrite.

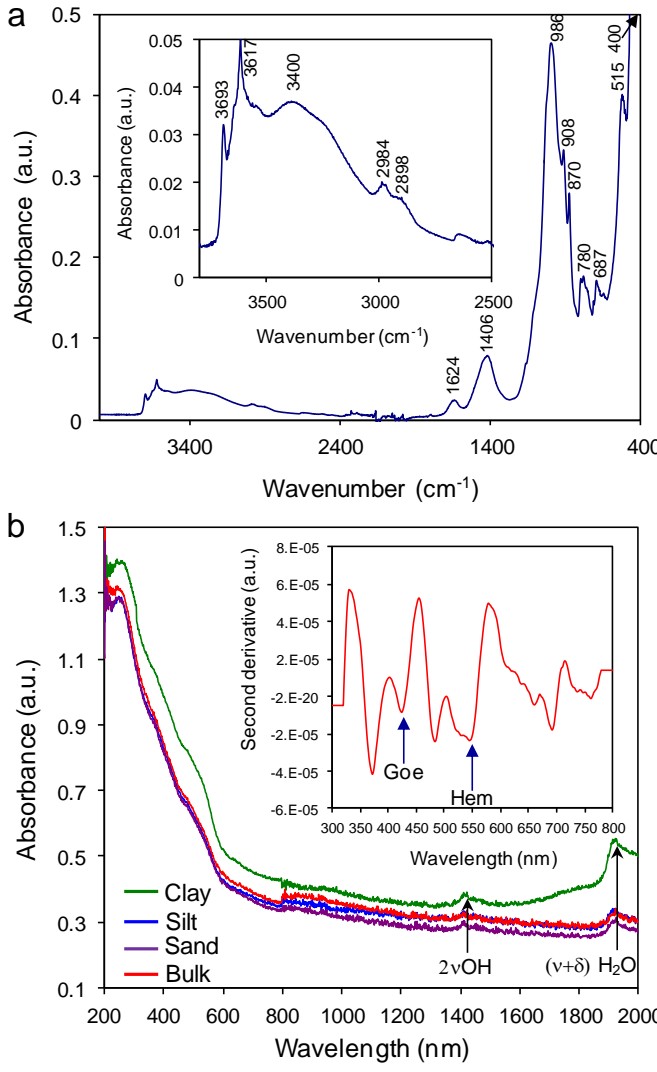

**Figure 11.** Spectral features of Saharan dust. a) FTIR spectrum of dust. The inset shows absorbance values for high wavenumbers (IR spectral region with lower wavelength values). Wavenumbers of main absorbance bands are indicated; b) UV-Vis-NIR spectra of bulk Saharan dust, and its three size fractions (clay, silt and sand).The inset shows the second derivative of the absorbance spectrum of the bulk sample. Blue arrows show the bands used for the calculation of goethite (Goe) and hematite (Hem) relative contents in the bulk sample.



The results of the FTIR analysis confirm the presence of the different mineral phases identified using XRD and TEM-AEM, as well as the presence of organic matter. They also show that this type of dust aerosol possesses strong longwave absorption, especially at thermal IR (6−24 μm) due to the abundant silicate, aluminosilicate and carbonate phases, along with minor iron oxyhydroxides. Absorption of IR radiation by desert dust has a direct positive (warming) radiative

forcing and, most importantly, it does not only operate during the day, as in the case of solar radiation, but also at night due to terrestrial thermal emission (Di Biaggio et al., 2014). Interestingly, Di Biaggio et al. (2014) showed that IR optical and radiative properties are highly variable among Saharan dust samples from different sources. This is confirmed by our results showing very similar spectral features but with absorbance values very different from those shown by the authors, specially at wavelengths >10 μm (wavenumbers <1000 cm$^{-1}$) where our dust samples displayed a much higher absorbance than those

of Di Biaggio et al. (2014).

Figure 11b shows the UV-Vis-NIR absorption spectra of the bulk dust and its three size fractions. In all cases, a strong increase in absorption at λ <600 nm was observed, and the absorbance was systematically higher for the clay fraction. The observed increase in the absorbance of the OH-stretching overtone (1412 nm) and the combination band of H$_2$O (1920 nm) of the clay fraction (Gionis et al., 2006), compared to those of the coarser fractions, is consistent with the higher amount

of clay minerals and iron oxyhydroxides in this finer fraction, both being responsible for the systematically higher shortwave absorbance. Remarkably, retrieved spectra (λ <1 μm) of the imaginary part of the complex refractive index, $k$, for several Saharan dust samples (Wagner et al., 2012) show strong similarities to the UV-Vis-NIR spectra shown here.

The UV-Vis-NIR spectra of the studied samples is standard for Saharan dust and has been associated with the presence of iron oxides, mainly goethite and hematite (Wagner et al., 2012; Formenti et al., 2014b). These oxides show a

remarkable increase in $k$ values at λ < 600 nm due to strong absorption associated with ligand-to-metal charge transfer transitions (Sherman and Waite, 1985). This is also the case of rutile, which has values of $k$ for UV radiation of the same order of magnitude of hematite (Utry et al., 2015), and should also contribute to the absorption spectra reported here. Note, however, that iron in the structure of clay minerals also contributes to the absorption of shortwave solar radiation because such octahedrally coordinated Fe$^{3+}$ can also experience charge transfer transitions. UV-Vis spectroscopic analyses of clays

such as illite, kaolinite, palygorskite and smectites with Fe contents very similar to those of the corresponding clays studied here, show strong absorption at λ < 600 nm due to oxo-to-iron (III) charge transfer (Karickhoff and Bailey, 1973).

From the second derivative of the UV-Vis spectrum (inset in Fig. 11b), the contents of goethite and hematite were calculated to be 49±1% and 51±1%, respectively. Our XRD results showed a higher goethite content of ~66 wt%. We consider, however, the UV spectroscopy results more reliable, due to the relatively high error associated with the

semiquantitative XRD analysis. The goethite content calculated using the second derivative of the UV-Vis spectrum is consistent with but slightly lower than the contents reported by Formenti et al. (2014b) for Saharan dust. The authors found goethite contents ranging from 52 up to 78 wt%. This discrepancy might likely be due to differences in the dust source areas. Note that an accurate evaluation of the content of these two oxyhydroxides is of outmost importance when determining the



direct radiative effect of desert dust, because the optical properties and, particularly, $k$ values of hematite and goethite differ significantly, the former phase showing stronger absorption at short wavelengths (Zhang et al., 2015).

### 3.8 Iron significance: Bioavailability

Iron in desert dust is a key player in a range of global biogeochemical processes. Iron is an essential micronutrient for all organisms, but is typically depleted in some inland water bodies, such as oligotrophic lakes (Vrede and Tranvik, 2006), and a large portion of open ocean waters (Jickells et al., 2005). Desert dust can supply iron to such areas, specially open ocean waters, enabling the proliferation of a range of microorganisms (e.g., phytoplankton) and the sequestration of $CO_2$ as biomass, thereby directly affecting primary production and indirectly influencing climate (Jickells et al., 2005). However, for its bioavailability, iron must be soluble or at least in a colloidal, poorly crystalline nanosized state, what has been defined as *filterable* iron, which passes through a 0.2 or 0.4 µm filter (see review by Raiswell and Canfield, 2012). But iron in desert dust minerals is typically insoluble (Shi et al., 2009). Three types of iron in desert dust have been defined (Shi et al., 2012): (i) amorphous or poorly crystalline iron oxyhydroxides (e.g., ferrihydrite) which is the most soluble iron fraction in desert dust; (ii) crystalline iron oxyhydroxides (goethite, hematite and magnetite) that are highly insoluble. These two groups form the so-called free iron; and (iii) structural iron, which is incorporated in the crystalline structure of aluminosilicates, mainly in clay minerals, and is also insoluble. Remarkably, the amount of soluble Fe in desert dust, expressed as the fractional iron solubility, FFS (i.e., fraction of soluble iron vs. total iron), typically increases during long-range transport from values ~1% to up to ~80% (Journet et al., 2008; Shi et al., 2012). This is due to various atmospheric processes including photoreduction (which according to Zhu et al., (1999), has only a minor effect), organic-mediated complexation, and chemical (acid) in-cloud and/or aereosol processing (Shi et al., 2009; 2012). Such an atmospheric processing, as well as the ultimate fate of iron once desert dust is deposited in distant locations, is strongly linked to where and how iron is incorporated in different desert dust minerals with different structure, solubility, crystallinity, and particle size. However, as pointed out by Raiswell and Canfield (2012), few studies of aeolian dust contain any detailed characterization of the iron mineralogy.

Our XRD results showed the presence of 1.1 wt% Fe free iron in crystalline iron oxyhydroxides (goethite and hematite). Our AEM analysis of individual clay minerals showed that structural Fe was also abundant in the Saharan dust particles. Considering the average Fe content in the different clay minerals identified here and their fractional content in the bulk sample, the structural iron in the clay minerals amounts to 2.04 wt% Fe (i.e., 57.3 % of the total Fe). This yields a total (free+structural) iron content of 3.14 wt %. This value is lower than the average total iron content of 3.56 wt% determined by XRF and ICP-OES analyses (see Table 3). It could be argued that the missing Fe (0.42 wt%) is incorporated in other silicates such as feldspars. However, the amount of Fe in such phases is either very low or negligible. If we consider the values of 0.13−0.54 wt% Fe in feldspars reported by Journet et al. (2008) and the fractional content of feldspars (0.13) in our dust samples, this yields 0.01−0.07 wt% structural Fe in such tectosilicates. This is indeed a negligible amount. It follows that the missing (free) iron has to be incorporated in the amorphous and/or poorly crystalline ferrihydrite detected using



TEM-SEAD, which can not be quantified as free iron by XRD because this later technique does not identify amorphous phases. The amorphous and/or poorly crystalline ferrihydrite would thus amount to 28 wt% of the free iron.

The amount of structural Fe in our studied dust is consistent with, although slightly higher than the values of ~40−50 % commonly reported for Saharan dust (Formenti et al., 2011; 2014a,b). The presence of such a relatively high amount of structural iron is an important finding due to its potential bioavailability (Journet et al., 2008; Formenti et al., 2014a). In dioctahedral clays, such as illite and kaolinite, as well as montmorillonite and MLC, iron is present in the octahedral layer as $Fe^{3+}$, alone or in association with minor amounts of $Fe^{2+}$ (Weaver and Pollard, 1973; Mestdagh et al., 1980; Johnston and Cardile, 1987). In the case of palygorskite, which has a mixed di- and trioctahedral character, $Mg^{2+}$ in the octahedral layer can be substituted by both $Fe^{2+}$ and $Fe^{3+}$ (Gionis et al., 2006). In purely dioctahedral clay minerals such as chlorite iron is incorporated as $Fe^{2+}$ substituting $Mg^{2+}$ in octahedral positions both in the 2:1 structural unit and in the interlayer brucite layers (Weaver and Pollard, 1973). Interestingly, Cwiertny et al. (2008) observed a correlation between $Fe^{2+}$ content in aluminosilicates and increased FFS after acid processing of desert dust, and Schroth et al. (2009) concluded that $Fe^{2+}$-bearing silicates are more soluble than iron oxyhydroxides, being the former an important source of bioavailable soluble iron. In addition to the effect that $Fe^{2+}$ may have on the potential solubility of structural iron, a significant amount of our studied clay particles with relatively high Fe content were <100 nm in size, which makes them significantly more soluble than larger particles (Raiswell and Canfield, 2012). Overall, these compositional and size effects may favor the release of soluble structural iron upon atmospheric processing, and/or post-atmospheric processing once the dust particles are deposited in the ocean or in inland water bodies.

In agreement with previous studies, the free iron in our samples is mainly present as hematite and goethite (Shi et al., 2012; Formenti et al., 2014b). However, we detected a significant amount of amorphous/poorly crystalline ferrihydrite. Although its amount is higher than that (up to 7.4%) reported by Shi et al. (2011) for Saharan soils, it is markedly smaller than the ferrihydrite content of ~71 % reported by Schroth et al. (2009) for north African dust deposited on a buoy in the northeast Atlantic Ocean. The existence of such a huge concentration of ferrihydrite in Saharan dust has been, however, questioned by Shi et al. (2011) and Raiswell and Canfield (2012) on the basis that the quantification of such an amorphous or poorly crystalline phase is difficult. The authors also state that the very low FFS (<1%) reported by Schroth et al.(2009) is not consistent with such a huge amount of ferrihydrite, because experimental evidence shows that ferrihydrite nanoparticles are significantly more soluble than clay minerals and/or crystalline goethite or hematite. Shi et al. (2012) conclude that ferrihydrite nanoparticles, such as those identified here, are the most likely source of soluble and bioavailable Fe in desert dust. How, where, and when such Fe-rich nanoparticles are formed has been a matter of intensive research and discussion. Two not mutually excluding possibilities have been considered: (i) iron-rich nanoparticles are already present in the dust source region, and are formed due to (limited) weathering of crystalline iron-containing phases such as clays (see above) and/or iron oxyhydroxides (goethite and hematite) (Shi et al., 2011; Raiswell, 2011), and/or (ii) they are formed during dust transport via atmospheric processing (Shi et al., 2012).





With a few exceptions (e.g., Shi et al., 2011), the presence of amorphous or poorly crystalline iron oxyhydroxides such as ferrihydrite in Saharan dust source regions has been neglected (e.g., Lafon et al., 2006; Debout et al., 2012; Formenti et al., 2014b). It is considered unlikely that metastable ferrihydrite could survive in the source region without rapidly transforming into more stable goethite or hematite (Shi et al., 2012). Nonetheless, ferrihydrite has been shown to remain

untransformed for a few hundred days at STP conditions (Raiswell and Canfield, 2012), and silica, clay minerals, and a range of organic substances (all present in Saharan soils and dust) reportedly contribute to its stabilization (Jambor and Dutrizac, 1998). It is thus very likely that the amorphous iron oxyhydroxide and/or poorly crystalline ferrihydrite nanoparticles in our dust samples were already present in the entrained soil. Indeed, dissolution of iron-containing clay minerals following intermittent exposure to aqueous solutions during, for instance, rain events, fluvial transport and/or

flooding in (ephemeral) lake waters, is a plausible mechanism for the formation of iron-rich nanoparticles (ferrihydrite) in the dust source areas (Shi et al., 2011; Canfield and Raiswell, 2012). This is consistent with results by Poulton and Raiswell (2005) showing that clay minerals in natural riverine environments commonly are associated with iron-rich nanoparticles.

Regarding the second hypothesis for the formation of iron-rich nanoparticles in Saharan dust, numerous field and laboratory studies have shown that atmospheric processing strongly contributes to the formation of soluble iron (Shi et al.,

2012). Shi et al. (2009) showed that Saharan dust wet deposited in the western Mediterranean included highly soluble, bioavailable, and poorly crystalline nanosized 2-line ferrihydrite, which contained trace concentrations of Al, Cr, Si and Ca. The later might indicate formation by atmospheric processing of clay minerals. The authors found no such nanosized ferrihydrite in Saharan dust dry deposited in the eastern Mediterranean (collected during a different event and presumably coming from a different source area). Parallel experiments using Saharan soils and pure synthetic goethite confirmed that

precipitation of nanosized ferrihydrite occurs after acid leaching, favored during (partial) drying and formation of wet mineral aerosols, and subsequent pH increase during simulated in-cloud processing (Shi et al., 2009; 2012). These observations might suggest that acid cloud processing during wet deposition led to the formation of iron rich nanoparticles. However, the lack of spatial and temporal relationship between the wet and dry deposition events studied by Shi et al. (2009) precludes drawing any final conclusion regarding whether the iron nanoparticles were already present in the entrained desert

dust prior to wet deposition or were the result of atmospheric processing. In any case, a prerequisite for in-cloud and aerosol acid processing of iron-containing phases is the interaction with acid pollutant gases (e.g., $SO_2$ and $NO_x$), and the absence of carbonates or their external mixing. Ito and Feng (2010) underlined that internally mixed carbonates will buffer atmospheric acid-processing of iron-containing phases, thereby strongly limiting the formation of potentially bioavailable, poorly crystalline or amorphous iron phases. Our HAADF analyses (Figs. 7 and 8) clearly show that carbonates were internally

mixed with iron oxyhydroxide nanoparticles and were not affected by acid dissolution. Moreover, we detected no sulfate by-products such as gypsum or (calcium) nitrates. It follows that the iron-rich nanoparticles in the Saharan dust deposited during the studied red rain event must have already been present in the source areas. This is an important result when considering the bioavailability of Fe in Saharan dust. It shows that a significant fraction of Fe in Saharan dust is already present in the source region as potentially bioavailable nanosized amorphous and/or poorly crystalline iron oxihydroxides (Shi et al.,



2011). It also shows that the presence of acid gases and their associated atmospheric acid-processing of iron-phases is not absolutely necessary to have nanosized, more soluble and potentially more bioavailable Fe-rich phases in desert dust. Our results suggest that the importance of such anthropogenic acid gases in enabling the delivery of soluble and bioavailable Fe in desert dust to open oceans, and their subsequent impact on $CO_2$ drawdown (Li et al., 2012) might be overestimated.

5       Another important aspect to consider regarding the bioavailability of iron-containing phases is their interaction with organic compounds such as carboxylic acids, or more complex molecules having different functional groups. They can complex Fe, facilitating the dissolution of iron-containing phases (especially clay minerals) and its bioavailablity (see review by Shi et al., 2012), as demonstrated for the case of several organic acids, such as oxalic and humic acids (Paris and Desboeufs, 2013). Our TG/DSC and FTIR results showed the presence of organic carbon. It is however not clear what role

such organic carbon played in the possible processing of iron oxyhydroxides during Saharan dust transport and scavenging. We can only hypothesize that the presence of abundant functional groups (e.g., carboxylic groups) in such organics could enable the complexation of Fe and facilitate its bioavailability.

### 3.9 Effects of mineralogy, mixing state, and PSD on dust direct radiative forcing

Scattering and absorption of incoming solar (shortwave) and outgoing thermal (longwave) radiation by desert dust aerosol

have a cooling effect at land surface and a warming effect at tropospheric levels (Carlson and Benjamin, 1980; Alpert et al., 1998). However, the magnitude and even the sign of the direct radiative forcing are not well constrained. While  some researchers consider that the net radiative forcing of mineral dust on the climate system is negative (Gieré and Querol 2010; Allen et al., 2016), others report that under specific scenarios the direct forcing can be positive (Carlson and Benjamin, 1980), leading to regional (Overpeck et al., 1996) or even global warming (Kok et al., 2017). The uncertainties regarding the

sign and magnitude of the direct radiative forcing of desert dust are rooted in the fact that they depend on many poorly constrained factors such as: (i) the characteristics of mineral dust (concentration, vertical distribution, PSD, shape, internal/external mixing, and composition/mineralogy) and (ii) external variables such as surface albedo below dust plumes, temperature at ground level, and presence/absence of clouds (Balkanski et al., 2007; Kemppinen et al., 2015). Composition/mineralogy and particle size, as well as mixing state, appear to be the most critical factors controlling dust

direct radiative forcing (Zhang et al., 2015).

       Regarding composition/mineralogy, iron-containing phases play a key role in the absorption and scattering of solar and terrestrial radiation (Tegen and Lacis, 1996; Sokolik and Toon, 1999; Zhang et al., 2015). However, most models for the direct radiative forcing of desert dust typically only consider the presence of hematite (e.g., Balkanski et al., 2007; Wagner et al., 2012). Our results, as well as those of others (e.g., Formenti et al., 2014b; Zhang et al., 2015), show that this is an

oversimplification that may have an important impact in the outcome of such models, because in addition to hematite, goethite and iron oxyhydroxide nanoparticles (ferrihydrite) are also present in significant amounts in Saharan dust. In addition, their actual mixing state is another important aspect of desert dust that has been generally neglected in climate models. While an external mixing of (alumino)silicates and iron oxyhydroxide particles in proportions typically found in



desert dust has a net negative radiative forcing, their internal mixing can lead to a net positive radiative forcing (Sokolik and Toon, 1999). The later situation is the one observed here: iron oxyhydroxides, generally forming nanosized aggregates and concentrated in the clay fraction, are internally mixed with carbonate, silicate and aluminosilicate particles (SEM and TEM results). This appears to be a general feature of Saharan (Deboudt et al., 2012; Wagner et al., 2012; Jeong et al., 2016) and Asian desert dust (Jeong and Achterberg, 2014). We observed that iron oxyhydroxides are the main contributor to the strong absorption of UV-Vis radiation regardless size-fraction (Fig. 11), which is consistent with our electron microscopy observations showing aggregates of iron-containing nanoparticles closely cemented with clay minerals, and covering (i.e., internal mixing) larger silt- and sand-sized particles (Figs. 6-8). Note, however, that clay minerals containing structural iron can also contribute to the absorption of shortwave solar radiation as indicated above. A few models consider the light absorption behavior of illite alone or internally mixed with iron oxyhydroxides, showing that illite actually displays relatively strong shortwave absorption (Zhang et al., 2015). However, no model has ever consider the effect that structural iron has on the absorption properties of other typical clay minerals present in desert dust such as smectites, kaolinite, palygorskite, chlorite and MLC. It should be noted that the combined effect of clay minerals and iron oxyhydroxides is actually responsible for the shortwave radiative effects of the studied Saharan dust. The longwave radiative effect, basically absorption of thermal radiation, is mainly associated with silicate and aluminosilicate phases (including clay minerals) as well as carbonate phases. It could be argued that the combined shortwave and longwave scattering/absorption of internally mixed clay minerals/iron oxyhydroxides/(alumino)silicates or carbonates will be the relevant and overall radiative effect of the Saharan dust. According to Sokolic and Toon (1999) such an internal mixing would likely have a net positive direct radiative forcing under specific circumstances (i.e., surface albedo, dust load, vertical distribution, and surface *T*).

In addition to these compositional/mineralogical and mixing effects on dust radiative forcing, another critical aspect to be considered is the PDS of desert dust. Smaller particles (i.e., clay fraction) are more effective in scattering solar radiation than larger ones. The later, in turn, are more effective in absorbing energy (solar and thermal) (Tegen, 2003; Otto et al., 2007). Most models of dust radiative forcing typically consider the smallest particles only (geometric diameter <10 µm) (Tegen and Lacis, 1996; Tegen, 2003). Such small sizes, however, do not represent the actual size of dust particles in major dust events, which in turn are the ones that most significantly contribute to entrain and transport desert mineral dust (Skonieczny et al., 2011; Mahowald et al., 2014; Kok et al., 2017). Indeed, one of the causes of uncertainty in climate models is that the size distribution of dust particles is poorly constrained and typically the amount of smaller particles is overestimated (Kok, 2011). Although large (>10 µm) particles have been considered to settle by gravitational forces within hours after entrainment (Tegen, 2003), experimental evidence shows that large, and even giant particles with size >50 µm, can be transported for days over distances of several thousands of km (Pitty, 1968; Franzén, 1989; Betzer et al., 1988). Experimental observations have also shown that during intense dust events a significant amount of large particles (> 30 µm) are transported for more than 12 h (Ryder et al., 2013), and fast gravitational settling seems to be prevented by atmospheric processes involving upward air movement due to solar heating of the dust and/or intermittent turbulence (Maring et al., 2003). Underestimating the contribution of large, long-range transported desert dust particles (geometric diameter > 10 µm),



has a direct impact in the outcome of radiative forcing models as well as in the estimation of global dust emissions (Kok, 2011). Models would overestimate cooling by fine particles due to their scattering of solar radiation, neglecting the fact that coarser particles can induce a net warming by absorbing both solar and thermal radiation (Otto et al., 2007; Kok et al., 2017). Also, most global circulation models are tuned to match radiative measurements, so that an overestimation of the radiative

cooling induced by clay-sized particles will be compensated by a reduction in the modeled quantity of emitted dust (Kok, 2011). Recent modeling results indicate that a positive direct radiative forcing at the top-of-the-atmosphere (TOA), leading to global warming, can be achieved when realistically considering the amount of very coarse (geometric diameter > 10 µm) mineral dust particles (Kok et al., 2017), which are quite abundant close to source areas and, as shown by our results, can reach far downwind regions during extreme dust events. It is thus very likely that coarse desert dust would have a net

positive radiative forcing close to source areas and over continental land. Although the residence time of such larger particles is only a few tens h, their persistent emission and transport may have a significant radiative effect not only locally or regionally, but also globally. This is particularly relevant under a global warming scenario resulting in increased desert dust strength as predicted by recent modeling results (Kok et al., 2018).

## 3.10 Health hazard

Most studies on the impact of Saharan mineral dust on human health have focused on its short term effects. In particular, a positive relationship between hospital incidences and mortality, and desert dust outbreaks have been established, especially for the case of $PM_{10}$ (Perez et al., 2008). However, both a negative and a positive correlation between desert dust $PM_{10}$ and mortality has been established (Karanasiou et al., 2012; Zhang et al., 2016). Yet, although there is a current lack of detailed knowledge, recent research shows that there is a clear positive correlation between exposure to natural mineral dust $PM_{2.5}$

and human mortality associated with respiratory and cardiovascular health issues (Zhang et al., 2016). Modeling of the impact of dust $PM_{2.5}$ estimates that the global fraction of cardiopulmonary deaths caused by desert dust aerosols is ~1.8%, this value being ~15−50% in countries of the so-called "dust belt" (Giannadaki et al., 2014). These studies underline the fact that the clay fraction of desert dust is the one that has the most deleterious effects in the short term.

Little is known, however, about the long term health effects of desert dust exposure and inhalation. This could be

the case of the potential fibrogenic and carcinogenic risk of such inhalable mineral particles. Such a potential long term health effect of desert dust has been, however, overly neglected. One exception is the study by Giannadaki et al. (2014) who found a link between cardiovascular and lung cancer death with desert dust $PM_{2.5}$. Nonetheless, the authors ignored the role of larger particles and the actual mineralogy of dust was not taken into account for health-risk evaluation. Overall, the role of specific minerals in desert dust, particularly the abundant clay minerals, and among them, the fibrous clays, was neglected.

Moreover, the possible link between such potentially carcinogenic fibrous minerals and Fe, an element which has been associated with increased risk for cancer development following exposure to fibrous minerals due to its capacity to generate free radicals via the Fenton reaction (Nolan et al., 1991; Ghio et al., 2004), was also neglected.



Several studies have focused on the evaluation of the potential health risk of clay minerals and associated phases present in dust (Plumlee et al., 2006). For instance, kaolinite has been reported to be a potential respiratory hazard. However, quartz, typically associated with clay minerals such as kaolinite, seems to play an overruling role in the respiratory illnesses associated with clay dust inhalation (Carretero et al., 2006). Indeed, silica dust inhalation has been shown to be fibrogenic and carcinogenic (Ding et al., 2002). Palygorskite, which is the most abundant clay mineral in the studied Saharan dust samples, is a non-regulated fibrous mineral reported to be carcinogenic and cytotoxic especially if fiber length is over 5 μm (Rödelsperger et al., 1987), even if its content is less than 1 % in mineral dust (Nolan et al., 1991). There are, however, conflicting results regarding the health effects of palygorskite. While a few studies have shown no toxicity to human embryonic intestinal cells or low toxicity to rat pleural mesothelial cells, palygorskite has been reported to induce hemolysis and cytotoxicity in mouse, rat and rabbit macrophages, and bovine and human endothelial cells (see Larson et al., 2016, and references therein). Moreover, inhalation tests in rats have shown that large palygorskite fibers can induce bronchoalveolar hyperplasia, alveolar tumors and mesothelioma (Donaldson and Borm, 2006). Iron presence either in the mineral structure or elsewhere (adsorbed colloidal particles) seems to enhance its carcinogenic potential (Nolan et al., 1991). This is highly relevant for the palygorskite fibers in Saharan dust deposited in the Iberian Peninsula and studied here, as well as for all the southern and eastern European areas where abundant palygorkite in desert dust has been detected. Importantly, in addition to their abundance in Saharan dust, as shown here, and their high crystallinity and fiber length over several μm, they include structural iron and are associated with abundant poorly-crystalline (colloidal) iron oxyhydroxides, typically attached to the clay mineral surfaces (Fig. S5). Note, however, that not all palygorskite fibers are equally hazardous. Poorly crystalline, small fibers (less than 1 μm long) have been shown to be non-carcinogenic (Larson et al., 2016).

## 4 Conclusions

We have studied with an unprecedented level of detail the physicochemical and compositional features (chemistry, mineralogy, size, and mixing state, as well as radiative-relevant properties) of Saharan dust particles wet deposited following an extreme winter red rain event that affected southern Spain during February 21−23, 2017.

This event was triggered by a north African cyclone, a recurrent synoptic situation leading to major Saharan dust outbreaks, often resulting in red rains and affecting the western Mediterranean area especially in winter and early spring. Such an extreme aeolian event accounts for most of the total yearly dust depositional flux to the Iberian Peninsula. Likely, major and extreme events like the one studied here also represent the most significant pulse(s) for the global desert dust mass transported in the atmosphere. The data presented and discussed here are thus of high significance to constraint input values, including dust composition, mineralogy, mixing state and size, as well as depositional fluxes, for more realistic dust and climatic models.



Abundant clay minerals (palygorskite, illite, smectite, mixed layer clays, kaolinite and chlorite), quartz, and carbonates (calcite and dolomite), with minor amounts of K- and Na-feldspars, iron oxyhydroxides (goethite, hematite and amorphous/poorly crystalline ferrihydrite), rutile and ilmenite were identified.

The specific mineralogy of phyllosilicates and the abundance of carbonates, complemented by the analysis of satellite imagery and back/forward trajectories, helped us to identify two probable dust source areas: (i) south/central Algeria, north Mali and northwestern Niger (PSA3), and (ii) north Algeria, south Tunisia and north-western Libia (PSA1). Activation of different source areas resulted in complex dust mixtures during extreme Saharan dust events such as the one studied here.

Detailed HAADF imaging and AEM analysis of individual clay mineral particles showed that they include abundant structural iron and are typically associated with nanogranular aggregates of iron oxyhydroxide nanoparticles. Clays plus iron-rich nanoparticles of amorphous and/or poorly-crystalline ferrihydrite tend to form rims lining quartz, carbonates and feldspars (internally mixed).

The internal mixing of calcite and clay minerals/iron-rich nanoparticles, and the absence of secondary calcium sulfates or nitrates, demonstrate that acid cloud processing of iron-containing clays and/or crystalline iron oxyhydroxides was not the main source for amorphous and/or poorly crystalline ferrihydrite nanoparticles. Our results show that the formation of such iron-rich nanoparticles, which are an important potential source of bioavailable iron, already occurred in the dust source areas. The claim that pollutant acid gases are a prerequisite for iron bioavailability in Saharan dust, implying that current reduction in anthropogenic acid gas pollutants may have a negative feedback on atmospheric $CO_2$ sequestration in ocean waters, should be reconsidered.

Iron-containing clays and iron oxyhydroxides internally mixed with silicate and carbonate particles are responsible for the strong absorption of solar (UV-Vis) and thermal (IR) radiation of the studied Saharan dust. These internally mixed phases form abundant aggregates with size >10 µm. Such abundant coarse and even giant particles can have a significant impact on the direct radiative forcing of Saharan dust, likely inducing a net warming effect. The later might have strong regional (i.e., northern Africa and south-western Europe) and even global climate implications.

The presence of abundant, inhalable, micrometer-sized palygorskite fibers, associated with Fe-rich nanoparticles in Saharan dust might pose a significant health hazard. This health issue should be considered in order to take appropriate preventive measures when extreme Saharan dust plumes reach densely populated areas (e.g., stay-indoors warnings, use of respirator masks).

Our results validate the multianalytical approach used here, as it enables the analysis of the mineralogical and physicochemical properties of Saharan dust mineral aerosols with an unprecedented level of completeness and detail. Such a level of completeness and detail is necessary to disclose sources areas and processes affecting transported desert dust, and to fully evaluate and model the multiple effects and impacts of desert dust.



Ultimately, we show that extreme events involving red rains thwart any other standard wet and dry Saharan dust deposition events taking place in the western Mediterranean. Their past, present, and future trends, and impacts should thus be taken into account when modeling and evaluating the manifold effects of the desert dust cycle.

**Acknowledgements:**

This work was supported by the Spanish Government (grant CGL2015-70642-R), the Junta de Andalucía (research group RNM-179), and the University of Granada (Unidad Científica de Excelencia UCE-PP2016-05). We thank the personnel of the Centro de Instrumentación Científica (CIC; University of Granada) for analytical assistance. We gratefully acknowledge the NOAA Air Resources Laboratory (ARL) for the provision of the HYSPLIT transport and dispersion model and READY website (http://www.ready.noaa.gov) used here. We also thank NASA/NOAA for Suomi NPP VIIRS as well as Terra and

Aqua MODIS satellite imagery gathered using the Land Atmosphere Near real-time Capability for EOS (LANCE) system and services from the Global Imagery Browse Services (GIBS), both operated by the NASA Earth Observing System Data and Information System (EOSDIS). Reanalysis of the synoptic scale meteorological scenario was provided by U.S. Department of Commerce | National Oceanic and Atmospheric Administration Earth System Research Laboratory | Physical Sciences Division NOAA, http://www.esrl.noaa.gov/psd/data/composites/hour/.

**Competing interests**: The authors declare that they have no conflict of interest.

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
