# Peer review of "Mineralogy and physicochemical features of Saharan dust wet deposited in the Iberian Peninsula during an extreme red rain event"

_Atmospheric Chemistry and Physics, 2018_

## Referee Comment (RC1) · J. Cuadros (Referee) · 23 Mar 2018

General assessment

This ms. characterizes the mineralogy and chemistry of the Saharan dust deposited in Granada, Spain, in one extreme dust event. It also investigates the trajectory of the winds carrying the dust and the dust sources. These results are compared with abundant literature of Saharan dust transport, mineralogy and chemistry. All the above is a very complete study carried out with competence and thoroughness. Beyond this,

the ms includes a lengthy discussion on three broad issues: Fe bioavailability in Saharan dust, effect of Saharan atmospheric dust on radiative forcing and effect on human health. These discussions are kept at a general, non-specific level and are given a relevance which is not warranted either by the data provided in the ms or the depth of the discussion. After all, the ms provides data from one event only, and this and extreme event, rare by definition. The ms does not contribute data with sufficient perspective to allow the wide discussion on the above issues. In my opinion, this discussion needs to be re-written in a focused way, more closely linked to the ms results and, if possible, using a quantitative approach that illustrates the magnitude of the relevance of the results. Such amended version of the ms would be within the scope of ACP. If the discussions on Fe bioavailability, effect on radiative forcing and effect on human health were removed the ms would have a much smaller scope, consisting on the detailed characterization of an extreme dust event.

The text can be condensed. Some ideas are repeated several times in the ms.

Major issues

Discussion of Fe bioavailability. A lot of this discussion reviews known facts and discusses them in a general way, without quantitative data. The significance of the results from the ms for Fe bioavailability is not sufficiently supported. In my opinion, the authors could try to assess the amount of bioavailable Fe in the investigated dust and compare it with quantitative results in the literature. They could also try to project the results to marine areas seeded by the Sahara, comparing with previous estimates. Even if these projections have large uncertainty, they will allow some assessment of whether the results in the ms suggest changes to our understanding of input of bioavailable Fe to the ocean from the Sahara.

The authors state that the amount of soluble Fe typically increasing during long-range transport, from ~1% up to 80%. The simplest mechanism to explain this is that the larger particles drop during transport, causing a concentration of smaller, more soluble

particles. With this mechanism, the total amount of soluble Fe in dust does not increase with transported distance. Fe-solubility, as indicated by the authors (lines 7-10 on p. 31), is mainly a function of particle size. I do not know that there is evidence of bioavailability of $Fe^{2+}$ or $Fe^{3+}$ in clay minerals, unless they are nano-particles. $Fe^{2+}$ is stable in non-expandable clay minerals, and there is virtually none of it in expandable clays. Ocean basins preserve a clay record which is generally consistent with clay mineralogy and composition in the continental sources, indicating low reactivity.

In their assessment of the amount of amorphous Fe phases in the Saharan dust, the authors do not discuss Fe in carbonates: is there any Fe in carbonates? How would this modify the figure of "missing" 0.42 wt% Fe?

Discussion of radiative forcing. This discussion is non-specific and misses some important facts. First, the ms investigates a type of dust event which is not frequent. The amount of time that dust particles such as those described in the ms exist in the atmosphere could be negligible in comparison with the usual atmospheric situation. Thus, the relevance of these data for radiative forcing is questionable. The authors indicate that small particles form aggregates with the large particles. This would mean that the small particles will settle quickly with the large ones and remain suspended only for the short time that the strong winds persist. Second, the authors have no knowledge of particle size distribution vertically in the atmosphere during the dust event; yet, this knowledge is essential to assess the effect on radiative forcing, as they indicate.

It is stated that although atmospheric dust generates negative radiative forcing, "internal mixing" of minerals may lead to positive forcing. Why is this? This point is central to the entire discussion and it has to be explained. However, how do the authors know that the mineral particle aggregation that they observe was not produced during the rain that settled the particles? The cementation of mineral particles is mentioned but it is not clear where this knowledge originates. We are all familiar with the strong cohesive forces between mineral particles generated by wetting them and letting them dry; one example of this is the preparation of oriented aggregates for clay examination. If

the internal mixing in the dust samples was a result of rain, the particles suspended in the atmosphere would have not been "internally mixed".

These considerations call for a concise discussion on radiative forcing. In such discussion the authors could point out some mineralogical facts from their samples that may not have been considered in models of dust-radiation interaction, leaving the question open of whether or not they are relevant to the issue.

Health hazard. This discussion consists on a review of previous studies and an enumeration of aspects that have not been considered in health-mineral studies. Again, the main criticism here is that data from the ms may not be relevant to health issues because they refer to infrequent events. Presumably, people living around the Sahara take shelter during extreme storms. The situation relevant to health is that of habitual, more moderate winds. Instead of pointing out aspects that have not been considered in these studies in the past, it would be more helpful to identify one or two questions learned from this study that may be of interest for health issues, whether or not they prove to be relevant in the long run.

At the end of the 1st paragraph in this section there is the assertion that the clay fraction of desert dust has the "most" harmful effects in the short term. This assertion seems to be founded on the studies of health effects of PM2.5. However, as there is no comparison with the effects of other PM values, the conclusion that the clay fraction has the most harmful effects is not supported.

In the three sections on relevance, the authors frequently assert that this or that issue or aspect has been "neglected" in previous studies. This is probably not the right word, as it carries a connotation of oversight or incompleteness in such studies. All three issues are very complex and the number of variables is beyond what any single study can consider. In this situation simplifications are necessary and the most relevant variables have to be selected. It would be preferable to indicate that certain analyses of dust composition would help to obtain more realistic results.

Technical issues

I am not qualified to comment on the investigation of wind trajectories.

P. 13, line 18. Avila et al (1997) did not report goethite, hematite or rutile (?). Of course, the amount of Fe and Ti oxide is small and can be missed in a mineralogical study. However, was not the dust red? If so, how did they account for this colour?

Methods. As sample aliquots were treated in different ways, it would be helpful to indicate at the beginning of the description of each technique what treatment the corresponding aliquot had received.

Figure 4. The lower part of the panels, below the maps, is unreadable. Would there be expanded versions of the figures in the final production?

Figure 5. There are several questions in this figure. Did the authors carry out background subtraction in the XRD patterns of panels c and d? See that the background is lower to the left of the low-angle peaks than to the right of these peaks (compare with the XRD patterns in panels a and b). Such procedure can deform the XRD pattern significantly in this area and blur the corresponding information. The goethite and hematite peaks are not indicated in panels c and d. It would be helpful to label panels c and d with "clay" and "silt" for an easier identification. The figure contains only information on $°2\theta$, whereas the caption and text use Å: non-frequent users of XRD will find this difficult to follow.

P. 15, lines 22-25. It is also possible that single sources contain kaolinite of two different crystal orders. The reasons for this can be multiple. Among them, wind and dust deposition form several areas within the Sahara.

Table 2 is completely unclear. It needs to be thoroughly checked for the correctness of the labels and heading. For example, it seems to indicate that the proportion of smectite continually increases from the clay to the silt fraction and to the bulk, which would be remarkable. Actually, the text in p. 17, line 13, is incompatible with the results

in Table 2.

Table 4 and p. 20, lines 32-34. The differentiation between smectite and illite-smectite (I-S, or the MLC, mixed-layered clays) is not supported. The authors ground this differentiation on tetrahedral substitution (of Si by Al). Typically, I-S minerals have more tetrahedral Al than smectite, but this is not the only or even most important difference. The central questions are that, in I-S, both the layer charge and K content are higher. Comparing the results for the analyses labelled "smectite" and "I-S" one can see that this is not the case: there are fewer interlayer cations in I-S (this should mean that the layer charge is lower, if the formulas are correct, which I have not ckecked), and the amount of K is basically the same. According to these data, it is not possible to discriminate between these two groups of analyses as corresponding to smectite and I-S. The results indicate, perhaps, smectite with a wide range of tetrahedral substitution. The authors mention a beidellitic component in smectite. Beidellite typically has more tetrahedral Al than montmorillonite.

P. 22, lines 1-3.The authors can check their data for a possible contamination of clay minerals with Fe-oxides. Structural formulas of uncontaminated dioctahedral clay minerals should fulfil the following conditions: Sum of tetrahedral cations = 4 (2:1 minerals) or 2 (kaolinite); the sum of octahedral cations is in the range 1.98-2.2; the interlayer charge calculated from the lattice cations is equal to that calculated from the interlayer cations (within perhaps $\pm0.01$); the interlayer charge is within the known range for the mineral suggested by the formula. The authors could include the above checks in Table 4.

P. 24, top. This is the first time that the authors refer to internal and external mineral mixing, and the meaning should be provided here. The explanation should provide the basis for the relevance that is attached to this difference in further discussion, which is not clear. In my opinion, this is an important point that requires complete clarity in the description of the physical differences between one and the other type of mixing.

P. 24, line 17: "well-cemented". This is not an observation from FESEM, obviously. Which technique provided this observation?

P. 25, lines 8-10. "The samples studied": with what technique? "In agreement with the results from the hydrodynamic size-separation": what were the data?

Figure 9. The inset spectrum in panel b cannot be read.

P. 26, lines 2-3. This assertion needs to be supported with a reference or an explanation.

P. 26, lines 6-9. Is this correct? Why nano-pores should render the dust more reactive than mesopores? Is it related to the area of mineral particles exposed to water/atmosphere? This point needs developing.

P. 26, last paragraph. Observing the DSC curves in Figure 10, it appears that the sharp peak corresponding to carbonate decomposition starts a $\sim$700 °C and ends some degrees above 800 °C. The slope of the DSC curves between 600 and 700 °C is less pronounced and probably due to dehydroxylation events. Also, water is very frequently lost for a range of temperature after the dehydroxylation temperature in clays. Perhaps the authors could recalculate the weight loss in the narrower temperature range indicated above to test whether the resulting carbonate content is then coherent with those of XRD and weight loss after acid treatment.

Description of IR results in pp. 28 and ff. This description has more detail than can be appreciated in Figure 11, and thus the authors could consider shortening it. Several IR bands referred to in the text are not labelled in the figure. There are several band assignments that need reconsideration or rewriting for a more accurate description. 1) Most, if not all, the IR absorption below 3500 cm-1 (excepting the organic bands) is due to hydration water, so it would be more accurate to say that the band centered at 3400 cm-1 corresponds to water. 2) The sharp bands at 3893 and 3617 cm-1 are from kaolinite; the illite and smectite bands at 3650-3550 cm-1 (depending on composition)

are always wider and they can be seen overlapping the 3617 cm-1 band, causing the wide base of this band. 3) The sharp peak at 870 cm-1 is from the carbonates, where this band is always intense. 4) The doublet at 800-780 cm-1 corresponds to quartz, as the authors indicate at some point, and this doublet is always intense. 5) Fe oxyhydroxides frequently have wide bands that are masked by sharper bands produced by other phases. Certainly there is a band overlapping the 800-790 cm-1 doublet of quartz and this could be originated by Fe oxyhydroxide, but nothing of this type is observable near the carbonate band at 870 cm-1. 6) Comparison of the intensity of bands at 1624 and 3400 cm-1, suggests that the band at 1624 cm-1 corresponds entirely to hydration water, and that there is no contribution of carboxylic vibrations here.

Figure 11, caption: add "ATR" before FTIR in (a).

Description of results and discussion of FTIR data. The expressions "longwave" and "shortwave" are loose. Wavelength ranges should be used instead.

P. 30, lines 7-10. These lines should be deleted. First, the authors do not provide the values from Di Baggio et al (2014) so one cannot compare them with those in the ms. Second, absorbance depends of several variables (particle size, roughness, experimental conditions), none of which is specified. Thus, this text is too vague to be informative.

P. 30, lines 16-17: What does it mean? What is remarkable about two spectra from Saharan dust looking similar (if this is the meaning of the text)? Perhaps this text can be deleted.

P. 30, line 20. In an Fe oxide the ligand is O and the metal Fe, so the text could be more specific indicating "O-to-Fe charge transfer".

P. 32, lines 9-10: chlorite can be dioctahedral and trioctahedral.

Spelling, etc.

[Figure]

I believe that "thwart" is used mistakendly instead of "dwarf". P. 2, line 25. "Worldwide" would be a more clear word than "global" in this phrase. P. 7, line 27. I believe that the correct reference is Fig. 1b and 1c. P. 9, line 9-10. I think that it would be better to refer to the accuracy of the forecast. After all, the forecast is trying to reproduce the atmospheric events. P. 18, line 22. "matches". P. 23, line 8. I suggest to delete "in situ" as it is not necessary. P. 24, line 21. "clean" is probably not the word to use. P. 26, lines 28-end. This long phrase should be broken up for clarity. P. 30, line 6: delete "terrestrial", as unnecessary.

Javier Cuadros

---

## Referee Comment (RC2) · Anonymous Referee #2 · 2 Apr 2018

General comments: This manuscript reports the mineralogical, chemical, morphological characterisation, as well as number size-distribution and optical properties of Saharan dust collected in intense wet deposition events occurred in February 2017 in Granada. Finally, this characterisation is used to discuss the potential biogeochemical implication of dust deposition for supplying soluble iron, the direct radiative forcing of dust and the health impact of dust. The physico-chemical data about transported dust are rare and hence always precious information. However, some shortcuts on the experimental description may question the findings and conclusions of this paper.

The manuscript is generally well-written, logically organized, and adequately illustrated. Abstract is succinct and accurate.

The analytical work is widespread and careful. However, the discussion about data is often supported by rough approach (see specific comments) and some critical information about protocol are missing, making sometimes the conclusions of this paper unpersuasive:

For deposition sampling (p4 and 5), the used protocol is not clear since the authors talk about both wet and dry dust? : "Wet deposited dust was collected at 3 different locations in the urban area of Granada .. In all cases, dry dust was collected (scrapped) directly from clean horizontal surfaces and/or clean ceramic rain/dust collection dishes (21 cm in diameter) with the aid of a spatula and/or a brush. Do you mean that the dry insoluble residue deposited on dishes after wet deposition is collected for analysis? In this case, the dust is collected after evaporation of rain water and it is possible that the precipitation process of salts or amorphous phases happened during this evaporation step. For example, the presence of fibrous calcite could support a potential precipitation. It's critical to detail this step of sampling since a possible precipitation modify all your discussion about the presence of nano-sized ferrihydrite in the collected dust, which is one of finding of this paper. Indeed, the authors argue on a key finding: "a significant fraction of Fe in Saharan dust is already present in the source region as potentially bioavailable nanosized amorphous and/or poorly crystalline iron oxihydroxides which is Fe is present in dust region as very soluble ferrihydrite"(p33, L34).. If ferrihydrite is very soluble, why do the authors detect it in the particulate insoluble phase of wet dust deposition?

In the same way, if the sampling corresponds to insoluble dry residue of dust, it is important to specify that during the discussion on the chemical composition. Indeed, this implies a potential underestimation of soluble species as Ca, Mg (see Fu et al., 2017) but could also support such of your data for few soluble trace metals (e.g. Fe/Al ratio).

Moreover, this step of evaporation could also enable the formation of aggregates between deposited dust (formation of crust) and hence modify the distribution between clay, silt, and sand ... The collection with a spatula or a brush of dust present also a risk to loss of the finest particles (not collected or re-suspended by brush movement). It's probably not very important for the methods of mass characterisation, but it's more problematic for individual analysis as microscopic observations. For example, that could explain the presence of large particles ($>10\mu$m) during SEM observations. The authors cannot exclude the impact of this potential aggregation on their results even for mass analyses. For example, the question appears since the clay fraction found here is smaller than observed in the literature, even during intense local erosion event (56% in Formenti et al., 2014a) and has to be discussed.

Without a detailed discussion about these experimental points, the conclusions of this paper could be wrong.

Specific comments:

P2, L 30 to P3L5: the authors argue an increase of Saharan dust input and dust plume intrusions in Mediterranean on the basis of various phenomena (drought, changes land uses..). However, recent works seem to emphasize a decrease of Saharan dust deposition in western Mediterranean since the beginning of 2000's due to probably changes of atmospheric circulation (Pey et al., 2013; Vincent et al., 2016..). As the dust deposition is in the heart of this paper, I think that it's important to mention these recent results.

In general, I found that the bibliography about dust characterisation is a little bit poor. I suggest to include in your comparison with the literature the works on the dust characterisation carried out over Atlantic as e.g. Lazaro et al., 2008; Patey et al., 2015.

P5, L15: How the organic matter is affected by the different standard treatments? What is the impact on mass budget of minerals?

P5, L20: For the analysis, it's not clear if the dust collected on the different locations are mixed or separately analysed. Please precise.

Figure 4: Could you provide forward trajectories from PSA1?

P18, L2: The chemical composition is for the bulk samples? Please precise.

P18 from L4: The ratio Si/Al = 3.6 and Ca/Fe = 0.55 in your sample are consistent with dust from Bodélé depression (PSA5) on the basis of Formenti et al., 2014a. This assumption seems be confirmed by the observations of septa in your TEM analysis. The dust could be originated from a larger zone than only PSA1 and PSA3 (see also Figure 2 for the 22 Feb). Please add a comment about these observations.

P24, L7: "These results have important implications regarding the possible atmospheric acid processing of the iron-containing phases, as we will discuss below." The calcium nitrate and sulphate formed during atmospheric processing are highly soluble, so it is not surprising the authors did not observe these species onto dust issued from wet deposition sampling. So this result has no important implication. This is not also a good evidence of the "negligible mixing with air masses including anthropogenic-derived pollutants".

P27, L25: Finally, a budget of composition of bulk dust should be provided including carbonate, organic matter and various minerals to give a global view of mass composition.

P31, L16: Several recent works emphasized that the range of solubility of Fe-bearing dust is less than 1% to 80%. Even atmospheric processes increase the iron dust solubility, the highest observed values of solubility are related to the presence of anthropogenic iron (e.g. Sholkowitz et al., 2012). Please modify this part by including the most recent literature.

P32, L2: "The amorphous and/or poorly crystalline ferrihydrite would thus amount to 28 wt% of the free iron. Âż. This conclusion is based on very rough estimation! so

please detail the calculation (2.04% issued from calculation from Table 4?) and add the uncertainties on the values since on the basis of free iron = 1.7%±0.5% of goethite (containing 63% of Fe) + 0.5%±0.4% of hematite (containing 57% of Fe, see Journet et al., 2008) = 1.3% ±0.7%, so a total of iron content = 3.34% ±0.7%, that is in the range of total iron content=3.43%-3.69% found by chemical analysis, meaning that no iron is associated to ferrihydrite..

For part 3.8, 3.9 and 3.10: All the discussions need to be re-written by including the reservations in link with sampling protocol about the presence of nano-sized ferrihydrite and size-distribution (See general comments)

Minors corrections:

P2, L4: "large" instead of "enormous"

P4, L 20: 48h instead of 42h

P8, L15: $\mu$g m-3 instead of $\mu$m m-3

Figure 4: forward trajectory for the 21 Feb should be presented for 50 m instead 500 m.

Table 1: Please precise that the wt% is for "treated dust samples" without carbonate fraction

P18,L14: Formenti et al., 2014a instead 2014b

Table 3: please precise the uncertainties on measurements for ICP and XRF analysis.

References:

Fu, Y., Desboeufs, K., Vincent, J., Bon Nguyen, E., Laurent, B., Losno, R., and Dulac, F.: Estimating chemical composition of atmospheric deposition fluxes from mineral insoluble particles deposition collected in the western Mediterranean region, Atmos. Meas. Tech., 10, 4389-4401, https://doi.org/10.5194/amt-10-4389-2017, 2017.

Lazaro, F.J., Gutierrez, L., Barron, V., Gelado, M.D. The speciation of iron in desert dust collected in Gran Canaria (Canary Islands)/combined chemical, magnetic and optical analysis. Atmospheric Environment 42, 8987-8996, 2008.

Patey MD, EP Achterberg, MJ Rijkenberg, R Pearce - Aerosol time-series measurements over the tropical Northeast Atlantic Ocean: dust sources, elemental composition and mineralogy, Marine Chemistry, 2015.

Pey, J., Querol, X., Alastuey, A., Forastiere, F., and Stafoggia, M.: African dust outbreaks over the Mediterranean Basin during 2001–2011: PM10 concentrations, phenomenology and trends, and its relation with synoptic and mesoscale meteorology, Atmos. Chem. Phys., 13, 1395–1410, doi:10.5194/acp-13-1395-2013, 2013.

Vincent, J., Laurent, B., Losno, R., Bon Nguyen, E., Roullet, P., Sauvage, S., Chevaillier, S., Coddeville, P., Ouboulmane, N., di Sarra, A. G., Tovar-Sanchez, A., Sferlazzo, D., Massanet, A., Triquet, S., Morales Baquero, R., Fornier, M., Coursier, C., Desboeufs, K., Dulac, F., and Bergametti, G.: Variability of mineral dust deposition in the western Mediterranean basin and south-east of France, Atmos. Chem. Phys., 16, 8749-8766, 10.5194/acp-16-8749-2016, 2016.

---

## Author Comment (AC1) · 8 Jun 2018

Reply to Referee # 1 Authors' Response (AR) in blue (see attached pdf)

General assessment This ms. characterizes the mineralogy and chemistry of the Saharan dust deposited in Granada, Spain, in one extreme dust event. It also investigates the trajectory of the winds carrying the dust and the dust sources. These results are compared with abundant literature of Saharan dust transport, mineralogy and chemistry. All the above is a very complete study carried out with competence and thor-

oughness. Beyond this, the ms includes a lengthy discussion on three broad issues: Fe bioavailability in Saharan dust, effect of Saharan atmospheric dust on radiative forcing and effect on human health. These discussions are kept at a general, non-specific level and are given a relevance which is not warranted either by the data provided in the ms or the depth of the discussion. After all, the ms provides data from one event only, and this and extreme event, rare by definition. The ms does not contribute data with sufficient perspective to allow the wide discussion on the above issues. In my opinion, this discussion needs to be re-written in a focused way, more closely linked to the ms results and, if possible, using a quantitative approach that illustrates the magnitude of the relevance of the results. Such amended version of the ms would be within the scope of ACP. If the discussions on Fe bioavailability, effect on radiative forcing and effect on human health were removed the ms would have a much smaller scope, consisting on the detailed characterization of an extreme dust event. The text can be condensed. Some ideas are repeated several times in the ms.

Authors' response (AR): We thank the referee for the thorough review of our Ms and for his overall positive opinion about the completeness of our study. We also thank him for underlining the competence and thoroughness of our work. We appreciate his comments regarding the need for a more focused discussion. In the revised version of the manuscript, a more focused discussion avoiding repetition and underlining the limitations of our results to draw too general conclusions is presented. Additionally, all the technical issues indicated by the referee have been addressed (see below). Note: unless specified, all references cited in this AR are included in reference list of the revised version of the Ms.

Major issues Discussion of Fe bioavailability. A lot of this discussion reviews known facts and discusses them in a general way, without quantitative data. The significance of the results from the ms for Fe bioavailability is not sufficiently supported. In my opinion, the authors could try to assess the amount of bioavailable Fe in the investigated dust and compare it with quantitative results in the literature. They could also try to

project the results to marine areas seeded by the Sahara, comparing with previous estimates. Even if these projections have large uncertainty, they will allow some assessment of whether the results in the ms suggest changes to our understanding of input of bioavailable Fe to the ocean from the Sahara.

AR. We feel that due to the importance of the topic and the broad audience of Atmospheric Chemistry and Physics, the review of facts regarding dust Fe bioavailability and impact is necessary to follow the results that we present and discuss in this section. Following this referee's suggestion, in the revised version of the Ms. we include the results of the analysis of the bioavailable soluble Fe determined from standard leaching tests using both MilliQ water (pH 5.6) and an ammonium acetate buffer solution (pH 4.7). The protocol for such dissolution tests and analyses are now indicated in the Materials and Method section. We have discussed both our quantitative results on amorphous and/or poorly crystalline Fe content in the studied dust as well as our new soluble iron results in lieu of the existing literature. Based on this quantitative comparison with published results, we now emphasize that the amount of bioabailable Fe (from dissolution of -colloidal- amorphous of poorly crystalline ferrihydrite, as well as from dissolution of clays) already present in dust source areas may have been underestimated, particularly because in the case of the amorphous or poorly crystalline phase, it is difficult to identify and quantify. We also point out that the fractional iron solubility, FFS, of our samples is double than the value of amorphous/poorly crystalline Fe-oxihydroxides we have determined upon a careful revaluation of our TEM-AEM results, as well as the quantitative XRD results (see answer to Referee #2). This is interpreted considering that the rest of soluble iron comes from dissolved structural iron (i.e., from the Fe-containing clay minerals). Finally, we quantitatively compare our FFS results with published ones for the Atlantic and Mediterranean areas.

The authors state that the amount of soluble Fe typically increasing during long-range transport, from âĹij1% up to 80%. The simplest mechanism to explain this is that the larger particles drop during transport, causing a concentration of smaller, more soluble

particles. With this mechanism, the total amount of soluble Fe in dust does not increase with transported distance. Fe-solubility, as indicated by the authors (lines 7-10 on p. 31), is mainly a function of particle size.

AR. The idea of establishing a relationship between transport distance/time and preferential deposition of larger particles leading to an apparent increase in soluble iron, as originally proposed by Baker and Jickells (2006) (for the case of Saharan dust transported across the North Atlantic) has been discredited by subsequent studies. In a study by Shi et al (2011a), cited in the revised version of the Ms, where both Baker and Jickells are co-authors, it is clearly demonstrated that physical size sorting has a very minor (if any) effect on iron solubility: other effects such as atmospheric processing (including photo-oxidation and aerosol acid processing) and mixing with anthropogenic (Sholkovitz et al., 2009; 2012) and/or biomass (Guieu et al. 2005, Paris et al. 2010) combustion particles explain the observed increase in Fe solubility. This discussion is now incorporated in the corresponding section of our revised manuscript.

I do not know that there is evidence of bioavailability of Fe2+ or Fe3+ in clay minerals, unless they are nano-particles. Fe2+ is stable in non-expandable clay minerals, and there is virtually none of it in expandable clays. Ocean basins preserve a clay record which is generally consistent with clay mineralogy and composition in the continental sources, indicating low reactivity.

AR: Regarding the potential bioavailability of Fe2+ or Fe3+ in clay minerals, note that Journet et al. (2008) leaching tests clearly demonstrated that clay minerals were a main supplier of soluble iron. Others (Cwiertny et al., 2008; Schroth et al., 2009; ) also pointed out that Fe2+ in clay minerals such as chlorite, but also present in minor amounts in kaolinite, illite and smectites (as we stated with proper referencing in our Ms), was a direct source of soluble iron.

In their assessment of the amount of amorphous Fe phases in the Saharan dust, the authors do not discuss Fe in carbonates: is there any Fe in carbonates? How would

this modify the figure of "missing" 0.42 wt% Fe?

AR: Our TEM-AEM analysis of carbonates did not show any Fe. The absence of iron in carbonates is confirmed by the lack of any shifting in the 104 Bragg peaks. Note that replacement of Ca in calcite or in dolomite by Fe2+ (Fe3+ does not get incorporated in rhombohedral carbonates) results in a reduction in the unit cell, which shifts the Bragg peak position to higher 2ïĄś angles. In the revised version of the Ms we state that carbonates include no Fe, and in the Supplementary Material we now include an EDS spectrum of calcite in the studied Saharan dust showing no Fe (Fig. S6).

Discussion of radiative forcing. This discussion is non-specific and misses some important facts. First, the ms investigates a type of dust event which is not frequent. The amount of time that dust particles such as those described in the ms exist in the atmosphere could be negligible in comparison with the usual atmospheric situation. Thus, the relevance of these data for radiative forcing is questionable.

AR: We now emphasize that the studied extreme dust event is indeed a rare event, whose temporal extension is limited. So its impact on the direct radiative forcing has to be temporally-limited. However, we stress that despite is short time-span, the impact of recurrent extreme events such as the one studied here, as well as those taking place in the Mediterranean area in a more or less yearly basis over the last years (e.g., Avila et el., 2007; Cabello et al., 2012; Titos et al., 2017), can indeed have a significant impact on the atmospheric radiative budget not only regionally but, in the long-term, also even globally. We state so in the revised version of the Ms.

The authors indicate that small particles form aggregates with the large particles. This would mean that the small particles will settle quickly with the large ones and remain suspended only for the short time that the strong winds persist. Second, the authors have no knowledge of particle size distribution vertically in the atmosphere during the dust event; yet, this knowledge is essential to assess the effect on radiative forcing, as they indicate.

AR: We have already discussed that large (coarse and giant) particles can be transported for periods of time longer than 12 h under the strong wing conditions during the extreme event studied here. Regarding the vertical distribution of particle load and size, we have to state that due to turbulent mixing during this event -as well as other major events-, it is very unlikely that particles undergo any kind of size segregation with dust plume height. Indeed the lidar results reported by Fernandez et al (2018) for this event (over Granada) show that there is no significant vertical change in the Ånsgtröm exponent (meaning that no significant change in dust particle size occurs with altitude within the dust plume, which spans from ∼1500 m to 3500 m asl). See the image below:

Fig. 1 AR. Backscatter coefficient, $\beta$-Ångström exponent, particle and volume depolarization profiles at 12:30 UTC on 22 February 2017 at Granada (From Fernandez et al. (2018) Atmos. Chem. Phys. Discuss. DOI; 10.5194/acp-2018-370) (SEE ATTACHED Fig and/or PDF version of the AR)

It is stated that although atmospheric dust generates negative radiative forcing, "internal mixing" of minerals may lead to positive forcing. Why is this? This point is central to the entire discussion and it has to be explained. However, how do the authors know that the mineral particle aggregation that they observe was not produced during the rain that settled the particles? The cementation of mineral particles is mentioned but it is not clear where this knowledge originates. We are all familiar with the strong cohesive forces between mineral particles generated by wetting them and letting them dry; one example of this is the preparation of oriented aggregates for clay examination. If the internal mixing in the dust samples was a result of rain, the particles suspended in the atmosphere would have not been "internally mixed".

AR: The internal mixing of particles forming heterogeneous aggregates lead to enhanced absorption because there is an increase in the effective refractive index of the aggregate. This is explained in detail by Sokolik and Toon (1999), a study already cited in our Ms. To clarify this point we now state that: " While an external mixing of (alumino)silicates and iron oxyhydroxide particles in proportions typically found in desert dust has a net negative radiative forcing, their internal mixing can lead to a net positive radiative forcing due to a change in the effective refractive index resulting in enhanced absorption (Sokolik and Toon, 1999)." Regarding the mineral aggregation issue, please see also our response to Referee #2 on this specific issue. We are fully confident that our dust particles were already internally mixed in the atmosphere. Note that clay rich particles such as the Iberulites, which according to Cuadros et al. (2015) form through atmospheric processing and are "cemented" by the strong cohesive forces between (clay) mineral particles generated by wetting and subsequent drying, reportedly, they readily disaggregate when in contact with water (Fiol et al., 2005). Moreover, such aggregates formed due to atmospheric processing are spheroidal, in contrast to the elongated, irregularly shaped, sharp-edged aggregates we observed using FESEM. These points were already stated in our Ms. Moreover, we observed no disaggregation or change in the PSD following intense sonication in water during successive laser scattering analyses, demonstrating that the aggregates were very strongly cemented (i.e., already at the source area), and they did not aggregate during atmospheric processing or during drying after wet deposition. This is further clarified in the revised version of our Ms.

These considerations call for a concise discussion on radiative forcing. In such discussion the authors could point out some mineralogical facts from their samples that may not have been considered in models of dust-radiation interaction, leaving the question open of whether or not they are relevant to the issue.

AR: As indicated above, in the revised version of the Ms. we now emphasize that the extreme dust event is indeed a rare event, whose temporal extension is limited, so its impact on the direct radiative forcing has to be temporally-limited. However, we stress that despite is short time-span, the impact of recurrent extreme events such as the one studied here, as well as those affecting the Mediterranean area in a more or less yearly basis over the last years, can indeed have a significant impact on the

atmospheric radiative budget.

Health hazard. This discussion consists on a review of previous studies and an enumeration of aspects that have not been considered in health-mineral studies. Again, the main criticism here is that data from the ms may not be relevant to health issues because they refer to infrequent events. Presumably, people living around the Sahara take shelter during extreme storms. The situation relevant to health is that of habitual, more moderate winds. Instead of pointing out aspects that have not been considered in these studies in the past, it would be more helpful to identify one or two questions learned from this study that may be of interest for health issues, whether or not they prove to be relevant in the long run.

AR: Although this is a rare event, its health impact can be significant. Despite its short-time spam, intense dust-event have been demonstrated to have a direct impact of patient hospitalization and death rates (Perez et al. 2008). We thus stress that recurrent extreme events such as the one studied here and those taking place in a more or less yearly basis over the last years/decades, plus the continuous high dust loads in Northern Africa can indeed have a significant health short- and long-term impact. Unfortunately, desert dust-related health effects (e.g., "desert lung", see Middleton, 2017) and mortality peak in the areas around the Sahara (Wang et al., 2016), which is evidence that people living in these areas do not take all the necessary precautions. Regarding our results, we already made a strong case showing that the presence of abundant micrometer-sized palygorskite fibers associated with iron nanoparticles can have a strong health impact. That was the ultimate focus of our discussion in this section. Following this referee's suggestion, we conclude pointing out that the possible link between palygorskite-rich desert dust and long-term health effects should be the focus of further research.

At the end of the 1st paragraph in this section there is the assertion that the clay fraction of desert dust has the "most" harmful effects in the short term. This assertion seems to be founded on the studies of health effects of PM2.5. However, as there is

no comparison with the effects of other PM values, the conclusion that the clay fraction has the most harmful effects is not supported.

AR: There are numerous studies showing that PM10 has a direct short-term health effect (Perez et al., 2008; Karanasiou et al., 2012) and there are also studies that report that, in comparison, PM2.5 appears to have an even more important impact on short-term health effects and mortality (Giannadaki et al., 2014). Indeed, Zhang et al. (2016) (cited in our Ms.) stated "Overall, for respiratory and circulatory mortality, both positive and negative associations have been reported for PM10 of desert dust, but only a positive relationship was reported between PM2.5–10 and mortality, and a positive relationship was also reported between PM2.5 and human mortality". This was stated in the previous version of the Ms. We agree that the way we stated the possible link between the clay fraction and harmful effect was too conclusive (based on the existing evidence). In the revised version of the Ms. we toned down our statement by indicating: "These studies suggest that the clay fraction of desert dust can be the one that has the most deleterious effects in the short term."

In the three sections on relevance, the authors frequently assert that this or that issue or aspect has been "neglected" in previous studies. This is probably not the right word, as it carries a connotation of oversight or incompleteness in such studies. All three issues are very complex and the number of variables is beyond what any single study can consider. In this situation simplifications are necessary and the most relevant variables have to be selected. It would be preferable to indicate that certain analyses of dust composition would help to obtain more realistic results.

AR: We agree that the term "neglected" may have the connotation of incompleteness. Therefore, we have replaced "neglected" with "overlooked", "not considered" or "ignored".

Technical issues I am not qualified to comment on the investigation of wind trajectories. P. 13, line 18. Avila et al (1997) did not report goethite, hematite or rutile (?). Of course,

the amount of Fe and Ti oxide is small and can be missed in a mineralogical study. However, was not the dust red? If so, how did they account for this colour?

AR: We agree that the red color in "red rains" indirectly indicates the presence iron oxyhydroxide phases... but the point is that these authors did not actually report the presence of the above-mentioned specific Fe-containing phases (i.e., hematite, and goethite), as well as rutile.

Methods. As sample aliquots were treated in different ways, it would be helpful to indicate at the beginning of the description of each technique what treatment the corresponding aliquot had received.

AR: Done. In the Methods section, we now state the specific treatment done to samples analyzed with each technique.

Figure 4. The lower part of the panels, below the maps, is unreadable. Would there be expanded versions of the figures in the final production?

AR: High quality versions of the Figures will be submitted for final publication. In any case, in the pdf version of the Ms. (downloaded from the publisher website), the lower part of the panels in Figure 4 display small captions but they are readable.

Figure 5. There are several questions in this figure. Did the authors carry out background subtraction in the XRD patterns of panels c and d? See that the background is lower to the left of the low-angle peaks than to the right of these peaks (compare with the XRD patterns in panels a and b). Such procedure can deform the XRD pattern significantly in this area and blur the corresponding information. The goethite and hematite peaks are not indicated in panels c and d. It would be helpful to label panels c and d with "clay" and "silt" for an easier identification. The figure contains only information on âŮę2$\theta$, whereas the caption and text use Å: non-frequent users of XRD will find this difficult to follow.

AR: No background subtraction was performed to any of the XRD patterns presented

in the figure. The rise in the background at angles 20-30 °2ïĄś (best observed in panel (c), sample heated at 550 °C), is simply due to the presence of poorly crystalline and/or amorphous phases. This was already indicated in the Ms (page 13, first paragraph: "Note also that amorphous phases (e.g., amorphous silica -in diatoms- and amorphous iron oxyhydroxides, see TEM results below) can not be detected using XRD, and their content was not negligible as shown by the broad hump at 18−32 °2ïĄś in the XRD pattern (Fig. 5a).". No deformation of the XRD pattern took place. Regarding the goethite and hematite peaks, they are now indicated in the revised Fig. 5, where panel c and d are now labeled "Clay" and "Silt", respectively. Selected d-hkl values for the main peaks of each clay phase in the dust (discussed in the main text of the Ms.) are now included in Fig. 5c and d.

P. 15, lines 22-25. It is also possible that single sources contain kaolinite of two different crystal orders. The reasons for this can be multiple. Among them, wind and dust deposition form several areas within the Sahara.

AR: We now discuss this possibility too.

Table 2 is completely unclear. It needs to be thoroughly checked for the correctness of the labels and heading. For example, it seems to indicate that the proportion of smectite continually increases from the clay to the silt fraction and to the bulk, which would be remarkable. Actually, the text in p. 17, line 13, is incompatible with the results in Table 2.

AR: To clarify this point we have included the following in the heading of Table 2: "Note that the contents of individual clay minerals determined from XRD analysis of the bulk [powder] sample are subjected to a very high uncertainty. More reliable semiquantitative results are obtained from oriented aggregates [OA]." We also indicate that the Total Silt+Clay fractions row refers to results from oriented aggregates adding "[OA]". In the main text we now add: "Note also that because the mass of the silt fraction was 2.06 times larger than that of the clay fraction (results of the hydrodynamic size-separation:

sand=2 wt%; silt=68 wt% and clay=30 wt%) the amount of smectites, which along with MLC typically concentrate in the clay fraction, was relatively high in the silt fraction. However, if we consider the unit mass of silt and clay fractions, the amount of smectite in the former (11%) is lower than the amount of smectite plus MLC in the later (15%)."

Table 4 and p. 20, lines 32-34. The differentiation between smectite and illite-smectite (I-S, or the MLC, mixed-layered clays) is not supported. The authors ground this differentiation on tetrahedral substitution (of Si by Al). Typically, I-S minerals have more tetrahedral Al than smectite, but this is not the only or even most important difference. The central questions are that, in I-S, both the layer charge and K content are higher. Comparing the results for the analyses labelled "smectite" and "I-S" one can see that this is not the case: there are fewer interlayer cations in I-S (this should mean that the layer charge is lower, if the formulas are correct, which I have not ckecked), and the amount of K is basically the same. According to these data, it is not possible to discriminate between these two groups of analyses as corresponding to smectite and I-S. The results indicate, perhaps, smectite with a wide range of tetrahedral substitution. The authors mention a beidellitic component in smectite. Beidellite typically has more tetrahedral Al than montmorillonite.

AR: We agree with the referee. We reevaluated every single MLC and smectite analysis (reported in Table S1) and realized that indeed some of the analyses did not comply with a proper MLC, but rather with smectite (and the opposite for the case of some smectite analyses, which were re-assigned as MLC). We have changed Table S1 accordingly and in Table 4 we now report the corrected average structural formulas for both smectite and MLC. A beidellitic component was considered because the Al content in tetrahedral positions was relatively high in some analyses.

P. 22, lines 1-3.The authors can check their data for a possible contamination of clay minerals with Fe-oxides. Structural formulas of uncontaminated dioctahedral clay minerals should fulfil the following conditions: Sum of tetrahedral cations = 4 (2:1 minerals) or 2 (kaolinite); the sum of octahedral cations is in the range 1.98-2.2; the interlayer

charge calculated from the lattice cations is equal to that calculated from the interlayer cations (within perhaps ±0.01); the interlayer charge is within the known range for the mineral suggested by the formula. The authors could include the above checks in Table 4.

AR: We checked every single AEM analysis and structural formulae reported in Table S1. Following the suggestion by the referee, and upon a careful analysis of the literature (crystal-chemistry of clay minerals) we have discarded some of the analyses and corresponding structural formulae of palygorskite, smectite and MLC (plus the one from kaolinite already indicated in the previous version of the Ms) either because they include anomalously high Fe contents and/or because they did not fulfilled the conditions listed by the referee. This is stated in the footnote of the revised Table S1. Only analyses that were considered reliable (uncontaminated and fitting compositional ranges for the specific clay minerals) were used to calculate the average structural formulae reported in the revised Table 4. Note that this re-evaluation of our AEM results has resulted in a slightly different value of structural Fe, as well as in the Fe content of palygorskite, smectite and MLC, that have affected the value of calculated "missing iron" (i.e., nanosized Fe-oxyhydroxides). The corrected values are now reported in the revised version of the Ms.

P. 24, top. This is the first time that the authors refer to internal and external mineral mixing, and the meaning should be provided here. The explanation should provide the basis for the relevance that is attached to this difference in further discussion, which is not clear. In my opinion, this is an important point that requires complete clarity in the description of the physical differences between one and the other type of mixing.

AR: To clarify this point, the first time internal/external mixing is mentioned, we now state: "Note that dust aerosols can be internally or externally mixed. Internal mixing refers to aggregates formed by different mineral particles, while an external mixing involves different mineral particles existing separately." P. 24, line 17: "well-cemented". This is not an observation from FESEM, obviously. Which technique provided this
observation?

AR: We already stated that " They were made up of an aggregate of smaller, well-cemented (i.e., the aggregates were intact after sonication), micrometer-sized particles (Fig. 9b)." Sonication showed no dissagregation of the large particles, thereby demonstrating that they were well-cemented.

P. 25, lines 8-10. "The samples studied": with what technique? "

AR: With laser scattering (see Materials and Methods). In the revised version of the Ms. we state so.

In agreement with the results from the hydrodynamic size-separation": what were the data?

AR: The results of the hydrodynamic size-separation data (i.e., "(results of the hydrodynamic size-separation: sand=2 wt%; silt=68 wt% and clay=30 wt%)") were already presented in section 3.3. when reporting and discussing the results of the XRD analysis of clays.

Figure 9. The inset spectrum in panel b cannot be read.

AR: We have added labels to each peak so the elemental composition can easily be visualized.

P. 26, lines 2-3. This assertion needs to be supported with a reference or an explanation.

AR: To support this assertion we have cited the classic paper by Sing et al. (1985) where the classification and origins of the shape of gas sorption isotherms era presented.

P. 26, lines 6-9. Is this correct? Why nano-pores should render the dust more reactive than mesopores? Is it related to the area of mineral particles exposed to water/atmosphere? This point needs developing.

[Figure]

AR: According to the International Union of Pure and Applied Chemistry (IUPAC) nanopores in the 5 nm size range are actually mesopores (mesopores are defined as pores with diameter between 2 and 50 nm). Indeed, such mesopores would strongly contribute to the exposed surface area of individual particles and/or aggregates, making them more reactive to atmospheric processing. In the revised version of the manuscript we state: "The presence of relatively abundant (nano)pores, an aspect of desert dust that has been typically neglected, is of relevance due to their effect on the dust's reactivity, and hygroscopicity because they would strongly contribute to the exposed surface area of aggregates, making them more reactive to atmospheric processing. Such pores could also affect the dust radiative properties (Kemppinnen et al., 2015)."

P. 26, last paragraph. Observing the DSC curves in Figure 10, it appears that the sharp peak corresponding to carbonate decomposition starts a âĹij700 âŮęC and ends some degrees above 800 âŮęC. The slope of the DSC curves between 600 and 700 âŮęC is less pronounced and probably due to dehydroxylation events. Also, water is very frequently lost for a range of temperature after the dehydroxylation temperature in clays. Perhaps the authors could recalculate the weight loss in the narrower temperature range indicated above to test whether the resulting carbonate content is then coherent with those of XRD and weight loss after acid treatment.

AR: We fully agree with the referee. The DSC curve clearly shows that decarbonation starts around 700 °C. Our previous studies showed that at a very slow heating rate of 2 to 5 °C/min, decarbonation of calcite starts at $\sim$600 °C, but at higher heating rates $\geq$ 15-20 °C/min, decarbonation starts at $\sim$700 °C (Rodriguez-Navarro et al., 2009). We have recalculated the carbonate content considering the weight loss in the T interval 700-900 °C. The actual carbonate content is 14.7$\pm$0.2 wt% (in excellent agreement with the carbonate content determined by XRD and acid decarbonation measurements). This is now stated in the revised version of the Ms.

Description of IR results in pp. 28 and ff. This description has more detail than can be

appreciated in Figure 11, and thus the authors could consider shortening it. Several IR bands referred to in the text are not labelled in the figure. There are several band assignments that need reconsideration or rewriting for a more accurate description. 1) Most, if not all, the IR absorption below 3500 cm-1 (excepting the organic bands) is due to hydration water, so it would be more accurate to say that the band centered at 3400 cm-1 corresponds to water. 2) The sharp bands at 3893 and 3617 cm-1 are from kaolinite; the illite and smectite bands at 3650-3550 cm-1 (depending on composition) are always wider and they can be seen overlapping the 3617 cm-1 band, causing the wide base of this band. 3) The sharp peak at 870 cm-1 is from the carbonates, where this band is always intense. 4) The doublet at 800-780 cm-1 corresponds to quartz, as the authors indicate at some point, and this doublet is always intense. 5) Fe oxyhydroxides frequently have wide bands that are masked by sharper bands produced by other phases. Certainly there is a band overlapping the 800-790 cm-1 doublet of quartz and this could be originated by Fe oxyhydroxide, but nothing of this type is observable near the carbonate band at 870 cm-1. 6) Comparison of the intensity of bands at 1624 and 3400 cm-1, suggests that the band at 1624 cm-1 corresponds entirely to hydration water, and that there is no contribution of carboxylic vibrations here.

AR: Missing bands are now labeled in the revised Fig. 11a. In addition, we have considered the suggestions for band assignment: (1) we now state that " The broad band centered at ∼3400 cm-1 corresponded to $\upsilon$OH stretching of interlayer (solvation) H2O of clay minerals, and structural water in ferrihydrite (Russell, 1979)"; (2) we agree that it is hard to resolve the smectite and illite OH bending and state so in the revised Ms; (3) Similarly, we agree that the 870 cm-1 band also had the contribution of carbonates (we state so in the revised version of the Ms). Note, however, that this band is much more intense than the most intense band of carbonates at 1406 cm-1: it follows that the band at 870 cm-1 has a strong contribution of other phase(s), that is Fe oxyhydroxides (as we stated in the Ms); (4-5) We agree that the doublet at 800-780 cm-1 has a strong contribution by quartz, but the band at 780 cm-1 also has the contribution of

OH in ferrihydrite, as we stated in our Ms; (6) It is very difficult to make any quantitative evaluation/comparison between the very broad band of water at ∼3400 cm-1 and the OH band at around 1624 cm-1. The absorbance of the OH band, which should be centered at 1644 cm-1 (we state so in the revised version of the Ms), is typically half of the absorbance of the 3400 cm-1 band. In our case the band at 1624 cm-1 is more intense. This is why we stated that in addition to the contribution of the OH bending, this band is also associated with carboxylic groups in organic matter. Note that the bands at 2890-2990 cm-1 confirm the presence of organic matter (in agreement with TG-DSC results).

Figure 11, caption: add "ATR" before FTIR in (a).

AR: Done

Description of results and discussion of FTIR data. The expressions "longwave" and "shortwave" are loose. Wavelength ranges should be used instead.

AR: Done.

P. 30, lines 7-10. These lines should be deleted. First, the authors do not provide the values from Di Baggio et al (2014) so one cannot compare them with those in the ms. Second, absorbance depends of several variables (particle size, roughness, experimental conditions), none of which is specified. Thus, this text is too vague to be informative.

AR: Done (the lines have been deleted)

P. 30, lines 16-17: What does it mean? What is remarkable about two spectra from Saharan dust looking similar (if this is the meaning of the text)? Perhaps this text can be deleted.

AR: With this sentence we wanted to stress that our experimental UV-Vis spectra were very similar to the wavelength-resolved values of the imaginary part of the refractive index (k-values) of actual Saharan dust samples. Note that measurements of UV-

Vis absorbance and k-values are determined using very different experimental and calculation protocols.

P. 30, line 20. In an Fe oxide the ligand is O and the metal Fe, so the text could be more specific indicating "O-to-Fe charge transfer".

AR: We now state: ". . . with ligand-to-metal (i.e., O-to-Fe) charge transfer transitions (Sherman and Waite, 1985)."

P. 32, lines 9-10: chlorite can be dioctahedral and trioctahedral. AR: We now state: "In trioctahedral chlorite iron is incorporated as Fe2+ substituting Mg2+ in octahedral positions both in the 2:1 structural unit and in the interlayer brucite layers (Weaver and Pollard, 1973)."

Spelling, etc. I believe that "thwart" is used mistakendly instead of "dwarf". P. 2, line 25. "Worldwide" would be a more clear word than "global" in this phrase. P. 7, line 27. I believe that the correct reference is Fig. 1b and 1c. P. 9, line 9-10. I think that it would be better to refer to the accuracy of the forecast. After all, the forecast is trying to reproduce the atmospheric events. P. 18, line 22. "matches". P. 23, line 8. I suggest to delete "in situ" as it is not necessary. P. 24, line 21. "clean" is probably not the word to use. P. 26, lines 28-end. This long phrase should be broken up for clarity. P. 30, line 6: delete "terrestrial", as unnecessary.

AR: The suggested corrections has been performed (except the case of "terrestrial" because we consider that this term clarifies the type of radiation we are referring to). Javier Cuadros

Please also note the supplement to this comment:
https://www.atmos-chem-phys-discuss.net/acp-2018-211/acp-2018-211-AC1-supplement.pdf
* * *
[Figure]

2018.

[Figure]

**Fig. 1.** Backscatter coefficient, $\beta$-Ångström exponent, particle and volume depolarization profiles at 12:30 UTC on 22 February 2017 at Granada (From Fernandez et al. (2018) Atmos. Chem. Phys. Discuss. DOI; 10.

---

## Author Comment (AC2) · 8 Jun 2018

Response to Referee # 2

(Authors' Response, AR in blue) (see attached pdf version in color)

Anonymous Referee #2 General comments: This manuscript reports the mineralogical, chemical, morphological characterisation, as well as number size-distribution and optical properties of Saharan dust collected in intense wet deposition events occurred in February 2017 in Granada. Finally, this

characterisation is used to discuss the potential biogeochemical implication of dust deposition for supplying soluble iron, the direct radiative forcing of dust and the health impact of dust. The physico-chemical data about transported dust are rare and hence always precious information. However, some shortcuts on the experimental description may question the findings and conclusions of this paper. The manuscript is generally well-written, logically organized, and adequately illustrated. Abstract is succinct and accurate. The analytical work is widespread and careful. However, the discussion about data is often supported by rough approach (see specific comments) and some critical information about protocol are missing, making sometimes the conclusions of this paper unpersuasive:

Authors' Response (AR): We thank this referee for his/her overall positive opinion on our study. We also thank this referee for the very detailed and careful review of our study and for pointing out issues that needed to be addressed and for his/her suggestions regarding how the overall quality of the manuscript can be improved. All of them have been addressed. Our answers are presented below (after each comment).

For deposition sampling (p4 and 5), the used protocol is not clear since the authors talk about both wet and dry dust? : "Wet deposited dust was collected at 3 different locations in the urban area of Granada .. In all cases, dry dust was collected (scrapped) directly from clean horizontal surfaces and/or clean ceramic rain/dust collection dishes (21 cm in diameter) with the aid of a spatula and/or a brush. Do you mean that the dry insoluble residue deposited on dishes after wet deposition is collected for analysis? In this case, the dust is collected after evaporation of rain water and it is possible that the precipitation process of salts or amorphous phases happened during this evaporation step. For example, the presence of fibrous calcite could support a potential precipitation. It's critical to detail this step of sampling since a possible precipitation modify all your discussion about the presence of nano-sized ferrihydrite in the collected dust, which is one of finding of this paper. Indeed, the authors argue on a key finding: "a significant fraction of Fe in Saharan dust is already present in the source region as potentially bioavailable nanosized amorphous and/or poorly crystalline iron oxihydroxides which is Fe is present in dust region as very soluble ferrihydrite"(p33, L34).. If ferrihydrite is very soluble, why do the authors detect it in the particulate insoluble phase of wet dust deposition?

AR. We apologize for the confusion that our poor description of the sampling procedure has created. Actually, we collected the dry residue remaining after the dust dispersed in rainwater (i.e., red rain) was deposited in the collection devices and the rainwater subsequently evaporated (in about 12 h after wet deposition). In no case "wet" samples were collected. Because full evaporation of the wet deposited dust was allowed without any possibility of loss (leaching) of the soluble residue, the finally collected dry dust included both the insoluble and the reprecipitated soluble residue. This is now clarified in the Materials and Methods section of the revised version of the Ms. We are aware that dissolution of a fraction of the most soluble phases (e.g., carbonates, sulfates and nitrates -if present-, as well as amorphous/poorly crystalline Fe-oxyhydroxides) should have occurred when in contact with rainwater. And following drying, such soluble fraction had to reprecipitate. On the one hand, our results show that no sulfates or nitrates were present in the dry residue (again, note that no lixiviation of the aqueous solution took place during sample collection). If such soluble phases would have been present in the original dust, they would be partially dissolved in the rainwater (until saturation) and reprecipitated after drying, and would therefore have been detected. On the other hand, we now present the results of a geochemical modeling using PHREEQC computer code showing quantitatively the fraction of carbonates and iron-oxyhydroxides (goethite, hematite and ferryhydrite), as well as the rest of silicate phases identified, that dissolved in the very small amount of rainwater that fell during the wet deposition event (2 mm, i.e. 2 L per square meter). For this task we considered rainwater with a starting pH of 5.6, at equilibrium with 399 ppm atmospheric $CO_2$, reaching equilibrium at pH 8 -computed by PHREEQC and experimentally measured after dissolution of Saharan dust in MilliQ water with starting pH 5.6 -in a wt/vol ratio of 18g dust/2L water- and equilibrated with atmospheric $CO_2$ at room T. For the case of the carbonates, the amount dissolved in rainwater is 2.3 wt% of the total amount of calcite (no dolomite dissolved because the Ca and carbonate released upon calcite dissolution plus the Mg released following dissolution of a fraction of Mg-containing palygorskite led to saturation with respect to this carbonate). This amount is consistent with the actual amount of dissolved Ca determined by an ion selective electrode following dissolution of the dust in MillyQ water (starting pH 5.6; solid/water ration of 18g/2L) and reaching equilibrium (pH 8). Such a small amount of calcite dissolved and later on reprecipitated during drying cannot be responsible for the formation of the abundant fibrous calcite; the most likely fate of such dissolved amount of carbonate was a limited regrowth of the exiting calcite crystals, because the nucleation of a new carbonate phase would be energetically unfavorable. The same applies for the case of the Fe-oxyhydroxides: PHREEQC simulations considering only the presence of goethite and hematites showed a negligible $3 \times 10^{-8}$ wt% and $9 \times 10^{-9}$ wt% dissolution for the former and later phase, respectively. Simulations considering also the presence of ferryhydrite in addition to goethite and hematite showed no dissolution of the last two phases, and a 0.31 wt% dissolution of the total amount of ferrihydrite. The PHREEQC modeling clearly shows that the amount of dissolved free Fe is almost negligible. Such a solubilized iron cannot account for the amount of amorphous and/or poorly crystalline Fe-oxyhydroxides (ferrihydrite) determined here. Our new modeling and calculations, as well as the results of the new dissolution test, are now presented and discussed in the revised version of the Ms.

In the same way, if the sampling corresponds to insoluble dry residue of dust, it is important to specify that during the discussion on the chemical composition. Indeed, this implies a potential underestimation of soluble species as Ca, Mg (see Fu et al., 2017) but could also support such of your data for few soluble trace metals (e.g. Fe/Al ratio).

AR. As indicated above, our sampling did not correspond to just the insoluble residue but the combination of the soluble and insoluble residues (i.e., no separation or leaching of the soluble residue took place).

Moreover, this step of evaporation could also enable the formation of aggregates between deposited dust (formation of crust) and hence modify the distribution between clay, silt, and sand . . . The collection with a spatula or a brush of dust present also a risk to loss of the finest particles (not collected or re-suspended by brush movement).

AR. As indicated above (and stated in the revised version of the Ms) the amount of dissolved (sparingly) soluble phases (i.e., carbonates) that later on reprecipitated during the drying step of the wet deposited dust is very minor. Considering that 2.3 wt% of the total amount of calcite was dissolved and this phase represented only 12 wt% of the bulk dust sample, this means that only about 0.27 wt % of the bulk sample was dissolved and reprecipitated. Such a small amount should not have any significant cementing or coarsening effect on the bulk sample, particularly if one consider that the dissolved carbonate and Ca ions would subsequently be incorporated on the existing calcite crystals contributing to their regrowth. Their reprecipitation as new calcite crystals possibly cementing other dust particles such as clays or tectosilicates is not energetically favorable (simple calcite regrowth is) and it is not consistent with our TEM observations showing individual fibrous calcite crystals not contributing to the cementation of dust particles. Dissolution-regrowth of calcite in the amounts determined here could not be affective for any crust formation leading to an irreversible post-depositional coarsening of the aggregates. On the other hand, regarding the potential selective loss of a fraction of the smaller particles in Sahara dust during the collection stage, we have to state that dust collection was performed with extreme care in order avoid resuspension in the air of the finer particles. Indeed, we observed no dust resuspension during our sample collection. We clarify this point in the Materials and Methods section. The possible causes for the coarseness of the studied Sahara dust (including a possible aggregation during drying of the wet deposited dust) were already discussed in the first version of the Ms, and are further clarified in this revised version (in lieu of the PHREEQC modeling and dissolution experiments, as well as in lieu of the absence

of changes in the particle size distribution, PSD, following successive laser scattering analyses of dust particles dispersed in water and subjected to intense sonication during PSD analysis).

It's probably not very important for the methods of mass characterisation, but it's more problematic for individual analysis as microscopic observations. For example,that could explain the presence of large particles (>10$\mu$m) during SEM observations.

AR. See our answers above and our thorough discussion on the textural features that led us to rule out the possibility of coarsening during wet scavenging and subsequent drying of the wet deposited dust (particularly, the lack of spherical aggregates -iberulites-, and the angular shape of the coarse and giant aggregates).

The authors cannot exclude the impact of this potential aggregation on their results even for mass analyses. For example, the question appears since the clay fraction found here is smaller than observed in the literature, even during intense local erosion event (56% in Formenti et al., 2014a) and has to be discussed.

AR. Note that we already stated that due to the extreme intensity of the studied dust event, mobilization of coarser particles took place. Such coarser particles are typically enriched in tectosilicates and carbonates. This helps to explain why the clay minerals in our sample are 47 %, a value which is not so dissimilar to the 56 % reported by Formenti et al., (2014), or the value of 52 % reported by Patey et al. (2015). Moreover, it is well within the range 30-66 wt% reported by the later authors. We state so in the revised version of the Ms.

Without a detailed discussion about these experimental points, the conclusions of this paper could be wrong.

AR. We fully agree and thank again this referee for pointing out these critical issues. See our responses above regarding how in the revised version of the Ms we have clarified the sampling procedure and the related issues.

Specific comments:

P2, L 30 to P3L5: the authors argue an increase of Saharan dust input and dust plume intrusions in Mediterranean on the basis of various phenomena (drought, changes land uses..). However, recent works seem to emphasize a decrease of Saharan dust deposition in western Mediterranean since the beginning of 2000's due to probably changes of atmospheric circulation (Pey et al., 2013; Vincent et al., 2016..). As the dust deposition is in the heart of this paper, I think that it's important to mention these recent results.

AR. We agree. Indeed, there are indications that changes in atmospheric circulation have an impact on the strength of Saharan dust deposition in the Western Mediterranean, especially those related with a negative summer NAO which led to a recent (2001-2011) decrease in Saharan deposition rates. This is now stated in the Introduction of the revised Ms along with the suggested references.

In general, I found that the bibliography about dust characterisation is a little bit poor. I suggest to include in your comparison with the literature the works on the dust characterisation carried out over Atlantic as e.g. Lazaro et al., 2008; Patey et al., 2015.

AR. In the revised version of the Ms we have included the suggested references on the characterization of Saharan dust over the Atlantic. We want to note, however, that the focus of our research is on an event affecting the Western Mediterranean. This is why we paid special attention to studies reporting on the characterization of desert dust affecting this area.

P5, L15: How the organic matter is affected by the different standard treatments? What is the impact on mass budget of minerals?

AR. The organic matter was analyzed by TG/DSC using the bulk dust samples that were not subjected to any pretreatment. This was explicitly stated when describing the protocol for TG/DSC analyses. The impact of the organic matter (OM) content the

mass budget of minerals is very minor because the mass of OM determined form TG results is ∼0.6 to 1.1 Wt %. This means that if one would consider the OM content to recalculate the mass budget of major and minor mineral phases, only a change in the first decimal would occur (and such a change would be within the standard deviation of the values determined here).

P5, L20: For the analysis, it's not clear if the dust collected on the different locations are mixed or separately analysed. Please precise.

AR. We performed preliminary XRD analyses of each individual sample, observing that no differences among them in terms of phase composition and amount existed. Considering these preliminary results and in order to get a more representative and sufficiently large amount of dust for each of the analysis performed, the three samples collected at the three different sites were mixed prior to the different analyses performed. This is now stated in the revised version of the Ms (Materials and Methods section).

Figure 4: Could you provide forward trajectories from PSA1?

AR. Done. In the same figure (Figure 4) we also present forward trajectories for a Bodélé point source (PSA5)

P18, L2: The chemical composition is for the bulk samples? Please precise.

AR. Yes, the chemical composition refers to the bulk samples. We clarify this in the revised version of the Ms.

P18 from L4: The ratio Si/Al = 3.6 and Ca/Fe = 0.55 in your sample are consistent with dust from Bodélé depression (PSA5) on the basis of Formenti et al., 2014a. This assumption seems be confirmed by the observations of septa in your TEM analysis.

AR. We performed additional forward trajectory analysis with a source located in the Bodélé depression. The results (see new Figure 4f) show that no dust originating from the Bodélé reached the Iberian Peninsula during the studied event. We now discuss

so in the revised version of the Ms. Note also that our Ca/Fe ratio is not 0.55 but 1.55., a value to high for a Bodélé provenance. Moreover, the presence of diatoms (i.e., the amorphous silica particles displaying septa observed using TEM), which indeed are abundant in dust entrained at the Bodélé, have been also reported for the case of dust transported from other North African dust source areas. This was already discussed in the previous version of the Ms.

The dust could be originated from a larger zone than only PSA1 and PSA3 (see also Figure 2 for the 22 Feb). Please add a comment about these observations.

AR. We agree: most likely the dust was originated in a larger area, which included PSA1 and PSA3 as main source sub-areas. We indicate so in the revised version of the Ms.

P24, L7: "These results have important implications regarding the possible atmospheric acid processing of the iron-containing phases, as we will discuss below." The calcium nitrate and sulphate formed during atmospheric processing are highly soluble, so it is not surprising the authors did not observe these species onto dust issued from wet deposition sampling. So this result has no important implication. This is not also a good evidence of the "negligible mixing with air masses including anthropogenic derived pollutants".

AR. See our answers above clarifying that the wet deposited dust (i.e., dust plus rainwater) was let to dry and subsequently collected. No leaching of the soluble fraction took place during the process, so no loss of the soluble phases (sulfates and nitrates) would have taken place if these phases were present in the deposited dust. Our observation that no nitrates or sulfates were present has indeed important implications and support our contention that negligible mixing with polluted air masses took place. Such a contention is supported by the Pb/Al ratios and Pb enrichment factors as well as by the V/Al and Ni/Al ratios (now reported). In the revised version we state: "In addition, V and Ni, which are typically enriched in anthropogenic combustion aerosols (Sholkovitz

et al., 2009), show a V/Al ratio of 0.1 x 10-2 and a Ni/Al ratio of 0.06 x 10-2. These values are very similar to the corresponding values of the continental crust (V/Al=0.08 x10-2 and Ni/Al=0.05 x 10-2) (Taylor and McLennan, 1985; Sholkovitz et al., 2009)."

P27, L25: Finally, a budget of composition of bulk dust should be provided including carbonate, organic matter and various minerals to give a global view of mass composition.

AR. As stated above, and within the context of TG/DSC results for carbonate phases and OM, we want to emphasize that (i) the impact of the organic matter (OM) content the mass budget of minerals is very minor because the mass of OM determined form TG results is ~0.6 to 1.1 Wt % and (ii) the recalculation of the carbonate content following the suggestion of referee #1 shows that the content determined using TG (14.7 wt%) is almost identical to that determined using XRD (RIR) (14 wt%). This means that the mass composition (mineral phases) presented in Table 1, would not change (other than the first decimal) if one would include the OM or the TG-derived carbonate contents in a new table. We believe that adding another table with the overall mass composition that almost repeat Table 1, is not necessary.

P31, L16: Several recent works emphasized that the range of solubility of Fe-bearing dust is less than 1% to 80%. Even atmospheric processes increase the iron dust solubility, the highest observed values of solubility are related to the presence of anthropogenic iron (e.g. Sholkowitz et al., 2012). Please modify this part by including the most recent literature.

AR. We fully agree. We have modified this statement considering the results of recent works that emphasize that indeed the most soluble Fe fraction is related to anthropogenic iron. We also indicate that another source of soluble iron is biomass burning (Guieu et al., 2005). We now cite the papers by Sholkovitz et al. (2009, 2012).

P32, L2: "The amorphous and/or poorly crystalline ferrihydrite would thus amount to 28 wt% of the free iron. Âz. This conclusion is based on very rough estimation! so

please detail the calculation (2.04% issued from calculation from Table 4?) and add the uncertainties on the values since on the basis of free iron = 1.7%±0.5% of goethite (containing 63% of Fe) + 0.5%±0.4% of hematite (containing 57% of Fe, see Journet et al., 2008) = 1.3% ±0.7%, so a total of iron content = 3.34% ±0.7% , that is in the range of total iron content=3.43%-3.69% found by chemical analysis, meaning that no iron is associated to ferrihydrite..

AR. In supplementary material we now present a table with all the input values and results used for the calculation of the Fe content in the different phases. We also explain point-by-point how the calculations to obtain soluble free iron were performed. We have to state that a complete reavaluation of our AEM analyses for the clay minerals has resulted in a more acurate quantification of structural iron. The new value (1.98 wt%) is now used for the calculation of "missing iron". We also made a mistake when calculating the Fe content of goethite plus hematite in our samples (note that the Fe content of hematite reported by Journet et al. 2008, is not correct: the Fe content in hematite is 70 %). This corrected value (1.41 wt%) is now used in our calculations of the "missing iron". Considering these corrected values, we obtain that the "missing iron", that is the amount of amorphous and/or poorly crystalline Fe-oxyhydroxides (ferrihydrite) is 1.3-8.2 % of the total iron (the range of values correspond to the range of total iron determined by ICP-OES and FRX). These results are included in the revised version of the Ms, along with a detailed discussion regarding the implications of these results. Moreover, due to the uncertainty and range of variation of the Fe content determined from AEM and XRD results, we now state "It should be noted that these values, calculated based on the amount of iron in each Fe-containing phase determined by XRD and the Fe content in clay minerals determined from TEM-AEM analyses, are subjected to significant uncertainty. Therefore, the amount of structural and free iron, including the amount of amorphous and/or poorly crystalline Fe-oxyhydroxides (ferrihydrite) presented above are not intended to be purely quantitative results, but a (rough) estimate of the Fe speciation in the studied dust. Nonetheless, these results confirm the TEM observations showing that the amount of nanosized amorphous and/or poorly

crystalline ferrihydrite is not negligible."

For part 3.8, 3.9 and 3.10: All the discussions need to be re-written by including the reservations in link with sampling protocol about the presence of nano-sized ferrihydrite and size-distribution (See general comments)

AR. Regarding the sampling protocol and its possible effect on the formation of ferryhidrite, see our answer above. Note also that we have now added and discussed the results of a leaching test (performed using two solutions, one with pH 4.7 and another with pH 5.6, aimed at determining (using ICP-OES) the actual value of soluble Fe (fractional iron solubility, FFS), which corresponds to amorphous/poorly crystalline ferrihydrite plus Fe coming from the dissolution of clay minerals. The new data which are presented and discussed support our contention that the contribution of amorphous and/or poorly crystalline ferrihydrite to the FFS is substantial. The new data also show that an additional pool of soluble iron is necessary to account for the ∼20% FFS. In the revised version of the Ms we discuss that such an additional soluble iron comes from the dissolution of clay minerals. The new FFS results enable us to discuss our results in lieu of existing published results for FFS measured in the Mediterranean and North Atlantic areas. Regarding the size-distribution, in section 3.9 we now further elaborate on the possibility that coarsening might be associated with wet scavenging/deposition and the subsequent drying prior to sampling, showing that coarsening during wet scavenging or during drying of the red rain once deposited, is very unlikely.

Minors corrections:

P2, L4: "large" instead of "enormous"

AR. Done

P4, L 20: 48h instead of 42h

AR. Done

P8, L15: $\mu$g m-3 instead of $\mu$m m-3

AR. Done

Figure 4: forward trajectory for the 21 Feb should be presented for 50 m instead 500 m.

AR. Done

Table 1: Please precise that the wt% is for "treated dust samples" without carbonate fraction

AR. We already did so: in the bottom of the Table where we stated "This size-fraction powder sample was subjected to carbonate elimination"

P18,L14: Formenti et al., 2014a instead 2014b

AR. Done

Table 3: please precise the uncertainties on measurements for ICP and XRF analysis.

AR. The uncertainties of the ICP and XRF analyses were already indicated in section 2. Methods, subsection 2.3. where we stated: For the case of ICP-OES " The instrumental error is $\pm 2\%$ and $\pm 5\%$ for elemental concentrations of 50 ppm and 5 ppm respectively." and for the case of XRF "The quality of the analysis was monitored with reference materials showing high precision with $1\sigma$ = 1.0–3.4% on 16 data-sets at the 95% confidence level."

References: Fu, Y., Desboeufs, K., Vincent, J., Bon Nguyen, E., Laurent, B., Losno, R., and Dulac, F.: Estimating chemical composition of atmospheric deposition fluxes from mineral insoluble particles deposition collected in the western Mediterranean region, Atmos. Meas. Tech., 10, 4389-4401, https://doi.org/10.5194/amt-10-4389-2017, 2017. Lazaro, F.J., Gutierrez, L., Barron, V., Gelado, M.D. The speciation of iron in desert dust collected in Gran Canaria (Canary Islands)/combined chemical, magnetic and optical analysis. Atmospheric Environment 42, 8987-8996, 2008. Patey MD, EP Achterberg, MJ Rijkenberg, R Pearce - Aerosol time-series measurements over

the tropical Northeast Atlantic Ocean: dust sources, elemental composition and mineralogy, Marine Chemistry, 2015. Pey, J., Querol, X., Alastuey, A., Forastiere, F., and Stafoggia, M.: African dust outbreaks over the Mediterranean Basin during 2001–2011: PM10 concentrations, phenomenology and trends, and its relation with synoptic and mesoscale meteorology, Atmos. Chem. Phys., 13, 1395–1410, doi:10.5194/acp-13-1395-2013, 2013. Vincent, J., Laurent, B., Losno, R., Bon Nguyen, E., Roullet, P., Sauvage, S., Chevaillier, S., Coddeville, P., Ouboulmane, N., di Sarra, A. G., Tovar-Sanchez, A., Sferlazzo, D., Massanet, A., Triquet, S., Morales Baquero, R., Fornier, M., Coursier, C., Desboeufs, K., Dulac, F., and Bergametti, G.: Variability of mineral dust deposition in the western Mediterranean basin and south-east of France, Atmos. Chem. Phys., 16, 8749-8766, 10.5194/acp-16-8749-2016, 2016. AA. All of these references are no cited and listed in the reference section.

Please also note the supplement to this comment:
https://www.atmos-chem-phys-discuss.net/acp-2018-211/acp-2018-211-AC2-supplement.pdf

---

## Author Comment (AC3) · 8 Jun 2018

Please see attached pdf file with changes to the Ms

Please also note the supplement to this comment:
https://www.atmos-chem-phys-discuss.net/acp-2018-211/acp-2018-211-AC3-supplement.pdf

2018.

---

## Author Comment (AC4) · 8 Jun 2018

**Supplementary Material**

**Mineralogy and physicochemical features of Saharan dust wet deposited in the Iberian Peninsula during an extreme red rain event**

5  Carlos Rodriguez-Navarro, Fulvio di Lorenzo, and Kerstin Elert

Department Dept. Mineralogy and Petrology, University of Granada, Fuentenueva s/n, 18002 Granada, Spain

*Correspondence to*: Carlos Rodriguez-Navarro (carlosrn@ugr.es)

10 **Content**

**Supplementary Tables S1-S3**
**Supplementary Figures S1-S8**

15

20

25

**Table S1. Structural formulae of clay minerals (all analyses) in Saharan dust from TEM-AEM analysis**

| | Si | Al IV | Al VI | Mg | Fe | Sum Oct[1] | K | Ca | Na | Sum Int.[2] |
|---|---|---|---|---|---|---|---|---|---|---|
| **Illite based on $O_{10}(OH)_2$** | | | | | | | | | | |
| 1 | 3.53 | 0.47 | 1.69 | 0.19 | 0.14 | 2.03 | 0.58 | 0.00 | 0.00 | 0.58 |
| 2 | 3.39 | 0.61 | 1.29 | 0.20 | 0.60 | 2.09 | 0.34 | 0.11 | 0.12 | 0.67 |
| 3 | 3.34 | 0.66 | 1.43 | 0.42 | 0.32 | 2.17 | 0.56 | 0.00 | 0.00 | 0.56 |
| 4 | 3.33 | 0.67 | 1.20 | 0.40 | 0.43 | 2.03 | 0.51 | 0.00 | 0.00 | 0.51 |
| 5 | 3.35 | 0.65 | 1.65 | 0.30 | 0.22 | 2.17 | 0.44 | 0.00 | 0.00 | 0.44 |
| Aver. | 3.46 | 0.54 | 1.49 | 0.20 | 0.37 | 2.06 | 0.46 | 0.06 | 0.06 | 0.63 |
| Std. | ±0.10 | ±0.10 | ±0.28 | ±0.01 | ±0.33 | ±0.04 | ±0.17 | ±0.08 | ±0.08 | ±0.06 |
| **Palygorskite based on $O_{10}OH$** | | | | | | | | | | |
| 1 | 4.41 | 0.00 | 0.70 | 0.54 | 0.07 | 1.30 | 0.00 | 0.00 | 0.00 | |
| 2 | 3.85 | 0.00 | 0.98 | 0.34 | 0.59 | 1.92 | 0.12 | 0.03 | 0.00 | |
| 3 | 3.95 | 0.00 | 0.83 | 0.73 | 0.41 | 1.96 | 0.05 | 0.00 | 0.00 | |
| 4 | 4.11 | 0.00 | 0.82 | 0.57 | 0.28 | 1.67 | 0.05 | 0.05 | 0.00 | |
| 5 | 4.25 | 0.00 | 0.61 | 0.80 | 0.16 | 1.57 | 0.07 | 0.02 | 0.00 | |
| 6 | 4.22 | 0.00 | 0.35 | 1.33 | 0.12 | 1.79 | 0.07 | 0.00 | 0.00 | |
| 7 | 3.94 | 0.00 | 0.80 | 1.00 | 0.24 | 2.04 | 0.05 | 0.04 | 0.00 | |
| 8 | 4.22 | 0.00 | 0.55 | 1.34 | 0.18 | 2.07 | 0.00 | 0.06 | 0.00 | |
| Aver. | 4.08 | 0.00 | 0.63 | 1.10 | 0.24 | 1.97 | 0.04 | 0.03 | 0.00 | |
| Std. | ±0.17 | ±0.00 | ±0.25 | ±0.42 | ±0.19 | ±0.11 | ±0.04 | ±0.03 | ±0.00 | |
| **Kaolinite based on $O_5(OH)_4$** | | | | | | | | | | |
| 1 | 2.02 | 0.00 | 1.82 | 0.00 | 0.15 | 0.00 | 0.00 | 0.00 | 0.00 | |
| 2 | 2.02 | 0.00 | 1.92 | 0.00 | 0.03 | 0.00 | 0.00 | 0.03 | 0.00 | |
| 3 | 2.01 | 0.00 | 1.98 | 0.00 | 0.02 | 0.00 | 0.00 | 0.00 | 0.00 | |
| 4 | 1.98 | 0.00 | 2.01 | 0.00 | 0.02 | 0.00 | 0.00 | 0.00 | 0.00 | |
| 5 | 2.01 | 0.00 | 1.87 | 0.07 | 0.06 | 0.00 | 0.00 | 0.01 | 0.00 | |
| 6 | 1.98 | 0.00 | 1.96 | 0.01 | 0.06 | 0.00 | 0.00 | 0.00 | 0.00 | |
| 7 | 2.05 | 0.00 | 1.82 | 0.00 | 0.88 | 0.00 | 0.00 | 0.00 | 0.00 | |
| Aver. | 2.01 | 0.00 | 1.91 | 0.01 | 0.06 | 0.00 | 0.00 | 0.01 | 0.00 | |
| Std. | ±0.02 | ±0.00 | ±0.08 | ±0.03 | ±0.05 | ±0.00 | ±0.00 | ±0.01 | ±0.00 | |
| **Smectite based on $O_{10}(OH)_2$** | | | | | | | | | | |
| 1 | 4.01 | 0.00 | 1.54 | 0.09 | 0.26 | 1.89 | 0.14 | 0.14 | 0.00 | 0.42 |
| 2 | 3.62 | 0.38 | 1.21 | 0.31 | 0.56 | 2.08 | 0.21 | 0.12 | 0.00 | 0.45 |
| 3 | 3.71 | 0.29 | 1.32 | 0.19 | 0.52 | 2.03 | 0.16 | 0.12 | 0.00 | 0.40 |

| | | | | | | | | | | |
|---|---|---|---|---|---|---|---|---|---|---|
| 4 | 3.51 | 0.49 | 1.59 | 0.21 | 0.37 | 2.16 | 0.17 | 0.02 | 0.00 | 0.20 |
| 5 | 3.38 | 0.62 | 1.64 | 0.16 | 0.30 | 2.09 | 0.09 | 0.21 | 0.00 | 0.51 |
| 6 | 3.84 | 0.16 | 1.51 | 0.22 | 0.26 | 2.00 | 0.05 | 0.17 | 0.00 | 0.39 |
| 7 | 3.64 | 0.36 | 1.51 | 0.37 | 0.28 | 2.16 | 0.15 | 0.06 | 0.00 | 0.27 |
| 8 | 3.55 | 0.45 | 1.82 | 0.15 | 0.16 | 2.13 | 0.15 | 0.03 | 0.00 | 0.21 |
| 9 | 3.42 | 0.58 | 1.73 | 0.25 | 0.23 | 2.21 | 0.11 | 0.05 | 0.00 | 0.21 |
| 10 | 3.65 | 0.37 | 1.01 | 0.89 | 0.47 | 2.37 | 0.11 | 0.00 | 0.00 | 0.11 |
| Aver. | 3.58 | 0.42 | 1.54 | 0.23 | 0.34 | 2.11 | 0.14 | 0.10 | 0.00 | 0.33 |
| Std. | ±0.15 | ±0.15 | ±0.20 | ±0.08 | ±0.14 | ±0.07 | ±0.05 | ±0.07 | ±0.00 | ±0.12 |
| Illite-smectite mixed layer based on $O_{10}(OH)_2$ | | | | | | | | | | |
| 1 | 3.25 | 0.75 | 1.77 | 0.26 | 0.24 | 2.27 | 0.14 | 0.02 | 0.00 | 0.18 |
| 2 | 3.57 | 0.43 | 0.93 | 0.12 | 0.92 | 1.98 | 0.24 | 0.19 | 0.00 | 0.62 |
| 3 | 3.71 | 0.29 | 0.91 | 0.35 | 0.77 | 2.03 | 0.31 | 0.12 | 0.00 | 0.55 |
| 4 | 3.66 | 0.34 | 1.57 | 0.42 | 0.13 | 2.12 | 0.30 | 0.05 | 0.00 | 0.40 |
| Aver. | 3.46 | 0.54 | 1.67 | 0.34 | 0.19 | 2.20 | 0.22 | 0.04 | 0.00 | 0.29 |
| Std. | ±0.29 | ±0.29 | ±0.14 | ±0.11 | ±0.08 | ±0.11 | ±0.11 | ±0.02 | ±0.00 | ±0.16 |

[1] Sum of octahedral cations

[2] Sum of interlayer charge ($M^+ + M^{2+}$)

Structural formulae in red are not fully reliable (either due to too high or too low Sum oct., or excessive Fe -due to possible contamination with iron oxyhydroxide nanoparticles-). Therefore, they are not considered for the calculation of the average structural formulae.

15

20

**Table S2**. **Results of PHREEQC geochemical modeling of Saharan dust dissolution in rainwater (for an equilibrium pH = 8)**

| Phases | Molecular weight | SI | Initial amount (mol/L) | Amount dissolved (mol/L) | % dissolved |
|---|---|---|---|---|---|
| ferrihydrite | 106.866 | 0 | 7.58E-04 | 2.32E-06 | 0.31 |
| goethite | 88.851 | 5.3 | 1.71E-03 | 0 | 0 |
| hematite | 159.687 | 11.53 | 2.79E-04 | 0 | 0 |
| kaolinite | 258.155 | 3.3 | 2.27E-03 | 0 | 0 |
| quartz | 60.083 | 1.21 | 3.41E-02 | 0 | 0 |
| calcite | 100.086 | 0 | 1.07E-02 | 2.47E-04 | 2.3 |
| dolomite | 184.399 | 1.44 | 9.67E-04 | 0 | 0 |
| plagioclase (albite) | 278.202 | 2.1 | 2.24E-03 | 0 | 0 |
| illite | 383.893 | 2.3 | 5.13E-03 | 0 | 0 |
| k feldspar (microcline) | 278.326 | 0 | 1.92E-03 | 2.59E-08 | 0.001 |
| rutile | 79.865 | 0 | 8.93E-04 | 2.26E-10 | 2E-5 |
| smectite (high Mg-Fe) | 404.191 | 0 | 1.56E-03 | 3.99E-11 | 2.5E-6 |
| sepiolite (for palygorskite) | 647.819 | 0 | 2.45E-03 | 1.29E-04 | 5.2 |

**Note**: A value of SI = 0 means that the specific phase dissolved until equilibrium (saturation). Conversely, values of SI > 0 mean that the specific phase(s) is highly supersaturated and cannot dissolve.

**Table S3. Calculation of the total structural Fe in the clay minerals of the Saharan dust**

| | Fractional amount in bulk *From XRD results (Table 2)* | Fe (mol per formula unit) *Based on one Si/formula unit* | Fe in clays (wt%) | Fe in bulk dust (wt%) |
|---|---|---|---|---|
| Illite | 0.145 | 0.093 | 5.18 | 0.75 |
| palygorskite | 0.170 | 0.060 | 3.36 | 0.57 |
| smectite | 0.047 | 0.085 | 4.76 | 0.22 |
| kaolinite | 0.082 | 0.030 | 1.68 | 0.14 |
| MLC | 0.010 | 0.048 | 2.66 | 0.03 |
| Chlorite | 0.019 | 0.250 | 14.00 | 0.27 |
| | | | Average Fe in clays | Total structural Fe |
| | | | 5.27 | **1.98** |

**Note**: the content of structural iron in each clay was determined by: (1) calculating the fractional amount of each clay mineral in the bulk dust (from Table 2); (2) dividing the amount (in moles) of Fe determined by TEM-AEM analysis (structural formulae in Table 4) by the total moles of tetrahedral cations in the structural formula of each clay mineral; (3) the resulting value was multiplied by the atomic weight of Fe (56), yielding the Fe present (wt%) in each clay mineral ; (4) this later value was in turn multiplied by the fractional amount of each clay mineral, thereby yielding the contribution of each clay to the "Fe in bulk dust (wt%)", and the total structural Fe.

[Figure]

5    **Figure S1.** Daily synoptic scale meteorological situation during the days before, during and after the red rain event based on NOAA/ESRL reanalysis for 850 hPa geopotential height. For each date, paired maps of geopotential height and wind field are presented.

15

[Figure]

5    **Figure S2.** Synoptic scale meteorological situation during the days preceding the development of the north African cyclone responsible for the extreme red rain event (NOAA/ESRL reanalysis). Averaged geopotential heigh (a) and wind field (b), and temperature at 300 hPa (c) and at 925 hPa (d) during February 18-19, 2017.

[Figure]

**Figure S3.** Semiquantitative XRD analysis of a Saharan dust sample collected during the extreme red rain event and performed using the Rietveld full pattern profile method. In addition to the experimental (red) and modeled (bleu) pattern profiles (upper left corner image), the residual curve is shown (lower left) as well as the results of the semiquantitaive analysis (graph on the right).

15

[Figure]

**Figure S4.** STEM-HAADF photomicrographs and corresponding EDX elemental maps of a clay-rich micrometer-sized aggregate of
10 Saharan dust particles. Based on the compositional analysis, particles of palygorskite (Pal), calcite (Cal), aggregates of iron oxyhydroxide
nanoparticle (Fe-rich), rutile (Rut), illite (Il), smectite (Sm), and kaolinite (Kao) were identified. The EDX spectrum corresponds to Fe-rich
nanoparticles attached onto the clay minerals (analysis #3 in HAADF image). Note: analyzed spots are outlined with green or yellow
(numbered) rectangles in the HAADF image.

[Figure]

**Figure S5.** STEM-HAADF photomicrographs and corresponding EDX elemental maps of a micrometer-sized aggregate of Saharan dust particles. Based on the compositional analysis, particles of palygorskite (Pal), calcite (Cal), goethite/hematite (Fe-Ox), aggregates of iron oxyhydroxide nanoparticle (Fe-rich), rutile (Rut), illite (Il), smectite (Sm), and kaolinite (Kao) were identified. The EDX spectrum corresponds to calcite in contact with Fe-rich nanoparticles (analysis #5 in HAADF image). Note the absence of S and N. Note: analyzed spots are outlined with green or white (numbered) rectangles in the HAADF image.

[Figure]

**Figure S6.** Time-evolution of the pH and free calcium concentration [Ca] following dissolution of Saharan dust in MilliQ water with a starting pH of 5.6. The green triangle shows the [Ca] value calculated from PHREEQC modeling.

[Figure]

4.spx  Date:11/6/2017 10:39:45 AM     HV:300.0kV  Puls th.:1.70kcps

**Figure S7. AEM-**EDX spectrum of an individual calcite grain in Saharan dust (with no sign of Fe contamination by ferrihydrite nanoparticles). Note the absence of Fe in the carbonate (the Cu signal is due to the TEM Cu grid).

[Figure]

**Figure S8.** TEM photomicrographs and EDX microanalysis of palygorskite fibers in Saharan dust. a) elongated single fiber of palygorkite with attached aluminosilicate particles (feldspars and clay minerals); b) Palygorskite fibers. The red circle points to Fe-rich (dark) nanoparticles attached to the palygorskite fibers; c) EDX spectrum of the Fe-rich nanoparticles in the circled area of (b). The Mg, Al, and
10    Si (along with small amounts of Ca) correspond to the palygoskite substrate,